# Large-scale changes of the semidiurnal tide over the North Atlantic coasts from 1846 to 2018

Lucia Pineau-Guillou[1], Pascal Lazure[1], and Guy Wöppelmann[2]

[1]IFREMER, CNRS, IRD, UBO, Laboratoire d'Océanographie Physique et Spatiale, UMR 6523, IUEM, Brest, France
[2]LIENSS, Université de la Rochelle-CNRS, La Rochelle, France

**Correspondence:** Lucia Pineau-Guillou (lucia.pineau.guillou@ifremer.fr)

**Abstract.**

We investigated the long-term changes of the principal tidal component $M_2$ over the North Atlantic coasts, from 1846 to 2018. We analysed 18 tide gauges with time series starting no later than 1940. The longest is Brest with 165 years of observations. We carefully processed the data, particularly to remove the 18.6-year nodal modulation. We found that $M_2$ variations are consistent at all the stations in the North East Atlantic (Cuxhaven, Delfzijl, Hoek van Holland, Newlyn, Brest), whereas some discrepancies appear in the North West Atlantic. The changes started long before the XX$^{th}$ century, and are not linear. The secular trends in $M_2$ amplitude vary from one station to another; most of them are positive, up to 2.5 mm/yr in the period since 1910. Since 1990, the trends switch from positive to negative values in the North East Atlantic. Concerning the possible causes of the observed changes, the similarity between the North Atlantic Oscillation and $M_2$ variations in the North East Atlantic suggests a possible influence of the large-scale atmospheric circulation on the tide. Our statistical analysis confirms large correlations at all the stations in the North East Atlantic. We discuss a possible underlying mechanism. A different spatial distribution of mean sea level (corresponding to water depth) from one year to another, depending on the low-frequency sea-level pressure patterns, could impact the propagation of the tide in the North Atlantic basin. However, the hypothesis is at present unproven.

## 1 Introduction

Tides have been changing due to non-astronomical factors since the XIX$^{th}$ century (Haigh et al., 2019; Talke and Jay, 2020). In the North Atlantic, secular variations have been observed at individual tide gauge stations, e.g. Brest (Cartwright, 1972; Wöppelmann et al., 2006; Pouvreau et al., 2006; Pouvreau, 2008), Newlyn (Araújo and Pugh, 2008; Bradshaw et al., 2016), New York (Talke et al., 2014), Boston (Talke et al., 2018), but also at regional scale, e.g. Gulf of Maine (Doodson, 1924; Godin, 1995; Ray, 2006; Ray and Talke, 2019), North Atlantic (Müller, 2011), and at quasi-global scale (Woodworth, 2010; Müller et al., 2011; Mawdsley et al., 2015). Long-term changes in tidal constituents are rather small at coastal stations, but tend to be statistically significant. The order of magnitude of these changes varies spatially, and may reach a few cm/century for $M_2$ amplitude. For example, Colosi and Munk (2006) reported changes of about 1 cm at Honolulu, Hawaii, between 1915 and 2000. Ray and Talke (2019) found trends varying from -1 to 8 cm/century in the Gulf of Maine over the last century.

Woodworth et al. (2010) and Müller et al. (2011) found trends of a few % per century in the Atlantic. The changes can be larger in many estuaries and rivers (Talke and Jay, 2020).

The physical causes of these changes can be multiple and difficult to disentangle. In particular, the complexity comes from the possible interaction between local and large-scale causes. Changes may have a local scale origin, such as changes in the

nearby environment (e.g. harbour development, deepening of channels, dredging, siltation) or changes in the instrumentation (e.g. tide gauge technology, observatory location, instrumental errors). For example, Familkhalili and Talke (2016) show that mean tidal range at Wilmington has doubled since the 1880s, due to channel deepening in the Cap Fear River Estuary. Changes may also have a large-scale origin, i.e. regional or global. Haigh et al. (2019) reported several possible large-scale mechanisms: (1) tectonics and continental drift, (2) water depth changes due to mean sea level rise or geological processes such as the Earth's

surface glacial isostatic adjustment (Müller et al., 2011; Pickering et al., 2017; Schindelegger et al., 2018), (3) shoreline position, (4) extent of sea-ice cover (Müller et al., 2014), (5) sea-bed roughness, (6) ocean stratification which may modify the internal tides and bottom friction over continental shelves (Müller, 2012), (7) non-linear interactions and (8) radiational forcing (Ray, 2009).

Several authors have explored Mean Sea Level (MSL) rise as a potential mechanism to explain $M_2$ changes. For example, simulations by Pickering et al. (2012) show that a 2m sea level rise could modify $M_2$ from -20 to 20 cm around the whole ocean. Idier et al. (2017) show that depending on the location, the changes can account for +/-15% of the regional sea level rise. Schindelegger et al. (2018) find changes of about 1–5% of the sea level rise. Beyond MSL rise, other mechanisms have been explored to explain $M_2$ changes. For example, Colosi and Munk (2006) attribute the changes of $M_2$ amplitude at Honolulu,

Hawai, to a 28° rotation of the internal tide vector in response to ocean warming. Ray and Talke (2019) suggest that long-term changes in stratification could play a role in the Gulf of Maine. Müller (2011) suggests a possible link between $M_2$ changes and atmospheric dynamics in the North Atlantic; he reported that the timeseries of the North Atlantic Oscillation (NAO) show similar characteristics to those of the tidal amplitudes and phases. In the Gulf of Maine, Pan et al. (2019) suggest that changes in the response of the nodal modulation of the $M_2$ tide from 1970s to 2013 may be linked with the NAO. In Southeast Asian

Waters, Devlin et al. (2018) show that the impact of atmospheric circulation (via the wind stress, through Ekman current) on the $M_2$ seasonal cycle may be significant and comparable to the effect of permanent (geostrophic) currents. In the North Sea, Huess and Andersen (2001) explain a large part of $M_2$ seasonal cycle by the role of atmospheric dynamics, whereas Müller et al. (2014) and Gräwe et al. (2014) suggest a major role of the thermal stratification. These examples show the diversity of mechanisms that play a role in tide changes. In the present paper, we focus on the role of MSL and atmospheric dynamics.

This paper has two main objectives. The first is to characterize the secular changes of the $M_2$ tide over the North Atlantic. We focus on the longest time series, i.e. starting no later than 1940. This approach is complementary to previous studies investigating $M_2$ changes focusing on smaller spatial scales, e.g. Brest (Pouvreau et al., 2006; Pouvreau, 2008), Gulf of Maine (Ray, 2006; Ray and Talke, 2019), or focusing on shorter temporal scales, i.e. recent decades (Woodworth, 2010; Müller, 2011). The

second objective is to detect if there is any large-scale coherence in the observed changes in the North Atlantic, and investigate the possible link with the atmospheric circulation, already mentioned by Müller et al. (2011), on the basis of qualitative criteria. Here, we further bring quantitative insights on the possible influence of the NAO, and discuss a possible NAO-related climate mechanism that can partly explain the observed changes.

The paper is organised as follows. The first section describes the data: the sea level data (i.e. tide gauges and their processing) and the atmospheric data (i.e. climate indices and sea level pressure data). The following section presents the results (i.e. $M_2$ variations and trends). We then discuss a possible link between the observed tidal changes and MSL, as well as climate indices.

## 2   Data

### 2.1   Sea level data

#### 2.1.1   Tide gauges selection

The tide gauge data were retrieved from the University of Hawaii Sea Level Center (UHSLC, website accessed April 2020). The dataset consists of 249 stations in the Atlantic Ocean, with hourly sea level observations. Two additional long-term stations - Delfzijl and Hoek van Holland - were provided by Rijkswaterstaat (RWS).

We selected the stations following three criteria: time series (1) starting before 1940, (2) with at least 80 years of data, (3) with tidal amplitude significant enough to detect trends, i.e. $M_2$ amplitude larger than 10 cm. Note that we selected only years with at least 75% of data (see section 2.1.2). Only 24 stations among the 249 followed the two first criteria (Figure 1). They are all located in the northern hemisphere. On the east side of the North Atlantic, Stockholm, Gedser, Hornbaek, Tregde and Marseille were discarded due to too small an $M_2$ amplitude (i.e. lower than 10 cm). These stations are located in the Baltic Sea (Stockholm, Gedser), in the strait separating the Baltic and the North Sea (Hornbaek), in the North Sea (Tregde), and in the Mediterranean Sea (Marseille). On the west side of the North Atlantic, Galveston, Pensacola and Cristobal were also discarded due to too small a tidal amplitude (i.e. lower than 10 cm). These stations are located in the Gulf of Mexico (Galveston, Pensacola) and the Caribbean Sea (Cristobal).

Finally, 18 stations followed the three criteria detailed above, and were selected for this study (see stations in bold in Figure 1, 16 stations are from UHSLC, and 2 from RWS). Among them, 5 are located on the North East Atlantic coasts (Newlyn, Brest, Hoek van Holland, Delfzijl and Cuxhaven - note that Hoek van Holland, Delfzijl and Cuxhaven are located in the North Sea) and 13 are located on the North West Atlantic coasts (Halifax, Eastport, Portland, Boston, Newport, New London, New York, Atlantic City, Lewes, Wilmington, Charleston, Fort Pulaski and Key West).

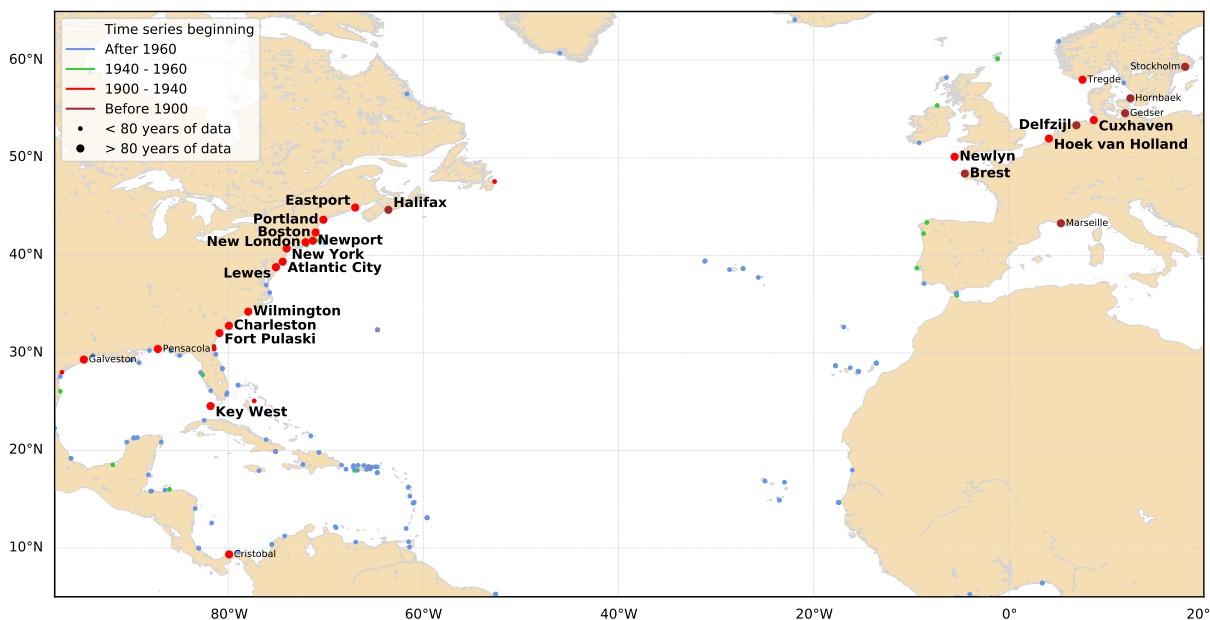

**Figure 1.** Tide gauges in the North Atlantic. Stations with time series starting before 1940 and longer than 80 years are labelled. Stations selected for this study are in bold.

The main characteristics of the 18 selected stations are summarised in Table 1. Among them, only Brest, Hoek van Holland and Halifax started in the XIX$^{th}$ century, respectively in 1846, 1879 and 1896 (Table 1, column 2). The number of years with data for each station varies between 81 and 165 years, Brest being the longest time series (Table 1, column 3).

### 2.1.2 Data processing

Harmonic analysis was performed in order to compute the $M_2$ amplitude. We used the MAS program (Simon, 2007, 2013), developed by the French Hydrographic Office (SHOM). This program gives results similar to the T_Tide harmonic analysis toolbox (Pawlowicz et al., 2002). For instance, Pouvreau et al. (2006) found no differences in the yearly amplitudes of $M_2$ at Brest over the period 1846 to 2005 using either T_Tide or MAS. Hourly time series were analysed yearly. Note that at Delfzijl and Hoek van Holland, data had to be interpolated every hour before 1970, as the temporal sampling was 3 hours. We

processed only years with at least 75% of data, to avoid $M_2$ seasonal modulation. In the North Atlantic, $M_2$ is affected by a seasonal variation of a few percent (Pugh and Vassie, 1976; Huess and Andersen, 2001; Müller et al., 2014; Gräwe et al., 2014). Considering only years with at least 75% of data resulted in excluding up to 15 years for a given station (Table 1, columns 3 and 4). We carefully removed the nodal modulation of $M_2$ amplitude (Simon, 2007, 2013), as described briefly in Appendix A. Finally, 3 station-years were discarded due to problems in the record (1953 and 1962 at Delfzijl, 1953 at Hoek van Holland),

and 2 more station-years due to doubtful $M_2$ values (1972 at Eastport, 1978 at Newport).

**Table 1.** Main characteristics of tide gauge records selected for this study. Name of the station, timespan, number of years with data, number of years analysed (i.e. with at least 75% of data), $M_2$ average amplitude and standard deviation over the period 1910-2010, $M_2$ nodal modulation, estimated trends in $M_2$ amplitude since 1910 and since 1990 (standard errors are 1-sigma).

| Name | Timespan | Nb of yrs with data | Nb of yrs analysed | $\overline{M_2}$ (cm) [1910-2010] | $M_2$ nod. mod. $f_{nod}$ | $M_2$ trends since 1910 (mm/yr) | $M_2$ trends since 1990 (mm/yr) |
|---|---|---|---|---|---|---|---|
| Cuxhaven | 1918-2018 | 102 | 101 | $135.05 \pm 3.68$ | 1.8 % | $0.68 \pm 0.10$ | $-0.47 \pm 0.41$ |
| Delfzijl | 1879-2018 | 138 | 138 | $125.58 \pm 6.96$ | 1.7 % | $2.02 \pm 0.09$ | $-0.09 \pm 0.24$ |
| Hoek van Holland | 1900-2018 | 88 | 82 | $76.95 \pm 2.63$ | 0.8 % | $0.85 \pm 0.06$ | $-0.45 \pm 0.14$ |
| Newlyn | 1916-2016 | 102 | 98 | $170.66 \pm 0.75$ | 3.3 % | $0.14 \pm 0.02$ | $-0.28 \pm 0.14$ |
| Brest | 1846-2018 | 165 | 158 | $204.54 \pm 0.91$ | 3.8 % | $0.13 \pm 0.02$ | $-0.36 \pm 0.12$ |
| Halifax | 1896-2012 | 99 | 95 | $62.83 \pm 0.64$ | 3.7 % | $-0.15 \pm 0.02$ | $0.32 \pm 0.17$ |
| Eastport | 1930-2018 | 90 | 82 | $263.51 \pm 2.50$ | 2.5 % | $0.80 \pm 0.07$ | $1.01 \pm 0.39$ |
| Portland | 1910-2018 | 109 | 104 | $135.07 \pm 1.84$ | 2.8 % | $0.56 \pm 0.03$ | $0.72 \pm 0.20$ |
| Boston | 1922-2018 | 98 | 96 | $136.57 \pm 1.03$ | 2.9 % | $0.27 \pm 0.03$ | $0.42 \pm 0.20$ |
| Newport | 1931-2018 | 89 | 84 | $50.86 \pm 0.41$ | 4.1 % | $-0.09 \pm 0.01$ | $-0.03 \pm 0.08$ |
| New London | 1939-2018 | 81 | 76 | $35.93 \pm 0.25$ | 3.5 % | $0.06 \pm 0.01$ | $0.03 \pm 0.05$ |
| New York | 1921-2018 | 95 | 80 | $65.13 \pm 0.83$ | 3.7 % | $0.33 \pm 0.02$ | $0.93 \pm 0.12$ |
| Atlantic City | 1912-2018 | 107 | 101 | $58.48 \pm 0.31$ | 3.8 % | $0.00 \pm 0.01$ | $-0.18 \pm 0.07$ |
| Lewes | 1919-2018 | 85 | 72 | $59.91 \pm 0.43$ | 3.1 % | $-0.06 \pm 0.02$ | $-0.33 \pm 0.06$ |
| Wilmington | 1936-2018 | 84 | 82 | $56.84 \pm 6.16$ | 1.7 % | $2.51 \pm 0.09$ | $1.80 \pm 0.20$ |
| Charleston | 1901-2018 | 101 | 100 | $76.40 \pm 1.33$ | 3.0 % | $0.32 \pm 0.03$ | $-0.02 \pm 0.08$ |
| Fort Pulaski | 1936-2018 | 84 | 78 | $100.60 \pm 1.01$ | 3.1 % | $0.18 \pm 0.04$ | $-0.01 \pm 0.17$ |
| Key West | 1913-2018 | 106 | 104 | $17.50 \pm 0.36$ | 2.9 % | $0.08 \pm 0.01$ | $0.13 \pm 0.02$ |

At all the stations, we computed the normalized $M_2$ amplitude, removing the average and dividing by the standard deviation over the period 1910-2010

$$Normalized\ M_2(t) = \frac{M_2(t) - \overline{M_2}_{[1910,2010]}}{\sigma_{M_2[1910,2010]}} \tag{1}$$

the average $\overline{M_2}$ and standard deviation $\sigma_{M_2}$ over the 1910-2010 period being given in Table 1 (column 5). The idea is to scale the data, in order to compare all the stations together.

## 2.2 Atmospheric data

### 2.2.1 Climate indices

We investigated the correlation between secular changes in the tide and climate indices, such as the North Atlantic Oscillation (NAO) or the Arctic Oscillation (AO) - also called Northern Annular Mode (NAM) (Hurrell, 1995; Hurrell and Deser, 2009; Thompson and Wallace, 2000; Thompson et al., 2000). These climate indices are related to the distribution of atmospheric masses. They are based on the difference of average sea-level pressure between two centres of actions (i.e. stations) over long periods (e.g. monthly, seasonal, annual).

The NAO is the major pattern of weather and climate variability over the Northern Hemisphere (Hurrell, 1995; Hurrell and Deser, 2009). Variations of NAO drive the climate variability over Europe and North America (Hurrell et al., 2003). We used the wintertime (December to March) Hurrell station-based NAO Index (retrieved from https://climatedataguide.ucar.edu/climate-data/hurrell-north-atlantic-oscillation-nao-index-station-based). It is based on the difference of normalized average winter sea-level pressure between Lisbon (Portugal) and Stykkisholmur/Reykjavik (Iceland). The normalization involves removing the mean (1864–1983) and dividing by the long-term standard deviation. The NAO index covers the period 1864-2019, with yearly values.

The Artic Oscillation (AO) is another index which resembles to NAO index. It is defined as the first EOF of northern hemisphere winter sea-level pressure data (Thompson and Wallace, 1998, 2000; Thompson et al., 2000). The AO index is highly correlated with the NAO. We used the wintertime Hurrell AO index (retrieved from https://climatedataguide.ucar.edu/climate-data/hurrell-wintertime-slp-based-northern-annular-mode-nam-index). The AO index covers the period 1899-2019.

To remove the interanual variability and estimate low frequency variations, climate indices were low-pass filtered with a 9-year mean filter.

### 2.2.2 Sea level pressure

We employed the Twentieth Century Reanalysis (20CR version 3 dataset) (Compo et al., 2011; Slivinski et al., 2019), a historic weather reconstruction from 1836 to 2015, with a 1° gridded global coverage. We computed the mean winter (December to February) sea-level pressure at each grid point over the period 1850-2015. We averaged from 1850 rather than 1836 (20CR starting date) to be consistent with the temporal coverage of the tide gauge measurements. We also computed yearly anomalies, i.e. removing the average sea-level pressure at each grid point.

## 3 Results

### 3.1 $M_2$ variations

For the North East Atlantic, the variations of normalized $M_2$ amplitude are presented in Figure 2 (a).

The first result is that since 1910, the variations show similar patterns at all the stations; $M_2$ amplitude decreases up to the 1960s, then increases, and decreases again from the 1990s. This suggests that these changes are probably due to large-scale processes, rather than local effects due to changes in the environment (e.g. harbor development, dredging, siltation) or instrumentation errors. The similar patterns between Brest and Cuxhaven may be surprising, as Cuxhaven is located in the North Sea, and not in the open Atlantic Ocean, and far away from Brest, around 1300 km. This indicates that the spatial scale of

the processes responsible for these changes must be at least as large as the North East Atlantic. Different authors have noticed the increase of tidal range from 1960 to 1990 in the southern North Sea. Hollebrandse (2005) found a gradual increase during the period 1955-1980 at all the stations of the Dutch coast (5 stations including Hoek van Holland) and the German coast (7 stations). Mudersbach et al. (2013) found a significant increase in $M_2$ amplitude at Cuxhaven since around the mid-1950s. Note that Cuxhaven is located in the German Bight; shallow depths and the shape of the coastline may induce some amplification.

Variations in $M_2$ at Cuxhaven are therefore sensitive to local effects, such as the migration of the underwater channels and the evolution of the tidal flats (Jacob et al., 2016). Moreover, Cuxhaven is located in the Elbe estuary, and some river engineering works, such as narrowing and deepening, may induce tidal amplification (Winterwerp and Wang, 2013; Winterwerp et al., 2013).

Before 1910, normalized $M_2$ values are higher at Brest than at Delfzijl. The construction of dykes that have gradually closed

the harbor of Brest since the end of the XIX$^{th}$ century may have altered the tide at Brest. The high values before 1910 may be due to local changes, in addition to large-scale changes. To go further, the potential role of these successive constructions needs to be investigated (https://en.wikipedia.org/wiki/Brest_Arsenal). Cartwright (1972) made a first attempt to evaluate the influence of reducing the width of access to the harbour but did not take into account a potential role of dredging for which we have no information. This example underlines the complexity of interpretation of the variations when changes of local and

large-scale origin occur at the same time. Note that in the following, we focus mainly on the XX$^{th}$ century, as most of the stations start after 1900 (15 out of 18 stations).

    The second result, is that there is no obvious linear trend in $M_2$ variations, but rather break or change points, $M_2$ increasing and then decreasing, depending on the periods considered. Overall, $M_2$ decreases from 1910 until 1960, increases again until

1980-1990, to finally decrease since 1990; note that the curve flattens between 1920 and 1940. Pouvreau et al. (2006) already noticed these variations at Brest and Newlyn, and suggested a long-period oscillation of around 140 years, rather than a steady secular trend. A careful analysis of the harmonic development of the tidal potential showed that no tidal component could explain this oscillation. Similarly, no linear combination of tidal harmonic components could explain it (Pouvreau et al., 2006). This indicates that these variations are not due to an astronomical component. However, in contrast to Brest, $M_2$ at Delfzijl

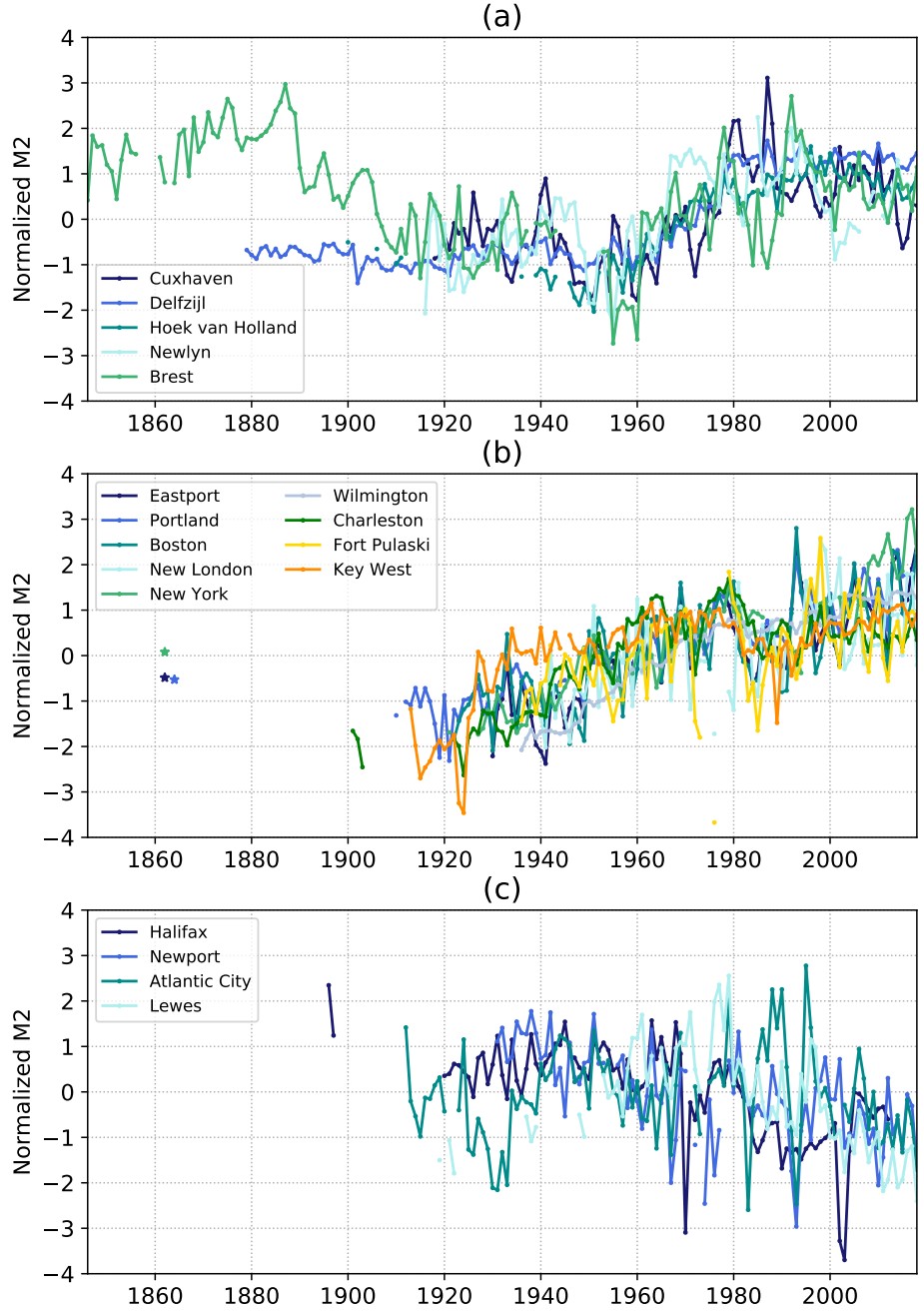

**Figure 2.** Normalized annual $M_2$ amplitude (a) in the North East Atlantic (b) in the North West Atlantic, stations with positive trends (c) in the North West Atlantic, stations with negative or no trend. The stars in (b) in the 1860s correspond to $M_2$ amplitude at Eastport and Portland from Ray and Talke (2019), and New York from Talke et al. (2014), after normalization (Eq. (1)).

.

stays flat between 1880 and 1920. The decrease observed at Brest between 1880 and 1920 may be due to harbour development and/or dredging (see above). This underlines the importance of sea level data archaelogy, for research studies related to long-term changes (Pouvreau, 2008; Woodworth et al., 2010; Marcos et al., 2011; Talke and Jay, 2013, 2017; Ray and Talke, 2019; Bradshaw et al., 2015, 2020).

The third result is that changes in $M_2$ have not the same order of magnitude at each station (see Figure B1 in Appendix B for time series of $M_2$). Note that Figure 2 represents normalized $M_2$, i.e. removing the average and dividing by the standard deviation. The order of magnitude of (not normalized) $M_2$ changes are roughly the same at Brest and Newlyn (standard deviations of 0.9 and 0.8 cm respectively, Table 1, column 5), but more than three times larger at Cuxhaven (standard deviation of 3.7 cm), and even larger at Delfzjil (standard deviation of 7 cm). This suggests that the North Sea may be more sensitive to

the processes responsible for these changes. Note also that the environmental setting of Cuxhaven and Delfzijl in the Elbe and Ems estuaries, respectively, could introduce some amplification (Winterwerp and Wang, 2013; Winterwerp et al., 2013).

For the North West Atlantic, the variations of normalized $M_2$ amplitude are presented in Figure 2 (b) and (c). The first feature is that $M_2$ amplitude varies differently in the North West and in the North East Atlantic. The second is that there are

discrepancies between stations, even when close to each other (e.g. Atlantic City and Lewes). We split the stations into two groups, in order to facilitate the detection of patterns, each being consistent in terms of trends: one with positive trend (group 1 in Figure 2 (b)), the other one with negative or no trend (group 2 in Figure 2 (c)).

The first group (with positive trends) consists of 9 stations (Figure 2 (b)). Three outcomes can be highlighted. The first is that

$M_2$ amplitude increases overall since 1900. However, between 1980 and 1990, all the stations slightly decrease, and since 1990 they increase again. The second outcome is that the rate of increase is very different from one station to another (keeping in mind that $M_2$ is normalized by standard deviation in Figure 2). Portland is increasing 1.4 times faster than Charleston (standard deviations being respectively of 1.82 and 1.33 cm), and 28 times faster than Key West (standard deviation being only 0.36 cm at Key West). The very slow increase of $M_2$ at Key West is due to a small tidal amplitude (i.e. only 17.5 cm mean amplitude

for $M_2$, see Table 1, column 5). The large increase in Portland may be explained by some amplification in the Gulf of Maine. In many semienclosed basins, resonance leads to tidal amplification (Talke and Jay, 2020; Haigh et al., 2019). In the Gulf of Maine, Ray and Talke (2019) reported that the tides in the Gulf are in resonance, with a natural resonance frequency close to the $N_2$ tide (Garrett, 1972; Godin, 1993). Tides may be then very sensitive to any changes in the environment (e.g. basin configuration - shape, depth - but also external forcing). The third outcome, and probably the most interesting one, is related

to the values of $M_2$ at Eastport, Portland and New York in the 1860s, estimated from Ray and Talke (2019) and Talke et al. (2014), and represented (after normalization) as stars on Figure 2 (b). These values are not consistent with the positive linear trends observed since 1900, which provides some consistency with the hypothesis formulated from the analyses of the data prior to the XX$^{th}$ century in Figure 2 (a): long-term variations introduce some breaks or change points, $M_2$ increasing and then decreasing, depending on the periods considered. The decrease observed between 1870s and 1920s at the four stations (Brest,

Eastport, Portland, New York) suggests a possible large-scale signal, in addition to local processes.

The second group (with negative or no trend) consists of 4 stations (Figure 2 (c)). Two points can be highlighted. The first is that $M_2$ decreases overall for Halifax, Newport and Lewes. This is less clear for Atlantic City, which is quite noisy and shows no significant trend. The second point is that at Halifax, $M_2$ values in 1896-1897 are higher than those after 1920. This suggests that the decrease may have started before the XX$^{th}$ century. Note that at Halifax, there is a long gap in the data

recording (1898-1919), which raises the possibility of an instrumentation origin in the observed decrease of the $M_2$ amplitude.

## 3.2 Estimated trends

We estimated the trends for $M_2$ amplitude at each station, using linear regression. We computed the trends over two periods: 1910-2018, which corresponds roughly to the whole period of data (only 5 stations start before 1910), and 1990-2018, which

corresponds to recent decades. Some tests showed that the later results were not very sensitive to the start date (moving 1990 to 1985 or 1995). The results are summarised in Table 1 (columns 7 and 8) and Figures 3 and 4.

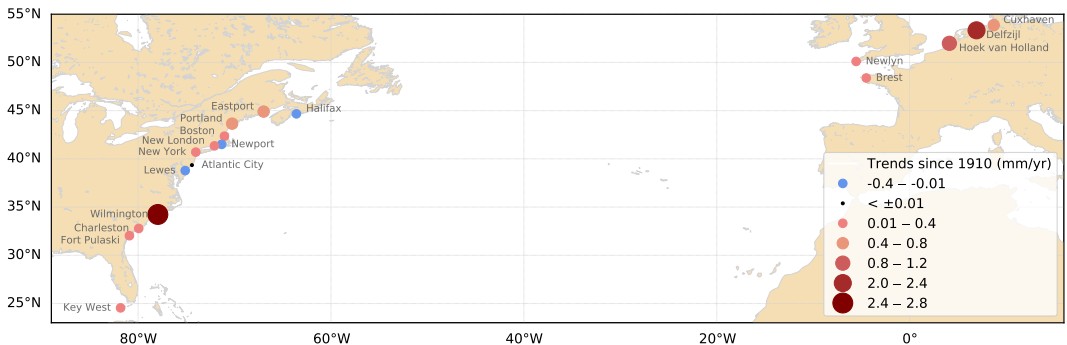

**Figure 3.** Estimated trends in $M_2$ amplitude over the period 1910-2018

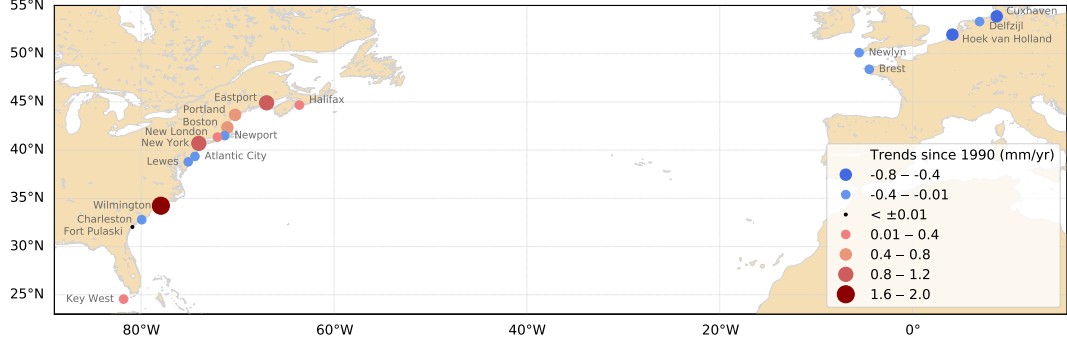

**Figure 4.** Estimated trends in $M_2$ amplitude over the period 1990-2018

The trends estimated from 1910 vary significantly from one station to another (Figure 3). They are positive overall (up to 2.5 mm/yr at Wilmington), which is consistent with previous findings (Araújo and Pugh, 2008; Ray, 2009; Woodworth, 2010; Müller et al., 2011; Ray and Talke, 2019). They are slightly negative at three stations (Halifax, Newport, Lewes), and one station shows no significant trend (Atlantic City). The estimates are statistically consistent with those found previously by different authors (e.g. $0.15 \pm 0.02$ mm/yr at Newlyn compared to $0.19 \pm 0.03$ mm/yr in Araújo and Pugh (2008), $0.56 \pm 0.03$ mm/yr in Portland, compared to $0.59 \pm 0.04$ mm/yr in Ray and Talke (2019)). In the North East Atlantic, the trends are consistent with each other (in terms of sign), which is not surprising as the stations vary similarly (Figure 2 (a)).

The trends estimated since 1990 are quite different from those estimated since 1910 (Figures 3 and 4), with more stations with negative trends: 9 stations out of 18 have post-1990 negative trends, whereas only 3 stations out of 18 have post-1910 negative trends (Table 1, columns 7 and 8). In the North East Atlantic, they all switch from positive to negative trends. This underlines (1) some recent changes in recent decades (Müller, 2011; Ray and Talke, 2019) and (2) the difficulty to estimate long-term trends from short records (i.e. less than 30 years), especially if the data are noisy (interannual variability) and the underlying processes non-linear (change points).

The largest trends since 1990 are mainly observed in semi-closed basins: Wilmington in the Cape Fear River Estuary, Delfzjil in Ems estuary, Cuxhaven in Elbe estuary, Eastport and Portland in the Gulf of Maine. This suggests a possible amplification due to resonance effects (e.g. Gulf of Maine) and/or propagation in shallow waters (e.g. Cuxhaven), in addition to local effects. The stations located in estuaries or in a harbour with a channel may have been subject to dredging. Channel deepening increases the water depths, which reduces the effective drag, and leads to tidal range amplification. This effect may be particularly large in estuaries (Ralston et al., 2019; Talke and Jay, 2020), and may explain the larger trends at Wilmington (Familkhalili and Talke, 2016) and Delzijl. Finally, the shifting locations of amphidromic points could also play a role (Haigh et al., 2019). In the North Sea, different authors show a possible migration of the present day amphidromes, under a 2 m sea-level rise scenario (Pickering et al., 2012; Idier et al., 2017).

The trends have to be interpreted very carefully as the $M_2$ variations are not linear, and may increase or decrease depending on the years; as a consequence, the estimated trends depend strongly on the period considered to estimate it. The interannual variability also plays an important role, and when substantial, trends can vary depending on the computational period. For example, at Cuxhaven, the large interannual variability leads to a large uncertainty on the trend computed since 1990 ($-0.47 \pm 0.41$ mm/yr).

# 4 Discussion

## 4.1 Possible link with mean sea level rise

The MSL rise could partly explain $M_2$ changes. Simulations show that MSL rise can result in an change of $M_2$ up to $\pm 10\%$ of the rise (Pickering et al., 2017; Idier et al., 2017; Schindelegger et al., 2018). Schindelegger et al. (2018) show that the sign of the observed $M_2$ trend is correctly reproduced at 80% of the tide gauges on a global scale, but their simulated trends tend to differ from observations by a factor 3 to 5, i.e. their simulations underestimate the $M_2$ response to MSL rise in terms of magnitude. Schindelegger et al. (2018) conclude that "magnitudes of observed and modeled $M_2$ trends are within a factor of

4 (or less) from each other in nearly 50% of the considered cases". The large discrepancies between the simulations and the observations strongly suggest that MSL rise is not the only process that may explain $M_2$ changes – other large-scale processes, in addition to local processes, may also play a role.

Figure 5 shows the annual MSL, after removing the average over the period 1910-2010, and filtering with a 9-year time

windows. The correlations between $M_2$ and MSL indicate that $M_2$ varies strongly with MSL (see section 4.2). However, $M_2$ variations show some variability in the North East Atlantic (Figure 2 (a)), which may not be explained with MSL rise alone.

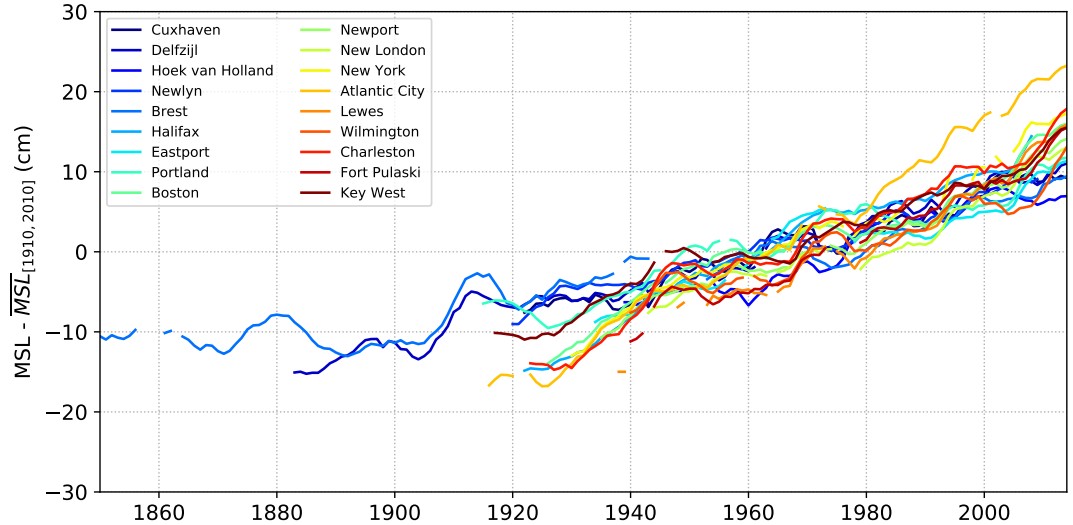

**Figure 5.** Annual Mean Sea Levels (MSL), after removing the average over the period 1910-2010. MSL values are filtered using 9-year windows.

## 4.2 Possible link with MSL and climates indices

Other processes than MSL rise may impact the tide (see section 1), such as the atmospheric circulation and the ocean stratification. Ocean and atmosphere are fully coupled, and air-sea fluxes are responsible for the exchange of momentum, water

(evaporation and precipitation budget) and heat at their interface. Among the wide range of possible interactions, two mechanisms have been explored for their ability to modify the tide. (1) The momentum flux (wind stress) and the gradient of sea level pressure which act on the barotropic tide and (2) the water and heat fluxes which induce changes in both temperature and salinity distribution in the ocean. The latter effect acts on the stratification which in turn could impact the tide in two different ways. The first way is the internal tide generation which transfers energy from barotropic and baroclinic motion and modifies surface tidal expression (Colosi and Munk, 2006). However, in the present study, most of the observations comes from coastal stations sheltered by wide continental shelves which dampen internal waves. More important is the second way: the stratification acts on the eddy viscosity profile by modifying currents profile and bottom drag over continental shelf, which in turn modify the $M_2$ surface expression (Kang et al., 2002; Müller, 2012; Katavouta et al., 2016).

Here, we focus on the effect of the atmospheric circulation on tide. We used pressure indices (NAO and AO) that are relevant to represent atmospheric circulation. The NAO index represents the difference of normalized sea level pressure between the Azores high pressure system and the Iceland low pressure one (Hurrell, 1995). It indicates the redistribution of atmospheric masses between the Subtropical Atlantic and the Arctic (Hurrell and Deser, 2009). In the North East Atlantic, the similarity between the variations of the low-frequency winter NAO index and those of $M_2$ (Figure 6) suggests a possible impact of large-scale atmospheric circulation on the tide. The NAO index varies from positive to negative phases. Filtering the interannual variability, the NAO index tends overall to decrease between 1910 and 1970, then increase until 1990, and once again decrease. The same way, $M_2$ amplitude tends to decrease up to 1960, then increase until 1990, and once again decrease. These similar patterns raise a possible connection between NAO and $M_2$ variation, already mentioned by Müller (2011) on the basis of qualitative criteria. In the following, we bring quantitative insights on the possible influence of NAO.

We computed the correlations (r-value) between normalized $M_2$ and climate indices, NAO and AO (Figure 7). $M_2$, NAO and AO are filtered using the same time window (9 years). The correlations are computed since 1910, to have similar periods for all the stations. The correlations are considered as significant only if the p-value is lower than 0.05 (95% significance level). The results are the following: (1) for NAO, 14 stations out of 18 show significant correlation. Note that at Brest, the correlation is significant since 1910, but not since 1864 (NAO index used in this study starts only in 1864). This can be explained by the $M_2$ larger amplitude over all the XIX$^{th}$ century, which decreases between 1890 and 1910 (Figure 2 (a)), possibly due to harbour development and dykes construction (see section 3.1). (2) In the North East Atlantic, all the stations are positively correlated with NAO. (3) The strongest correlations (i.e. greater than 0.5) are in the northern part of the North Atlantic, with strong positive correlations at Cuxhaven and Hoek van Holland, and strong negative correlation at Halifax (-0.55). (4) For AO, we found similar, but overall larger, r-values. This is not surprising as these two indices are highly correlated.

To go further in the relative contribution of MSL and NAO in $M_2$ variability, we fitted two linear regression models on $M_2$ variations. In the following, $M_2$, MSL and NAO are filtered over 9-year time windows and normalized. At all the stations, we

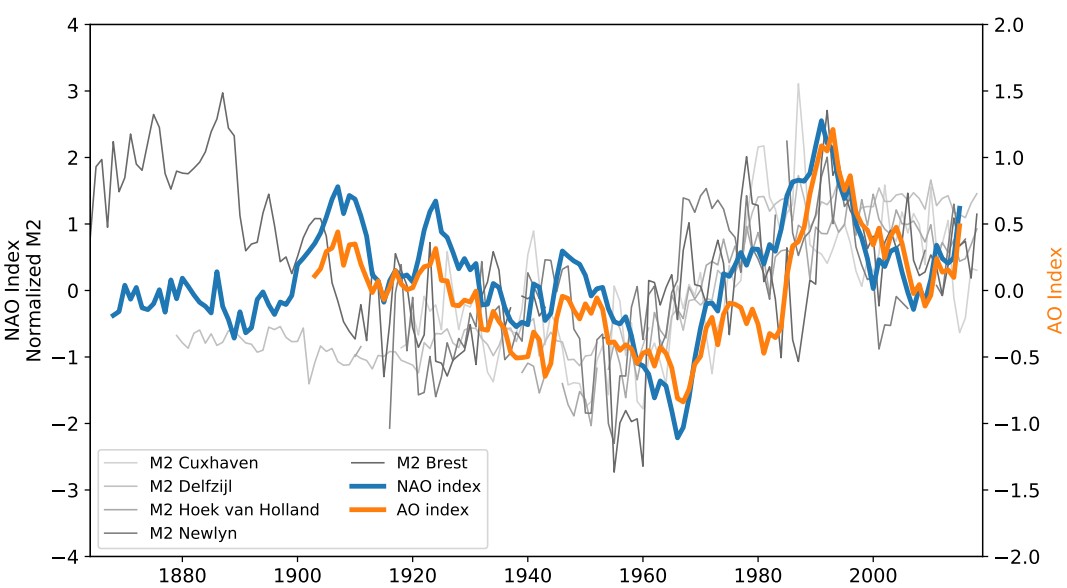

**Figure 6.** Low frequency winter NAO and AO indices, obtained with a 9-year mean filter. Normalized annual $M_2$ amplitudes in the North East Atlantic (from Figure 2 (a)) are also plotted in grey.

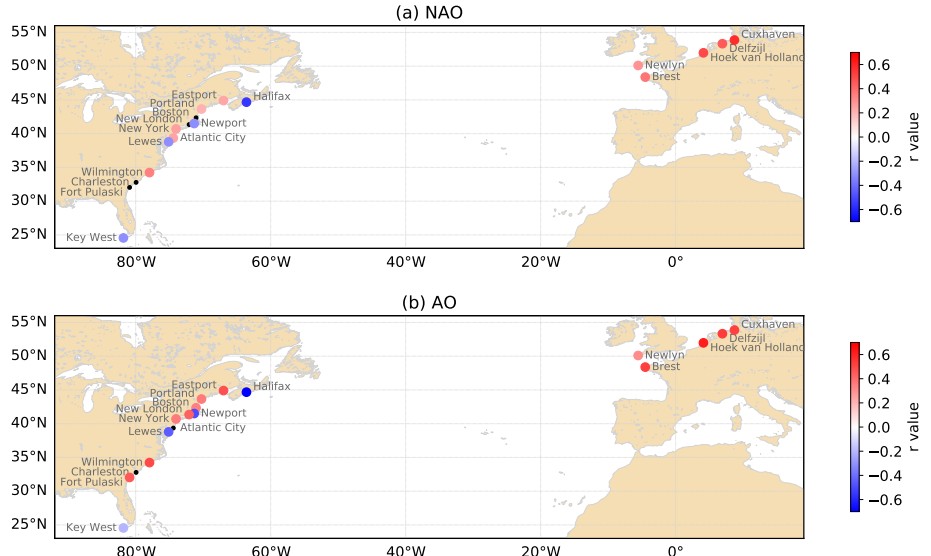

**Figure 7.** Correlation (r-value) since 1910 between $M_2$ and (a) North Atlantic Oscillation and (b) Arctic Oscillation. Black dots are stations with no significant correlation. $M_2$, NAO and AO are filtered using the same time window (9 years).

fitted $M_2$ variations with a MSL linear regression model (model 1), and a MSL and NAO multiple linear regression model 305 (model 2). Models 1 and 2 may be expressed as:

$$Model\,1 = \alpha_1 MSL \qquad (2)$$

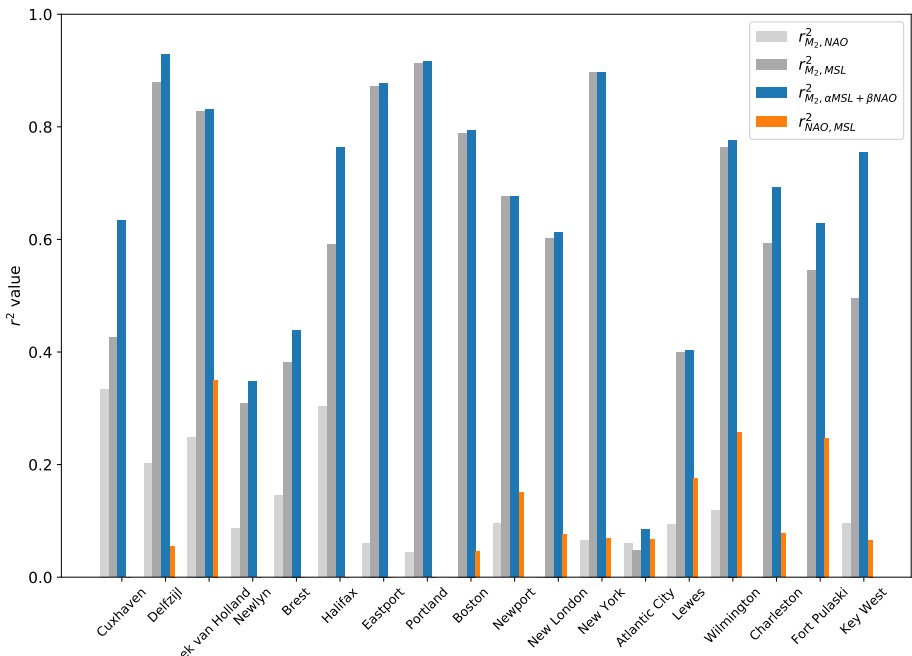

**Figure 8.** Variance explained ($r^2$-value) since 1910 between $M_2$ and NAO, $M_2$ and MSL, $M_2$ and fitted model $\alpha MSL + \beta NAO$ (model 2), NAO and MSL. $M_2$, NAO and MSL are filtered using the same time window (9 years). Note that there is no orange bar for NAO-MSL when the correlation is not significant ($p > 0.05$).

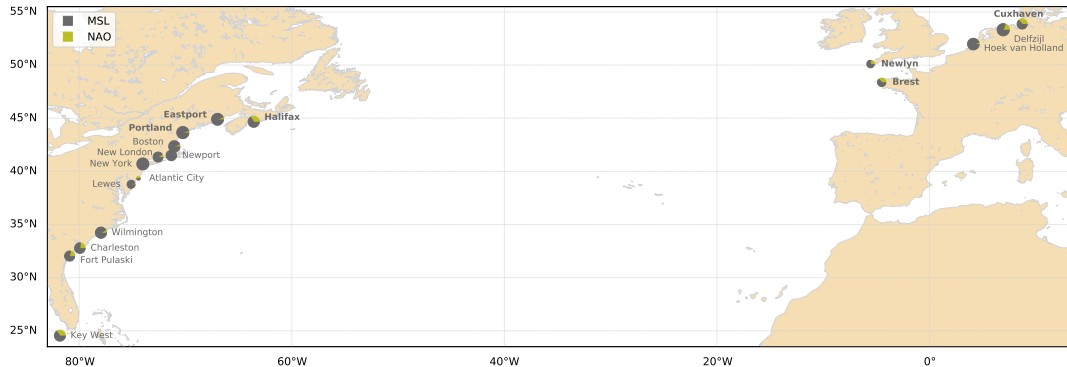

**Figure 9.** Relative contribution of $\alpha$ compared to $\beta$ in the fitted model $\alpha MSL + \beta NAO$. Black dots are stations with no significant $M_2$-NAO correlation. The size of each large dot is proportional to the correlation between $M_2$ and the fitted model. Stations with no MSL-NAO correlations are labelled in bold.

$$Model\ 2 = \alpha MSL + \beta NAO \tag{3}$$

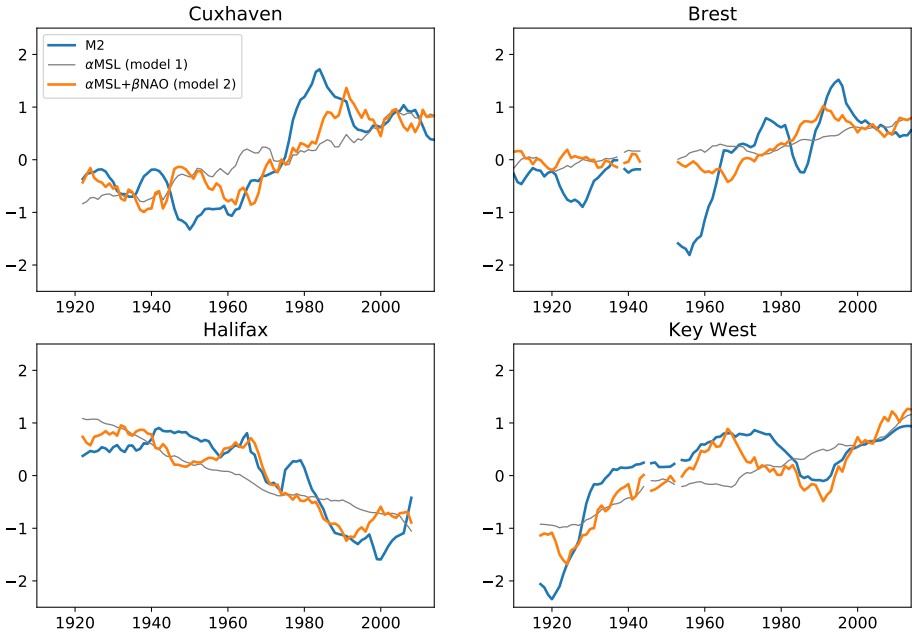

**Figure 10.** Variations since 1910 of $M_2$, $\alpha$MSL (model 1), $\alpha$MSL+ $\beta$NAO (model 2). $M_2$, NAO and MSL are filtered using the same time window (9 years).

The correlations between $M_2$ and model 1 (MSL) and model 2 (NAO and MSL) are presented in Figure 8. We checked if there was correlation between NAO and MSL at the stations (there is no correlation at 6 stations, and r-value is between 0.2 and 0.6 at 8 stations, see Figure 8 and discussion below). The results are the following: (1) $M_2$ varies at first order with MSL (Figure 8). (2) The introduction of the NAO (model 2) allows to increase the predictive performance of the model, beyond the inherent effect of adding an additional regression parameter. Indeed, on average, the Akaike Information Criterion (AIC) is 99.9 for model 2, instead of 112.7 for model 1. On average, the $r^2$-value is 0.67 for model 2 instead of 0.61 for model 1. At some stations, the increase is quite large. For example at Cuxhaven, the $r^2$-value jumps from 0.42 to 0.64 between model 1 and 2. (3) The ratio $\frac{\beta}{\alpha+\beta}$ represents roughly the relative contribution of the NAO compared to the total effect of MSL and NAO (Figure 9), as MSL and NAO are normalized. We found a significant contribution at some stations (e.g. more than 30% at Cuxhaven and Halifax), whereas it is negligible at others (e.g. only 5% at Portland). 8 stations out of 18 show large NAO contribution (> 20%). The North East Atlantic seems to be more sensitive to the NAO. Note that the interpretation of the results is tricky when MSL-NAO correlation is significant (orange bars in Figure 8). For example, at Hoek van Holland, the relative NAO contribution is very small, mainly because MSL and NAO are highly correlated ($r = 0.59$). Figure 10 shows $M_2$ variations along with the predictions from the two models, at all four stations where the NAO contribution is significant ($\frac{\beta}{\alpha+\beta} > 0.25$) and the correlation between $M_2$ and model 2 is large enough ($r > 0.3$). At Cuxhaven, Halifax and Key West, the model 2 (MSL and NAO dependent) better captures the $M_2$ variations than the model 1 (MSL dependent); at Brest, the improvement is less significant. The trend-switch observed since the 1990 in the North East Atlantic could be partly explained

by the influence of the NAO on the tide.

These results suggest that a NAO-related mechanism may explain part of the variability of $M_2$. The underlying mechanism could be due to the difference of spatial distribution of water level, depending on the NAO index. Figure 11 (a) shows the average sea-level pressure during the period 1850-2015, derived from the Twentieth Century Reanalysis (20CR) (Compo et al., 2011; Slivinski et al., 2019). A positive NAO winter (e.g. 1989) corresponds to a situation with a stronger pressure gradient than average, between the two pressure systems of Azores and Iceland (Figure 11 (c)). By contrast, a negative NAO winter (e.g. 1969) corresponds to a weaker gradient pressure than usual (Figure 11 (b)). This way, from one year to another, the large-scale atmospheric masses are distributed differently, and as a consequence, the water volumes are also distributed differently in the Northern Atlantic. In a situation of NAO$^+$, the surface waters are pushed onshore, moving from Iceland to the European coasts of France, Spain and Portugal. Figure 12 (a) shows the redistribution of the sea-level pressure, between two years with high and low NAO indices (here 1989 and 1969). Note that this is an extreme situation, as these years have strong positive and negative indices. The changes in terms of water level may vary from -15 cm to 24 cm, assuming an inverse barometer response of sea level. This variation of a few tens of cm is probably negligible offshore, but may have some impact on tide propagation along the continental shelves and in shallow waters. It could also shift slightly the amphidromic points. Assuming that these changes have a similar impact (in terms of magnitude) on $M_2$ as MSL changes, that is, $\pm$ 10% in shallow waters according to recent simulations (Pickering et al., 2017; Idier et al., 2017), we find that they can yield changes in $M_2$ amplitude up to a few centimeters. In other words, their order of magnitude is in agreement with the changes observed in $M_2$ (Table 1).

We conducted further investigations in the North Sea to test if the magnitude of sea-level pressure changes induced by large-scale atmospheric circulation (Figure 12 (a), a dozen of hPa) can generate the observed decadal-scale $M_2$ changes at Cuxhaven (Figure 12 (b), few cm). Note that $M_2$ changes due to large-scale atmospheric circulation are only a small part of the total observed changes (20 cm at Cuxhaven), as the changes are also due to MSL rise. The underlying mechanism invoked in the present paper (i.e. the influence of the atmospheric circulation on the tide) is close to the one described in Huess and Andersen (2001), except that we are considering a longer time scale (decadal instead of seasonal). Huess and Andersen (2001) explain partly $M_2$ seasonal variations through the effect of atmospheric circulation. They ran a barotropic model in the North Sea, forced (1) with tides only and (2) with both tides and meteorological fields. Their results show that the $M_2$ seasonal modulation is better captured when the model is forced with both tides and meteorological fields (their Figure 2, top right, amplitude higher than 10 cm in the German Bight) rather than with tides only (their Figure 2, top left, amplitude lower than 5 cm in the German Bight). It is important to underline that their model is barotropic, and that there is no effect of stratification, which may also play a role in $M_2$ changes (see 3.3.6 in the review of Haigh et al. (2019)). At seasonal scale, we computed monthly (instead of yearly) $M_2$ variations at Cuxhaven over 5 years (2010-2015), and we obtained results in agreement with Huess and Andersen (2001). That is, a seasonal cycle with a range of around 15 cm, maximum in summer and minimum in winter (Figure 12 (d)). According to Huess and Andersen (2001), this seasonal cycle is partly due to the atmosphere circulation. We then computed the differences of monthly sea-level pressure between January and July 2015 (Figure 12 (c)), and obtained values

close to the ones in Figure 12 (a) in terms of magnitude (a dozen of hPa). This shows that the order of magnitude of sea-level pressure changes between a NAO$^+$ and NAO$^-$ years (Figure 12 (a), a dozen of hPa) may lead to the $M_2$ observed changes at Cuxhaven (Figure 12 (b), a few cm, similar to its seasonal variations). The assumption that changes of a dozen of hPa in the North Sea may generate a sea level response of few centimeters is reasonable, but dedicated simulations should be conducted to confirm or discard the water volume redistribution hypothesis. Note that here, we followed the hypothesis mentioned in Huess and Andersen (2001), who consider that the atmospheric circulation may be partly responsible of $M_2$ seasonal variations in the North Sea. But there are other hypotheses; Müller et al. (2014) and Gräwe et al. (2014) rather consider that the stratification plays a major role in the North Sea. However, it is difficult to disentangle the respective contribution of each of these two processes in $M_2$ seasonal changes, only from the available observations.

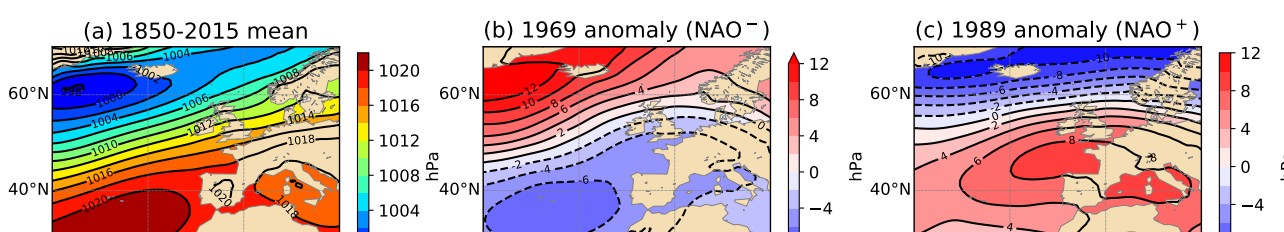

**Figure 11.** Winter sea-level pressure over the North East Atlantic (a) average over 1850-2015 (b) anomaly in 1969 (NAO$^-$) (c) anomaly in 1989 (NAO$^+$). Contour intervals are every 2 hPa.

## 5 Conclusions

We investigated the long-term changes of the principal tidal component $M_2$ over the North Atlantic coasts. We analysed 18 tide gauges with time series starting no later than 1940. The longest is Brest with 165 years of data. We carefully processed the data, particularly to remove the 18.6-year nodal modulation.

We found that $M_2$ variations were consistent at all the stations in the North East Atlantic (Cuxhaven, Delfzijl, Hoek van Holland, Newlyn, Brest), whereas variations appear between stations in the North West Atlantic. The changes started long before the XX$^{th}$ century, and are not linear. The trends vary significantly from one station to another; they are overall positive, up to 2.5 mm/yr, or slightly negative. Since 1990, in many stations, the trends switch from positive to negative values. The significant differences between the trends since 1910 and 1990 indicate caution when interpreting trends based on short records, i.e. less than 30 years, especially if the data are noisy (interannual variability) and the underlying processes non-linear (change points).

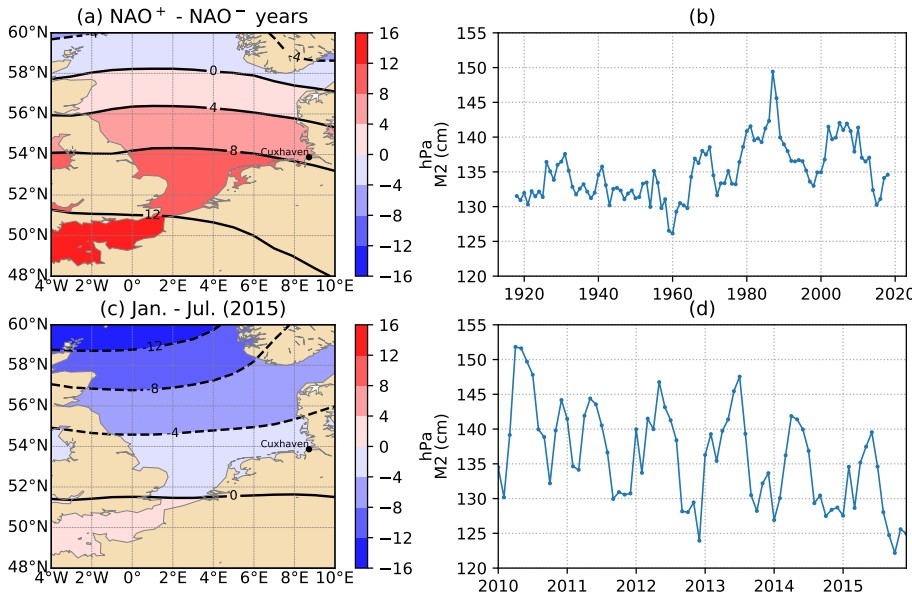

**Figure 12.** At decadal time scale: (a) Difference of winter sea-level pressure between 1989 (NAO$^+$) and 1969 (NAO$^-$) in the North Sea (b) Annual $M_2$ amplitude at Cuxhaven from 1918 to 2018. At seasonal time scale: (c) Difference of monthly sea-level pressure between January and July 2015 in the North Sea (d) Monthly $M_2$ amplitude at Cuxhaven from January 2010 to December 2015. Contour intervals are every 4 hPa in (a) and (c).

Concerning the causes of the observed changes, $M_2$ varies primarily with the MSL, but MSL rise is not sufficient to explain the variations alone. The similarity between the North Atlantic Oscillation and $M_2$ variations in the North East Atlantic suggests a possible influence of the large-scale atmospheric circulation on the tide. Our statistical analysis confirms large correlations at all the stations in the North East Atlantic. The trend-switch observed since 1990 could be the signature of the large-scale atmospheric circulation on the $M_2$ tide. The underlying mechanism would be a different spatial distribution of water level from one year to another, depending on the low-frequency sea-level pressure patterns, and impacting the propagation of the tide in the North Atlantic basin. In the future, dedicated modelling studies should be undertaken to confirm or discard this hypothesis. These simulations should also allow to estimate the effect of the wind (through the Ekman current) and currents on $M_2$ changes (Devlin et al., 2018).

In this study, we focused only on $M_2$ amplitude. A similar analysis on the phase lag would draw a more complete picture of the $M_2$ variations (Müller, 2011; Woodworth, 2010; Ray and Talke, 2019). Other constituents are also affected. Results show that $S_2$ amplitude decreases at all the stations located in the North West Atlantic, and in contrast, tend to increase in the North East Atlantic (not shown). The large-scale decrease of $S_2$ observed in the North West Atlantic is consistent with previous studies (e.g. Ray, 2006, in the Gulf of Maine). Further investigations should be definitely conducted to extend this

work to more constituents.

The historic data show that the changes started long before the $XX^{th}$ century. This conclusion would not have been possible without the huge work of data rescue undertaken over the past decades (e.g. Pouvreau et al., 2006; Pouvreau, 2008; Bradshaw et al., 2016). This underlines the great importance of sea level data archaeology, which allows to extend and improve historical datasets (Pouvreau, 2008; Woodworth et al., 2010; Marcos et al., 2011; Talke and Jay, 2013, 2017; Ray and Talke, 2019; Bradshaw et al., 2015, 2020; Haigh et al., 2019). This is essential for studies related to climate change.

Finally, we should mention several additional limitations and perspectives in this study. (1) We processed the time series considering that they were quality controlled. A fuller analysis of the data quality before processing would probably be valuable. (2) We did not investigate the history of each station. There are probably some local changes (e.g. environment or instrumentation) that may explain a part of the variability of $M_2$ amplitude, and some discrepancies between stations. (3) The tide gauges are located mainly in harbours. They are affected at the same time by local and regional/global scale changes, that are difficult to separate. Moreover, they may be not representative of changes offshore. A similar study based on satellite altimetry data would probably be of great interest, even if temporal scale for satellite data is still rather short (i.e. < 30 years) compared to climate-scale processes. (4) We focused mainly on the UHSLC dataset, which consists of 249 stations in the Atlantic Ocean. Other relevant stations (that are not in this dataset) may be considered in future studies. (5) We did not investigate the impact of storminess on the tide. Dedicated studies are necessary to estimate if changes in storminess could affect significantly tidal constituents. (6) We used only winter AO and NAO indices, which show more variability than annual indices. A similar analysis with annual indices shows similar results for the correlation with AO or NAO (positive correlation on the North East Atlantic). With annual rather than monthly indices, the difference of pressure fields will decrease, and as a consequence, the magnitude of the sea-level response will also decrease. Further investigations should be conducted on this point.

## Appendix A: Nodal modulation

The $M_2$ component is subject to a 18.6-year modulation, separated from a neighboring line in the tidal potential ($m_2$) whose Doodson number differs in its $5^{th}$ frequency (255 555 and 255 545 for $M_2$ and $m_2$, respectively) (Doodson and Warburg, 1941; Pugh and Woodworth, 2014). This $5^{th}$ frequency corresponds to $N'$, the negative of the mean longitude of the Moon ascending node - hence the "nodal" term - whose period is 18.6 years. Note that there is also another component close to $M_2$, whose Doodson number differs only from the $5^{th}$ frequency (255 565), but it is negligible, its amplitude in the tidal potential being only 0.05% of $M_2$, whereas $m_2$ amplitude is 3.7 % of $M_2$ (Simon, 2007, 2013). With one year of hourly data, the two components $M_2$ and $m_2$ cannot be separated by a yearly harmonic analysis (at least 18.6 years are necessary). As a

consequence, $M_2$ amplitude is modulated by $m_2$. However, we can estimate this modulation, and remove it. The harmonic formulation is expressed schematically as a sum of harmonic components

$$h(t) = \sum_i a_i cos(V_i(t) - \kappa_i) \tag{A1}$$

where $h(t)$ is the sea level height at time $t$, $V_i(t)$ is the astronomical argument (computed from Doodson number) and $a_i$, $\kappa_i$ the amplitude and phase lag of each component. Considering that $M_2$ and $m_2$ are very close in terms of frequency, we can assume that their phase lags are similar ($\kappa_{M2} \simeq \kappa_{m2}$). As their difference of astronomical arguments is $V_{m2} - V_{M2} = N' + \pi$, the $M_2$ and $m_2$ contributions to the total water level may be expressed as

$$h_{M2}(t) + h_{m2}(t) = h_{M2}(t)[1 + f_{nod}cos(N' + \pi)] \tag{A2}$$

where $f_{nod}$, the nodal modulation, is the ratio of the amplitude of $m_2$ and $M_2$. As $M_2$ and $m_2$ are very close in terms of frequency, $f_{nod}$ is generally considered as close to the ratio of their amplitude in the tidal potential, $A_{m2}$ and $A_{M2}$

$$f_{nod} = \frac{a_{m2}}{a_{M2}} \simeq \frac{A_{m2}}{A_{M2}} \simeq 0.037. \tag{A3}$$

The negative of the mean longitude of the Moon ascending node is expressed simply as a function of time (p . 116 in Simon (2007), p. 112 in Simon (2013))

$$N' = -N = 234.555 + 1934.1363T + 0.0021T^2 \tag{A4}$$

with $N'$ in degrees, and $T$ the time elapsed since 2000/01/01 at 12:00, expressed in Julian centuries (36 525 days).

The tidal program we used (MAS) corrected $M_2$ applying the usual 3.7% nodal modulation (Eq. (A3)). However, this value may vary significantly from one station to another; Ray (2006) reported values ranging from 2.3 % to 3.6 % in the Gulf of Maine. Here, we computed directly $f_{nod}$ from the observed data, proceeding as follows. (1) We added the default nodal correction $1 + 0.037cos(N' + \pi)$ to the $M_2$ variations. (2) We detrended the obtained signal removing the last Intrinsic Mode Function (IMF) of an Empirical Mode Decomposition (EMD) (Huang et al., 1998); note that the EMD is an analysis tool which partitions a series into 'modes' (i.e. IMFs), the last one being the trend of the signal. (3) We fitted a function $a_{m2} cos(N' + \pi)$ to this detrended signal to estimate $a_{m2}$, $N'$ being expressed as in Eq. (A4). (4) We finally computed $f_{nod}$ as the ratio between $m_2$ and $M_2$ amplitudes (Eq. (A3)). Figure A1 (a) shows an example of estimate of $M_2$ modulation at Newlyn: the fit leads to a nodal modulation of 3.3 %. Note that this value is consistent with Woodworth (2010) (3.2 %), whereas Woodworth et al. (1991) gave a slightly different value (2.8 %). Figure A1 (b) shows the impact of this value rather than the default one: oscillations of 18.6 years are clearly reduced. Note that in this study, the $m_2$ amplitude - and then the nodal correction - could have been computed from the full time series harmonic analysis, as records are longer than 18.6 years. However, the method presented

here to compute the nodal correction can be applied even for time series shorter than 18.6 years.

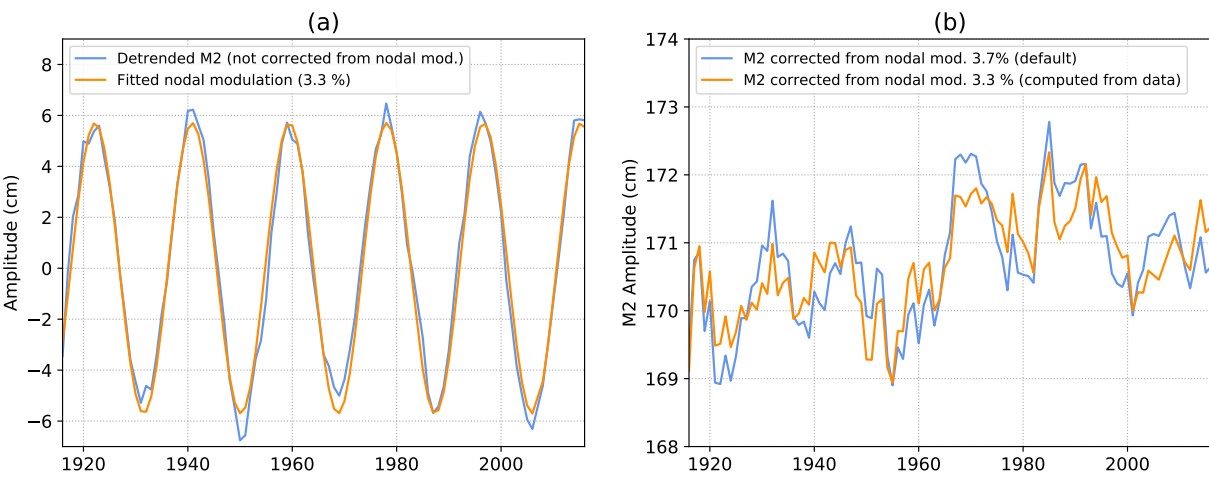

**Figure A1.** (a) Estimation of the nodal modulation of $M_2$ amplitude (mean removed) at Newlyn (b) Impact on $M_2$ amplitude of the nodal modulation correction at Newlyn. $M_2$ is detrended in (a) to better fit the nodal modulation.

The computed nodal modulations are summarised in Table 1 (column 6). They vary from 0.8 to 4.1 %. Note that these values are consistent with those obtained by previous authors (Ray, 2006; Müller, 2011; Woodworth, 2010; Ray and Talke, 2019). Only the value at Charleston differs significantly: 3.0 % in our study compared to 3.7% in Müller (2011).

## Appendix B:  Time series of annual $M_2$ amplitude at all the stations

*Author contributions.*  LPG analysed the data and wrote the paper. PL and GW contributed to the interpretation of the data and the writing of the paper.

*Competing interests.*  The authors declare no competing interests.

*Acknowledgements.*  This work was supported by the Research Theme "Long-term observing systems for ocean knowledge" of the ISblue project "Interdisciplinary graduate school for the blue planet", co-funded by a grant from the French government under the program "Investissements d'Avenir" (ANR-17-EURE-0015). The sea level observations were provided by the University of Hawaii Sea Level Center - retrieved from ftp://ftp.soest.hawaii.edu/uhslc/rqds, accessed April 2020. The sea level data at Delfzijl and Hoek van Holland were provided

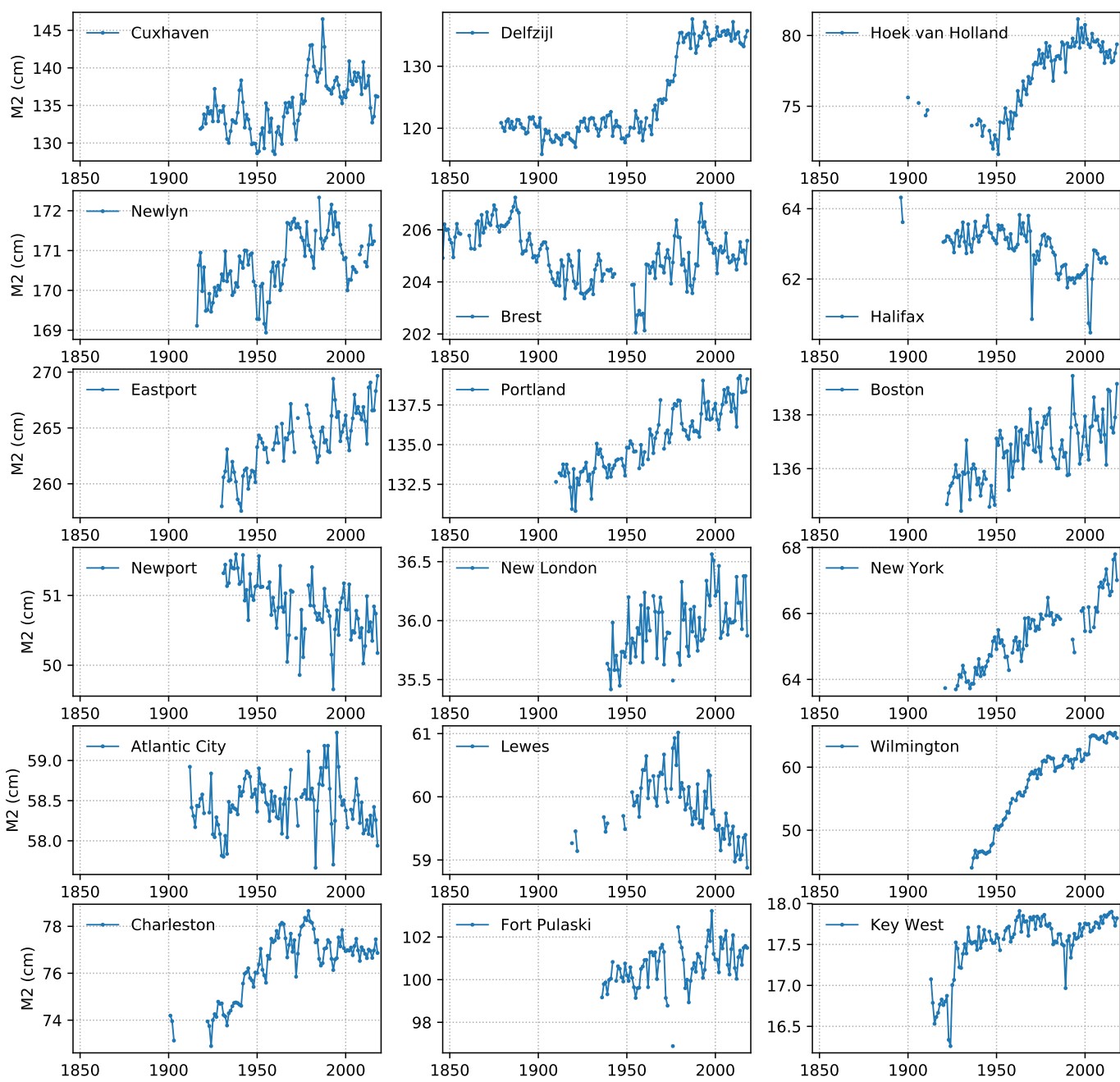

**Figure B1.** Annual $M_2$ amplitude at the 18 selected tide gauges

.

by Rijkswaterstaat (RWS) Service Desk, Netherlands. The climate indices (NAO and AO indices) were provided by the the Climate Analysis Section, NCAR, Boulder, USA - retrieved from https://climatedataguide.ucar.edu/climate-data/, accessed April 2020. The AMO index

was provided by NOAA Physical Sciences Laboratory - retrieved from https://psl.noaa.gov/data/timeseries/AMO/, accessed April 2020. The harmonic analysis program MAS was provided by the French Hydrographic Office (SHOM). Support for the Twentieth Century Reanalysis

Project version 3 dataset was provided by the U.S. Department of Energy, Office of Science Biological and Environmental Research (BER), by the National Oceanic and Atmospheric Administration Climate Program Office, and by the NOAA Physical Sciences Laboratory. The authors very warmly thank the two reviewers (Stefan Talke and an anonymous reviewer) and the Editor (Philip Woodworth) for their careful reading and their many constructive comments, which allowed to greatly improve the paper.

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
