# Peer review of "Large-scale changes of the semidiurnal tide along North Atlantic coasts from 1846 to 2018"

_Ocean Science, 2020_

## Short Comment (SC1) · 10 Jul 2020

Understanding the long-term changes of tides is important and useful. This paper investigated long-term tidal change in the North Atlantic and discussed a possible underlying mechanism which I think is pretty interesting. I want to provide some suggestions which may further improve this paper.

First, the title of this paper is 'Climate-scale changes of the semidiurnal tide…..', however, the authors only analyzed M2 tide but ignored S2 tide which is also important. In addition to, the gravitational forcing of S2, oscillations in barometric pressure, changes in ocean temperature, and onshore-offshore wind have also been argued as contributing to the sea surface variations at the S2 frequency(Feng et al., 2015). The non-gravitational component of S2 is called the radiational tide and its amplitude has been estimated to be 10–18% of the gravitational amplitude, depending on geographical region and the physical parameters concerned (e.g., pressure, wind stress, and/or thermal forcing) (Feng et al., 2015). It seems that S2 tide is more easily influenced by changes of atmospherical circulation than M2 tide. Thus, it is necessary to check whether the changes of S2 tide are similar to North Atlantic Oscillation (NAO) which will prove underlying mechanism proposed by this paper.

Second, this paper is very similar to Müller (2011) which found the rapid change in semi-diurnal tides in the North Atlantic since 1980. This paper seems to revisit the Müller's work and change 1980 to 1990. The authors need to clearly describe the difference of two papers. By the way, this paper calculates the post-1990 trend and post-1910 trend to show the rapid change in M2 tide since 1990. I think post-1990 trend is meaningless because the length of post-1990 records is too short. I think that you can calculate the trend of 1910-1990 and post-1910.

At last, although it seems that M2 variations are similar to NAO, it is very difficult to prove statistical validity since the data are too short. The authors should point out this in the paper.

Reference:
Feng, X., M. N. Tsimplis, and P. L. Woodworth (2015), Nodal variations and long-term changes in the main tides on the coasts of China, *J. Geophys. Res. Oceans*, 120, 1215–1232.
Müller, M. (2011), Rapid change in semi-diurnal tides in the North Atlantic since 1980, Geophys. Res. Lett., 38, L11602.

---

## Referee Comment (RC1) · Anonymous Referee #1 · 12 Jul 2020

In this paper, nine very long (>80-year) tide gauge records along North Atlantic coasts are analyzed for secular changes in the M2 amplitude. The series are compared both among each other and with climate mode indices in an attempt to relate the observed amplitude changes to large-scale forcing mechanisms. Unfortunately, the paper is of limited scope and the methods are not innovative. The arguments put forth to link the observed M2 changes to the North Atlantic Oscillations (NAO) are fallacious (see below) and invalidate exactly that part of the paper that is thought to break fresh ground compared to similar analyses in the past (e.g., Mueller 2011, GRL). These shortcomings are not easily redressed in a revision, and I therefore recommend the manuscript to be rejected. Overall, the study offers too little new insight. It merely highlights similarities without investigating the underlying processes.

[Figure]

Major issues:

[-] The authors' explanation of the pronounced M2 increase over 1960–1990 at the three European stations (∼3 cm at Brest and Newlyn, ∼10 cm at Cuxhaven) in terms of typical NAO sea-level pressure patterns is flawed. The response of sea level (or water column thickness) to changing pressure loading on time scales longer than a few weeks is static isostatic (IB), creating a sea-level difference of about 6–10 cm at the location of the tide gauges, according to the authors' plots (Figures 8 and 9). These changes in water depth are simply too small to cause the observed M2 amplitude trends. Typically, sea-level changes alter the propagation characteristics of tides in shallow water such that one can expect perturbations in the M2 amplitude of 1–5% relative to the imposed water depth change, see the modeling results by Schindelegger et al. (2018). Cuxhaven may be an exception of that rule, although a 6-cm increase in sea level from atmospheric pressure will engender less than 2 cm changes in the M2 amplitude (Figure 8 of Schindelegger et al.).

The authors circumvent the problem by assuming higher sea-level changes in areas distant to the gauges (-20 cm near Island) and picking a 10% sensitivity of M2 amplitudes to water depths from literature – an inordinate value that only holds in very shallow settings (e.g., estuaries) and not across entire shelf regions.

Moreover, Figure 8 displays higher atmospheric pressures around the European mainland in 1989, implying that sea level actually dropped relative to 1969, opposite to what is shown in Figure 9. So that's an inconsistency on its own, but more alarmingly for the authors' theory, numerical modeling (see references) strongly suggests that an increase of M2 in the German Bight, as observed at Cuxhaven, actually requires local sea level to rise, not to fall.

[-] A key argument is that the low-frequency winter NAO index (Figure 7) is similar to the evolution of the M2 amplitude at the three European stations. Such an important point in the paper should be substantiated by an appropriate plot (in which annual M2

changes would be filtered using the same 9-year running median as the NAO index). More importantly, the mentioned similarity is never established in a quantitative sense, e.g., by tabulating correlation measures and their statistical significance considering effective degrees of freedom. In fact, the Brest time series in Figure 3a seems to have rather little in common with the NAO time series as it has a dip in the 1980s (when NAO steadily increases) and features an all-time high in the late 19th century (when NAO just erratically switches sign). For Newlyn and Cuxhaven, I expect the correlations to be higher, although the timing of individual peaks might be different. Such phase lags and leads are not easily explained in terms of physics; certainly not within the framework proposed here, because both sea level and subsequently tides would adjust instantaneously to NAO-related atmospheric pressure loading.

[-] The data basis on the European Shelf (three stations) is very shaky. It would be desirable to make the analysis more robust by adding results from tide gauges that are somewhat shorter but still provide good coverage of the period with distinct variability in NAO (1960 onwards).

[-] The Introduction leaves a lot to be desired. It is incoherent, lacks any quantification as to the size of observed tidal changes and does not tell the reader why he/she should bother. A very good example of clarifying the relevance of this subject matter up front is Mawdsley et al. (2015, https://doi.org/10.1002/2014EF000282).

Minor comments (most of these issues are indications of the authors' unsteadiness regarding the physics of tides):

[-] The Introduction's first sentence is wrong. Tides have been changing also prior to 19th century, e.g., due to Earth's continental and glaciation cycle.

[-] The main tidal constituents in the North Sea are presently not in a state of resonance (as argued on lines 172 and 219). They are rather described as Kelvin waves, dampened as they propagate from the Northwest through the basin in cyclonic fashion.

[-] Lines 250–252: First, stratification will not only change in response to heat fluxes, but also due to the advection of water masses, evaporation, salt dilution, etc. Second, in a discussion of stratification effects on "tides", one must use very precise language, in particular distinguish between barotropic, baroclinic, and surface (barotropic + baroclinic) tides. Third, the process identified by Kang et al. (2002) as cause for tidal seasonality in the Yellow/East China Sea is mixing strength (changes of vertical eddy viscosity) and not barotropic-to-baroclinic energy conversion.

[-] Lines 231–234: The simulations of Pickering et al. (2017) show exactly the opposite of what is described here (that is, their Figure 1a highlights an M2 increase in the German Bight, not a decrease).

[-] I understand the pragmatic approach of normalizing M2 changes to show results from different stations in one plot, but it would still make sense to include some absolute numbers (e.g., by using text or secondary Y axes) to facilitate quantitative comparisons among stations and allow for a meaningful interpretation of results derived later on.

[-] Annual tidal harmonics are computed from data spanning full years, but the discussion of atmospheric pressure changes only focuses on snapshots from winter months – another inconsistency in the analysis. Surely, if annual averages of atmospheric pressure fields are considered, the magnitude of the static sea-level response would decrease even further.

---

## Referee Comment (RC2) · Stefan Talke (Referee) · 14 Jul 2020

Summary: The manuscript "Climate-scale changes of the semidiurnal tide over the North Atlantic coasts from 1846 to 2018", by Pineau-Guillou et al., evaluates 9 long tide data sets on both sides of the Atlantic to investigate whether there is evidence for basin-scale perturbations to tides caused by climate variations or climate change. It is found that M2 for 3 gauges in the northeastern Atlantic follow a similar pattern as the decadally filtered NAO index. No clear correlation is found in the western Atlantic. It is noted, however, that rates of M2 change go negative at many of the 9 stations after around 1990.

Evaluation: It is an interesting idea to try to discern whether there are coherent, basin-

scale variations in M2 in the northern Atlantic, and one that has proved challenging to find in the past (to my knowledge). It would be quite an interesting result if M2 patterns caused by climate variability only show up in long data sets. Therefore, if properly done, it would be interesting to convincingly prove (or disprove) the hypothesis that (for example) long-term NAO patterns affect tides coherently. However, as presently conceived and presented, am not convinced that the authors have really shown that this is occurring (or disproven it). Issues include:

1. There is no attempt at a statistical correlation between the tide records and climate records such as the NAO index. While the tidal records in Europe superficially follow a similar trend as the filtered NAO, it is quite possible for data sets with few degrees of freedom (here, a decadal median) to resemble each other by random chance. For this reason, it is important to do some sort of significance testing and report statistics such as R2 and the p-value. Similarly, would be good to verify that the 6 US East coast records show a statistically insignificant correlation.

2. Overall, it would seem to me that the data selection is incomplete. On the European side, there are some Dutch coastal records that predate 1920, such as Hoek van Holland and Delfzijl (maybe also others; perhaps check if the GESLA data set has them). Similarly, there are additional records on the US East Coast that could be used. These include Fernandina (1897-present; available from NOAA) and Sandy Hook (Available since ∼1910 from NOAA) (Note that Baltimore (1902-present) and Philadelphia (1901-present) also exist, but are not coastal stations). Moreover, there are a number of M2 estimates extending into the 1800s in New York Harbor (Talke et al. 2014 supplement and Chant et al., 2018), Boston (Talke et al., 2018), Eastport (Ray and Talke, 2019), and Long Island Sound (Kemp et al., 2017). Some additional estimates of tidal range at coastal stations deep into the 1800s are found in Talke & Jay (2020). These can be used as a proxy for M2 if divided by ∼2.

3. Further, it's not clear how significant the data cut-off of 1920 is. If 1930 or 1940 were used, for example, how might results or conclusions change? A later cutoff would

enable inclusion of a number of additional US East Coast stations (e.g., Mayport, Fort Pulaski, Sewells Point, Willets Point, Providence, Eastport, etc). Perhaps the same is true for Europe. How much would conclusions or patterns change if a slightly less restrictive date-cutoff were used? It would be good to use a statistics-driven reason for the date cutoff (e.g., degrees of freedom in a correlation analysis, or something like that), and to check the effect of relaxing the cutoff. Regardless, my guess, based off of Figure 2 of Talke & Jay 2020, is that patterns of M2 change might be even less coherent if more data are used. This is in part because there are so many local processes that can affect tide gauges, even those at or near the outlet of an estuary (for example, Mayport, Lewes, Charleston, Fernandina, Sewells Point, Boston, and Sandy Hook in the US; Cuxhaven in Germany; and the Saint John gauge in Canada are all "coastal" gauges that are actually at the mouth of or within an estuary). On the other hand, there is an interesting and not completely explained decrease in tidal range and M2 at Sandy Hook, The Battery, and Boston from the mid and early 1800s until the 1920s, and then an increase. In other words, it's possible there is a larger Northwest Atlantic Signal, in addition to local processes. However, while models such as Schindelegger et al. 2018 are able to see coherent trends, it has been challenging to see it in the data (as mentioned in this manuscript as well, also in Ray 2006).

4. A similar data comment holds for the trend-switch which is observed around 1990. It would be quite interesting if this is a coherent signal throughout the Atlantic. To prove, would suggest that there are many more gauges that are available that could be used to test this hypothesis. Again, a trend switch in 6 out of 9 gauges need not be statistically significant or could be argued to be local in nature. However, would be more significant if you had 30 or 40 gauges and found a similar percentage shift between the 1950 to 1990 trend, vs. 1990 to the present, That would be quite interesting. Also, the effect of moving the date (1985 or 1995 instead of 1990) might be worth investigating. If results depend on start date, the interpretation becomes less clear (and vice versa).

5. The paper would also be improved by digging more deeply into mechanisms. Some

discussion of how the NAO affects sea-level (and therefore, perhaps M2) is made, but it is quite qualitative. There are, however, a number of process-based studies that look into tidal changes at local, regional, and oceanic scale (see the Haigh et al. 2020 and Talke &Jay, 2020 reviews for references). The Atlantic is known to be near resonance, and there could be coupling between the shelf and the deep ocean (Arbic and Garret papers). Tide changes in the Gulf of Maine (and for that matter, Long Island Sound) can in theory radiate out to the larger ocean (e.g., Godin 1993). Also, it is known that the M2 amphidrome in confined, shallow seas moves with sea-level changes (see the references in the Haigh et al. review, or the Lee et al. 2017 and Ross et al. 2017 papers on the Chesapeake). Based on this, also perhaps on basin scale modeling (e.g., Schindelegger et al. 2018), what sorts of coherent patterns might be expected in the Atlantic based on historical sea-level rise? How much might this be spatially variable based on sea-level change caused by long-term NAO patterns? What might be the magnitude of the signal? The Schindelegger et al. 2018 paper shows that coherent changes across the basin are possible with sea-level rise, but are relatively small, for a given increment of sea-level change. Also, they show that some locations in the Atlantic are anti-correlated. How much of a sea-level change is needed, roughly, before a coherent longterm signal is findable in tide gauge data (given noise in data, etc)? One needs to know whether it's even possible (by the mechanisms listed) to obtain a secular coherence in basin-scale variability.

In other words, a more clear hypothesis of what a basin scale shift in M2 tides might look like and whether it is detectable (given current understanding of processes) might help with the interpretation. What sort of excursion in M2 would you expect the NAO to cause, based on how it affects sea-level? Setting up a hypothesis with specific criteria that can be proven or disproven might help. It's ok, in my opinion, to have a paper with a non-detect result (a possible outcome here). However, since the set of non-detect papers is infinitely large, the added value could come from adding scientific or statistical insight into the problem. A great example of this is the Haigh et al. 2014 paper which showed that one would need to wait a couple decades before being able to analyze

recent sea-level acceleration at an acceptable level of confidence. Is something similar true here? My qualitative guess is that it might be hard to see a basin scale coherence in data, but that regional scale effects that are driven by a similar process like sea-level fluctuations can perhaps be detected (see for example the Devlin et al. papers). Basically, the paper would be improved by more specifically investigating what sort of change is needed (and what sort of data quality/signal to noise is needed) before it might become possible to discern coherent climate effects on tides across the entire basin.

Detailed comments:

Line 14: Would also cite the review of Talke & Jay, 2020, since the historical changes in tidal range shown therein are relevant to this paper.

Line 19: "Long-term changes in tidal constituents are rather small" Would modify this to specify "at coastal stations". As shown in Talke & Jay 2020 (and refs therein), the secular change at many estuary and tidal river gauge stations is huge.

Line 20 "still poorly understood"– Not sure I would say this. Some of the mysteries are being solved (see the review papers), while some issues remain. Maybe rephrase?

Line 46-48: Check grammar; grammar of list Is not quite right.

Line 60-63: There are some M2 results for 19th century US stations that you could/should use. See for example the supplement of Talke et al., 2014 or Chant et al., 2018 for New York and Sandy Hook. See Ray & Talke (2019) for Eastport and Portland. See also Talke et al., 2018 for Boston. Finally, there are multiple tidal ranges shown in the Talke & Jay 2020 review paper. These can be divided by two to get an estimate of M2 over time.

Also, Sandy Hook data from around 1910 is available at the NOAA site. The datum is wrong, but that shouldn't matter for tidal analysis.

Table 1: Since you are using Cuxhaven, why not also use some of the Dutch stations?

In the records I have, Hoek van Holland starts in 1900, and Delfzijl starts in 1876. Maybe there are earlier ones as well—see for example the Hollebrandse 2005 thesis. You could check if they are in the Gesla dataset and/or contact the Dutch. Data used to be available at waterbase.nl, but not sure that works anymore.

Table 1: Why is Fernandina (1897-present) not used?

Table 1: Unclear what the meaning of mean sea-level is. What is the datum? Why not include the trend, rather than an absolute measurement (which is not necessarily meaningful).

Line 69: "This constraint resulted in excluding between 1 and 9 years". Unclear what you mean. You mean for each station?

Line 70 seasonal variation: where? Again, non-coastal stations will see more variability. Also, in the North Sea the change is higher (e.g., Graewe et al. 2014). So, maybe be specific and mention the Atlantic.

Line 74 to 91: This would seem to be pretty standard nodal correction theory. Unless you can explain what is unique about your approach, would suggest greatly condensing this and simply citing an older study that discusses this in more detail

Line 98-110: What is the rational for using the more complex method here? As you later state, it doesn't lead to significantly different results. I guess it's interesting that it can work for small time series. Do you see any evidence of changes to nodal cycle? There are a few papers on this recently. However, am not convinced that there is a physical reason for these observations, vs. just statistical noise. Could be something to look into, though—if there is a coherence between nodal cycle variations in the western and eastern Atlantic, would be worth commenting on. Otherwise, not sure that you need the complex approach to nodal cycle characterization.

Line 117, Equation 5: Can you explore/motivate the use of the standard deviation a bit more? A potential issue is that sigma may also reflect errors in the gauge data

(e.g., timing errors, etc). Some exploration would be good as to whether this is a factor. For example, does sigma change as a function of time? If there is a decrease in sigma around 1990 or 1995—when new digital gauges started being used, at least in the US—then it might indicate that instrumental issues are potentially affecting your results. See for example Zaron & Jay, 2014, who concluded that some constituent trends in the Pacific are spurious. Using some sort of method to validate the causes of sigma would therefore be good. How can we be sure that a few years of non-optimal data are not biasing sigma? The method of Zaron & Jay, 2014 could be used, or the method used by Talke et al. 2018 to assess timing errors could be used (see their supplement).

Figure 2: This figure may have some educational/explanatory value, but it's not really a new result. One could consider removing.

Line 128 maybe remove "are essential, as they"? Doesn't' really add much to sentence

Line 156 "is no linear trends" should be "is no linear trend" or "are no linear trends" Line 158 "curve is flattening" should be "curve flattens"

Line 159 Remove "yet" in "yet noticed"

Line 163-164 Not sure that the lack of an astronomical explanation automatically implies a solid earth-ocean-atmosphere coupling system cause. Am not even sure what is meant by that. A few more logical steps are needed before a reader can believe that

Figure 3: Charleston is a harbor city with a channel that has probably been subjected to dredging, though I haven't looked into it extensively. Can you discuss how/whether this impacts results?

Line 165: The Delfzijl station starts in 1876, so would be worth comparing to Cuxhaven. It's probably somewhat impacted by long term changes to the Ems estuary tides. Then again, Cuxhaven is probably influenced by the large change to Elbe tides. See for example Winterwerp et al. 2013. In general, there are quite a few papers out of

Germany (e.g., Jensen et al. 2003, 2005 conference papers, and maybe Mudersbach et al. 2013 (?) that discuss a big increase in tidal range from about 1960 to the 1990s on the German coast. More recently, I've been told this has slowed or reversed (though I'm not sure there is a paper on that yet). Another good reference is the Hollebrandse 2005 Master's thesis on Dutch gauges.

Line 166 The Talke & Jay (2013, 2017) paper and report are good references for sea-level/tide data archaeology, as are Peauvreau 2008 and some of the papers by Marta Marcos.

Line 172 Again, note the Winterwerp et al. 2013 paper that includes the Elbe. There are probably some German references too. The Talke & Jay 2020 and Haigh et al. 2020 reviews discuss tidal resonance (see also references therein).

Line 182 The Ray (2006) and Ray & Talke (2019) papers discuss change in M2 trend in the 1980s in the Gulf of Maineâ̆Ťmaybe reference.

Line 186 There are many other papers that have explored Gulf of Maine resonance besides Ray & Talke. That is not perhaps the best example. See e.g. the discussion and references in the Talke & Jay 2020 or Haigh et al. 2020 review, in addition to the Garret and Godin reference.

General comment: The Godin 1993 reference, and also for that matter the Arbic and Garret and Arbic et al.papers, are interesting because they discuss how resonance on a small scale (Gulf of Maine, Continental Shelf) can affect the larger Atlantic. See also the Platzman papers on resonance from the 1970s. All this could/should be discussed and investigated, since it gets at the idea that there might be a mechanism through which western and eastern Atlantic tides could be coupled. Is there reason to believe there might be? In a sense, this is an implicit hypothesis that is being investigated here, through correlation with climate indices. However, it would be helpful to motivate and explore physical mechanisms as well. Further, it might be helpful to explicitly pose a hypothesis in the introduction, such as "is there any evidence for correlated/coupled

changes in tides that might provide evidence for cross-Atlantic connectivity"?

Line 190-194: Why not use the Eastport data points from Ray & Talke, or at least discuss? The composite Pulpit Harbor/Bar Harbor data set might also be worth discussion. Boston is a possibility, too, though it is influenced by local processes as well (see the Talke et al. 2018 paper...).

Line 198 : It would be good to compare Atlantic City to Sandy Hook and The Battery (see Talke et al, 2014 and Chant et al., 2018). In fact, the case of Sandy Hook and The Battery/Governors Island are interesting, since there is a marked decrease from the 1860s until the 1920s or 1930s, and then an increase. Chant et al. (2018) show an even bigger change in nearby Newark Bay, though the 19th century data there are based on very short time series. In any case, the results are sort of consistent with the results at Brest. Dredging may have at least somewhat caused the 20th century amplification (see Ralston et al., 2019), and work at the channel mouth may have cause the early 20th century changes (Marmer, 1935).

Also, Boston showed a similar, large decrease in tidal range through the 1920s, then an increase. While this is likely in large part local, Talke et al. 2018 did note that it's similar to the pattern observed at Sandy Hook.

Line 218-219: Would also look into/discuss amphidrome changes. See the Haigh et al. review and references therein.

Line 218-223: These are very short paragraphs and not that well developed. Some more thought would be good. For example, "The trends have to be interpreted carefully" is perhaps an obvious statement (hopefully there is not a case when it is ok to interpret trends haphazardly...).

Line 226—Somewhat misleading statement. Ray & Talke (2019) are referencing other results when they state that MSL rise only partly explains trends. Furthermore, they only focused on Gulf of Maine. Would instead look into some of the studies that

have more carefully looked at SLR effects, such as Schindelegger et al. 2018 or Greenberg et al. 2012.

Line 228 "than mean sea" should be "as mean sea"

Line 230-235: The Pickering papers are for large sea-level rise scenarios, but don't retrospectively look at 20th century rise (if memory serves). Hence, is it a fair comparison? There are probably some papers or reports that discuss reasons for North Sea changes more thoroughly—please look into and review.

Figure 6: One could include the Portland sea-level data point from Talke & Ray 2019. If you include the Battery, then a longer data set is possible. Not sure however if this graph is needed or is critical for the story. It is not really a result of this study, just a replotting of other results. There is no clear analysis of how tides might be influenced by SLR—it's basically just a literature review.

Line 247-250. Wouldn't storminess also impact tidal constituents, at least on the shelf or in a harbor? I think there are some references on that. I came across a Pugh reference at some point for the Irish Sea, if memory serves. The Graewe et al. 2014 reference also discuss this for the North Sea, I think. In any case, wind stress and wave breaking and these sort of things represent an input of turbulent kinetic energy and could in theory affect tides at some stations, if there are climate-based shifts in storminess. In the context of this paper, Talke et al. 2014 showed that the probability of large storm tides in New York goes up when the NAO is negative. There are also known NAO effects in Europe (see the Woodworth et al. (2007?) paper). Does this matter for tides? Might be something to at least investigate.

By the way, it's not clear to me that a measured decrease in M2 during periods of stormy weather is a real change in M2. Another (perhaps not mutually exclusive) explanation is that depth changes during storms alter the phase speed of the tides, such that they arrive a bit earlier than usual (See for example Horburgh and Wilson 2007). A period with a lot of ups and downs in mean sea-level is going to cause lots of phase

speed variations, more spectral spread (cusping), but decreased amplitude. Just as timing errors can cause a decrease in measured M2 (see Zaron & Jay 2014), so would changes in phase speed.

Line 254 "possible role of stratification" for what? Would clarify, e.g., something like "possible role of stratification on secular tidal trends"

Line 255 "between these processes"—what processes? Maybe be specific.

Line 266 How can the NAO decrease globally? It is specific to the North Atlantic.

Line 278 "Pushed southern" should be "pushed southerly".

Line 289 Might be good to discuss the role of wind earlier. See notes above.

Line 291-292 Maybe, but there is quite a bit of variance in all the plots and it seems like a couple curves looking similar could easily happen from random chance. Unless you can figure out the statistical robustness of these results, would perhaps avoid ascribing M2 behavior at a few locations to NAO.

Line 332 The Devlin et al. papers discuss correlations between sea-level anomalies and tidal anomalies, and possible reasons for them. In a way, you are trying to do something similar, but over a larger time scale. However, there is little statistical correlation or significance testing done here. Would suggest this be done.

Please also note the supplement to this comment:
https://os.copernicus.org/preprints/os-2020-56/os-2020-56-RC2-supplement.pdf
* * *

---

## Editor Comment (EC1) · Philip Woodworth (Editor) · 14 Jul 2020

July 2020

I have some detailed, most technical comments on this draft with an editor hat on. The two formal reviews should be inspected for more on the science. There are rather a lot of them but many are trivial.

Philip Woodworth

Title - what does 'climate-scale' mean? At line 9 you refer to 'large-scale' which seems a more sensible description. Or 'basin-scale' maybe.

- The trends in M2 amplitude

- from one station to another

- 0.7 mm/yr in the period since 1910

- distribution of water level

- Tides have been changing .. factors since the XIXth century ..

- large-scale (cf. line 9)

- scale –> scales

- ditto

43-44 - I would drop this sentence

- thus only an accounted for change in ...

- years of data

- On the east side of the North Atlantic

- due to too small an M2

- On the west side of the North Atlantticc

- due to too small a tidal

- in Figure

- synthesised –> summarised

- in 1846 and 1896 respectively

- performed in order to compute

- similar to the

- I would drop 'largely .. community'. It may be true but its wide use is not relevant.

- of the yearly

- M2 correctly.

- seasonal variation of typically a few ..

A better reference for this would be:

Pugh, D. T. and Vassie, J. M. 1976. Tide and surge propagation off-shore in the Dowsing region of the North Sea. Deutsche Hydrographische Zeitschrift, 29, 163–213, doi:10.1007/BF02226659.

- instead of the 50% here

Table 1 caption line 1 - tide gauge records selected

Table 1 caption line 3 - .... modulation, estimated trends in M2 amplitude since

Table 1 - I don't see why you have column 5 (MSL average) which has no importance to this study, MSL being measured relative to an arbitray datum at each site.

- lead to the exclusion of more years.

This is obvious isn't it? So how did your results change with 75%?

- retrieved –> removed (Simon, 2007,2013) as described briefly below.

Drop 'Here .. method'

- reword:

.. an 18.6 modulation, separated from a neighbouring line in the tidal potential (m2) whose Doodson number differs in its 5th frequency ... respectively)) (cf. Doodson and Warburg, 1941; Pugh and Woodworth, 2014). This .., the negative of the ..

- but it is negligible, its amplitude in the tidal potential being ..

- .. and m2 cannot be separated by a yearly harmonic ..

- expressed schematically

- are the amplitude and phase lag [not phase shift]

- shift –> lags astronomic –> astronomical is given by

- The negative of the mean ... is expressed simply

- from one station .. We added the default ..

- to this detrended

- drop the comma

- please replace the hyphen with a colon. A hyphen looks like a minus sign.

Fig 2 caption. This should better say:

(a) Estimation of the modulation of M2 amplitude (mean removed) at Newlyn, (b) Impact of M2 amplitude ...

- ... (NAM) (Hurrell reference). These climate ...

- stations) over long periods

- what does 'Variations in the NAO are essential' mean? You mean important?

- The normalization involves ..

drop 'long-term'

- what do you mean by 'with yearly values' when you have said you are using wintertime values? I would drop these 3 words

This section should mention the AMO also as you use it below.

- the eastern

- consistent with the temporal coverage of the tide gauge measurements.

- Brest and Newlyn

- drop the brackets

- .. changes must be at least ..

- flattened yet –> already

- of the tidal

Fig 3 - nice plot

Fig 3 caption line 3 - The blue star in

- allow to confirm at larger timescale (?)

- in Figure

- into two groups

- ditto

- drop 'globally' (twice). They are not global which means 'worldwide' to most people

- ditto increases overall

- decrease, and since 1990 only

- one station

- which provides some confirmation of the hypothesis

- from the Brest

- drop globally decreases overall

- synthesised –> summarised

- one station drop globally positive overall

- found previously

- Lewes? You must mean Portland?

the latest –> recent and (2)

- in the tide

- mean sea level rise can result in an increase in M2 of .. of the MSL rise ... the same sign as mean ..

- define SONEL

- falling slightly

- give reference to GoM land movements fig 6 caption - remove (see Table 1 column 5) and remove that column - it has no importance.

- catch –> account for when they are forced with a meteorological field. What does this mean?

- affect

- in long-term

- and the Atlantic ..

The AMO is not referred to as an index in section 2.2.1. Also it is an SST index and not an air pressure one

Fig 7 caption line 1- you said before you used wintertime values not annual ones

- .. could be due to differences in the spatial heights –> level

- year –> winter

- usual distributed differently ... south

- height –> level

- volume

-preciptation

- on the scale

Fig 8 caption line 1 - .. pressure over the NE Atlantic

- coldest –> lowest sea surface temperatures were ..

Fig 9 line 1 - Changes in mean sea level due to the difference ... (NAO-) assuming an IB response of sea level.

- differences calls for –> indicates

- explain the variations alone.

- heights –> level analysis of the phase lag

- right! See above. How does that change the results?

- deep –> fuller

- tide gauges are obviously on the coast! Drop that. Harbours is relevant.

- the e20... refers to Haigh et al. This reference needs correcting.

Also you have some names with initials before the surname e.g. Trimble.

Finally, you might want to refer to Talke and Jay (Annu. Rev. Mar. Sci. 2020. 12:121–51) especially from the perspective of changing tides in estuaries such as Cuxhaven.

---

## Author Comment (AC1) · 14 Sep 2020

**Response to Referee Comment #2 (Stefan Talke)**

First of all, the authors would like to warmly thank Dr Stefan Talke for his careful reading of the paper and his many constructive comments. We tried to do our best to implement them, and they allowed to greatly improve the manuscript.

The reviewer comments are in bold, our replies are in normal font.

**Summary: The manuscript "Climate-scale changes of the semidiurnal tide over the North Atlantic coasts from 1846 to 2018", by Pineau-Guillou et al., evaluates 9 long tide data sets on both sides of the Atlantic to investigate whether there is evidence for basin-scale perturbations to tides caused by climate variations or climate change. It is found that M2 for 3 gauges in the northeastern Atlantic follow a similar pattern as the decadally filtered NAO index. No clear correlation is found in the western Atlantic. It is noted, however, that rates of M2 change go negative at many of the 9 stations after around 1990.**

**Evaluation: It is an interesting idea to try to discern whether there are coherent, basin-scale variations in M2 in the northern Atlantic, and one that has proved challenging to find in the past (to my knowledge). It would be quite an interesting result if M2 patterns caused by climate variability only show up in long data sets.**

We thank the reviewer for this comment. We agree that the idea of discerning some coherent basin-scale variations in M2 is interesting, and challenging. We changed the title of the paper (as suggested by the Editor), and replaced "Climate-scale changes" by "Large-scale changes".

**Therefore, if properly done, it would be interesting to convincingly prove (or disprove) the hypothesis that (for example) long-term NAO patterns affect tides coherently. However, as presently conceived and presented, am not convinced that the authors have really shown that this is occurring (or disproven it).**

We share the reviewer comment, and we analyzed more deeply the data to go further in the hypothesis. We developed a new part related to the statistical analysis, which led to new results. We also added new stations. More details are in the following.

**Issues include:**

**1. There is no attempt at a statistical correlation between the tide records and climate records such as the NAO index. While the tidal records in Europe superficially follow a similar trend as the filtered NAO, it is quite possible for data sets with few degrees of freedom (here, a decadal median) to resemble each other by random chance. For this reason, it is important to do some sort of significance testing and report statistics such as R2 and the p-value. Similarly, would be good to verify that the 6 US East coast records show a statistically insignificant correlation.**

We agree with the reviewer, and we made a statistical analysis of the data.

1) We computed the correlation between normalized M2 variations and climate indices (NAO and AO). To be consistent, we filtered out M2 variations on the same time window as NAO and AO (9 years). We computed the correlation since 1910, to have similar periods for all the stations. We considered that the correlation was significant only if the p-value was lower than 0.05 (95% statistical significance). The results are the following: for NAO only, 10 stations out of 12 show significant correlation. We found the strongest positive r-value in the North East Atlantic (Cuxhaven, Brest, Newlyn), with a maximum of 0.58 at Cuxhaven. This confirms the possible causal relationship between M2 variations and NAO, as suggested in the paper. We also found a strong anti-correlation with Halifax (-0.55). For AO, we found similar, but overall larger, r-values. This is not surprising as these two indices are highly correlated. We propose to add a figure showing the r-value at all the stations with (a) NAO and (b) AO.

Note that at Brest the data record starts in 1846. The correlation with NAO is significant from 1910, but not from 1864 (NAO index used in this study starts only in 1864). This can be explained by the M2 larger amplitude over all the XIXth century, which decreases between 1890 and 1910 (Figure 3a). This inconsistency was already noticed by the anonymous reviewer #1. However, the construction of dykes that partially closed the harbor of Brest since the end of the XIXth century may have altered the tide. To go further, the potential role of these successive constructions needs to be investigated (https://en.wikipedia.org/wiki/Brest_Arsenal). Cartwright (1972) made a first attempt to evaluate the influence of reducing the width of access to the harbour but did not take into account a potential role of dredging for which we have no information. This example underlines the complexity of interpretation of the variations when local and large-scale changes occur at the same time.

In the following, M2 variations, MSL and NAO are filtered over 9-year time windows and normalized. MSL are now corrected for land movement (estimations from SONEL website), which led to more consistent MSL trends at the basin scale.

2) We computed the r-value between M2 variations and two linear regression models. First, we fitted M2 variations with MSL variations only (model 1). We then computed the residual (M2 variations – model 1), and fitted this residual with a NAO linear model. The objective is to estimate the relative contribution of MSL and NAO in M2 variability. Models 1 and 2 may be expressed as:

$$Model\ 1 = \alpha MSL$$
$$Model\ 2 = \alpha MSL + \beta NAO$$

We checked that there was no significant correlation between NAO and MSL at the stations (there is no correlation at 7 stations, and the r-value is between 0.2 and 0.4 at 5 stations; note that there is no NAO-MSL corrrelation at the stations in the north (Halifax, Brest, Cuxhaven, Newlyn), where we found a significant NAO contribution compared to MSL - see below).

The three main results of this analysis are the following: the first result is that at first order, M2 varies with mean sea level (strong r-value for model 1). The second result is that the introduction of NAO in the model (model 2) allows to systematically increase the correlation (stronger r-value for model 2 than for model 1). This confirms that NAO-related mechanisms may explain part of the variability of M2. At some stations, this increase is quite large. For example, at Cuxhaven, the r-value is 0.65 for model 1, but reaches 0.85 for model 2. The third result is that we can estimate at each station the relative NAO contribution (compared to MSL) in M2 variability. Indeed, as in model 2 MSL and NAO are normalized, the ratio $\beta/\alpha$ represents roughly this relative NAO contribution. We found a significant NAO contribution at some stations (e.g. more than 30% at Halifax), whereas negligible at others (e.g.

5% at Eastport). Values suggest that the northern part of North Atlantic is more sensitive to NAO, with quite similar values. This suggests a possible basin scale coherence, with correlation on the northeast side, and anti-correlation on the northwest side. We propose to add in the paper a table including the r-value for models 1 and 2 and the ratio β/α. We also propose to add a figure with M2 variations, model 1, and model 2 at the stations where the correlation with model 2 is significant (r-value>0.3) and the NAO contribution significant ( β/α>0.25). This new figure shows that the model 2 better captures the M2 variations than model 1.

**2. Overall, it would seem to me that the data selection is incomplete. On the European side, there are some Dutch coastal records that predate 1920, such as Hoek van Holland and Delfzijl (maybe also others; perhaps check if the GESLA data set has them). Similarly, there are additional records on the US East Coast that could be used. These include Fernandina (1897-present; available from NOAA) and Sandy Hook (Available since ∼1910 from NOAA) (Note that Baltimore (1902-present) and Philadelphia (1901-present) also exist, but are not coastal stations). Moreover, there are a number of M2 estimates extending into the 1800s in New York Harbor (Talke et al. 2014 supplement and Chant et al., 2018), Boston (Talke et al., 2018), Eastport (Ray and Talke, 2019), and Long Island Sound (Kemp et al., 2017). Some additional estimates of tidal range at coastal stations deep into the 1800s are found in Talke & Jay (2020). These can be used as a proxy for M2 if divided by ∼2.**

We agree with the reviewer that it would be interesting to analyze more stations, and consequently we have changed our criteria of selection to extend the number of tide gauges. The stations suggested by the reviewer are missing because they did not match the criteria for tide gauge selection (see paragraph 2.2.1): we selected tides gauges from UHSLC, with time series starting before 1920, with at least 80 years with data, and a significant tide (M2>10 cm). For these reasons, the following stations were not selected:
- Hoek van Holland and Delfzijl are not in the UHSLC sea level database; note that they are in the GESLA dataset only for surges (not for sea levels). However, as we could easily download French stations online (data.shom.fr from the French Hydrographic Office), we analyzed 2 other stations from the North Sea (Calais, Dunkerque) – the results are provided later in this document.
- Fernandina is in the UHSLC database, but with only 62 years of data (<80 years), and over two distinct periods (1898-1923 and 1985-2018). For this reason, this station was not selected.
- Sandy Hook and Long Island are not in the UHSLC database. However, New York has been selected (see below), and as these 3 stations are very close together (<80 km). New York can supplement the study by covering this area.

New-York, Eastport and Boston are in the UHSLC, but starting respectively in 1920, 1929 and 1921. Following the reviewer's suggestion, we changed our criteria (times series starting before 1930 instead of 1920), in order to include these three stations. Note that Pensacola (in the Gulf of Mexico) and Tregde (in the North Sea) also start before 1930 (respectively 1923 and 1927), but were discarded due to the small tidal amplitude (M2<10 cm).

Finally, three stations were added – New-York, Eastport and Boston – leading to a total of 12 stations instead of 9. We also added values of M2 in 1862 at Eastport (from Talke and Ray, 2019), and New York (from Talke and Ray 2014). Note that this 1862 value was estimated from the supplementary Fig. S20. For Boston, there was no M2 value in Talke et al. (2018), however we mentioned the observed decrease of M2 between 1870s and 1920s (Talke et al., 2018).

We have added in the limitations and perspectives, that other possible relevant stations could be analyzed, provided different selection criteria are adopted, among them on the US coast Sandy Hook and Long Island Sound (Kemp et al., 2017), and in the North Sea Hoek van Holland and Delfzijl.

**3. Further, it's not clear how significant the data cut-off of 1920 is. If 1930 or 1940 were used, for example, how might results or conclusions change? A later cutoff would enable inclusion of a number of additional US East Coast stations (e.g., Mayport, Fort Pulaski, Sewells Point, Willets Point, Providence, Eastport, etc). Perhaps the same is true for Europe. How much would conclusions or patterns change if a slightly less restrictive date-cutoff were used? It would be good to use a statistics-driven reason for the date cutoff (e.g., degrees of freedom in a correlation analysis, or something like that), and to check the effect of relaxing the cutoff. Regardless, my guess, based off of Figure 2 of Talke & Jay 2020, is that patterns of M2 change might be even less coherent if more data are used. This is in part because there are so many local processes that can affect tide gauges, even those at or near the outlet of an estuary (for example, Mayport, Lewes, Charleston, Fernandina, Sewells Point, Boston, and Sandy Hook in the US; Cuxhaven in Germany; and the Saint John gauge in Canada are all "coastal" gauges that are actually at the mouth of or within an estuary).**

The effect of relaxing the cut-off has been tested (1930 instead of 1920) and even adopted (see above point 2.). This conducted to add three stations in the North West Atlantic coast (New-York, Eastport and Boston). The M2 variations at these stations are very similar to the ones of North West Atlantic with positive trends, i.e. Portland, Charleston, Key West (Figure 3 (b)). Similarly, for these 3 new stations, M2 decreases since 1980, and then increases since 1990, particularly for New York. We propose to update Figure 3 (b) with these 3 new stations. We also propose to update the figures with the trends (Figures 4 and 5).

**On the other hand, there is an interesting and not completely explained decrease in tidal range and M2 at Sandy Hook, The Battery, and Boston from the mid and early 1800s until the 1920s, and then an increase. In other words, it's possible there is a larger Northwest Atlantic Signal, in addition to local processes. However, while models such as Schindelegger et al. 2018 are able to see coherent trends, it has been challenging to see it in the data (as mentioned in this manuscript as well, also in Ray 2006).**

Yes, we added on Figure 3 (b) values of M2 in 1862 at Eastport (from Talke and Ray, 2019), New York (from Talke and Ray 2014), and mentioned at Boston the observed decrease of M2 between 1870s and 1920s (Talke et al., 2018). The normalized values in the XIX[th] at Portland, Eastport and New York are close together.

**4. A similar data comment holds for the trend-switch which is observed around 1990. It would be quite interesting if this is a coherent signal throughout the Atlantic. To prove, would suggest that there are many more gauges that are available that could be used to test this hypothesis. Again, a trend switch in 6 out of 9 gauges need not be statistically significant or could be argued to be local in nature.**

We agree with the reviewer that it would be interesting to find a coherent signal throughout the Atlantic. With the new data (see previously point 2.) and new statistical analysis (see previously point

1.), we are now able to partly interpret this trend-switch since 1990. The switch occurs in stations that are significantly influenced by NAO (high ratio β/α). In contrast, the stations that are not influenced by NAO - but mainly by MSL – show no switch, and even an increase of their trend (acceleration).

**However, would be more significant if you had 30 or 40 gauges and found a similar percentage shift between the 1950 to 1990 trend, vs. 1990 to the present, That would be quite interesting.**

We agree with the reviewer that it would be interesting to have more tide gauges. However, this would lead to add short records, whereas the present paper focuses on long-term records starting no later than 1930 and with at least 80 years with data. Moreover, the study would then be closer to Müller (2011), who selected tide gauges with at least 35 years of data prior to the year 1980, leading to 17 stations. Finally, shorter series are affected by stronger correlations between the NAO and the MSL (as they increase since 1960, see below the case studies of Calais and Dunkerque), which is problematic for the statistical analysis, when fitting model 2 to distinguish the influence of the MSL from that of the NAO. From 1910, only 5 stations among 12 show a significant correlation between MSL and NAO ( on average, r=0.28), whereas from 1960, this figure jumps to 9 stations, with higher r-value (on average, r=0.40).

**Also, the effect of moving the date (1985 or 1995 instead of 1990) might be worth investigating. If results depend on start date, the interpretation becomes less clear (and vice versa).**

As suggested by the reviewer, we investigated the effect of moving the date (1985 or 1995 instead of 1990). This does not change significantly the results. When we computed recent trends (from 1990) instead of long-term trends (since 1910), 5 stations showed a trend-switch (Newlyn, Brest, Cuxhaven, Halifax, Charleston). Moving the date from 1990 to 1985 leads to similar result: 3 stations show the same trend-switch (Newlyn, Cuxhaven, Halifax), and 2 stations show a significant decrease in the trend – but not enough to be a switch (from 0.13 to 0.01 mm/yr at Brest, from 0.32 to 0.06 mm/yr at Charleston, between 1910-trend and 1985-trend). Moving the date from 1990 to 1995 leads also to similar results: 4 stations show the same trend-switch (Brest, Cuxhaven, Halifax, Charleston), and 1 station shows a significant decrease in the trend – but not enough to be a switch (from 0.14 to 0.04 mm/ yr at Newlyn, between 1910-trend and 1995-trend). As a consequence, the interpretation of the results is not highly sensitive to the start date. We propose to add a short comment on this robust aspect in the the manuscript.

**5. The paper would also be improved by digging more deeply into mechanisms. Some discussion of how the NAO affects sea-level (and therefore, perhaps M2) is made, but it is quite qualitative. There are, however, a number of process-based studies that look into tidal changes at local, regional, and oceanic scale (see the Haigh et al. 2020 and Talke &Jay, 2020 reviews for references). The Atlantic is known to be near resonance, and there could be coupling between the shelf and the deep ocean (Arbic and Garret papers). Tide changes in the Gulf of Maine (and for that matter, Long Island Sound) can in theory radiate out to the larger ocean (e.g., Godin 1993). Also, it is known that the M2 amphidrome in confined, shallow seas moves with sea-level changes (see the references in the Haigh et al. review, or the Lee et al. 2017 and Ross et al. 2017 papers on the Chesapeake). Based on this, also perhaps on basin scale modeling (e.g., Schindelegger et al. 2018), what sorts of coherent patterns might be expected in the Atlantic based on historical sea-level rise? How much might this be spatially variable based on sea-level change caused by long-term NAO patterns? What might be the magnitude of the signal? The Schindelegger et al. 2018**

**paper shows that coherent changes across the basin are possible with sea-level rise, but are relatively small, for a given increment of sea-level change. Also, they show that some locations in the Atlantic are anti-correlated. How much of a sea-level change is needed, roughly, before a coherent longterm signal is findable in tide gauge data (given noise in data, etc)? One needs to know whether it's even possible (by the mechanisms listed) to obtain a secular coherence in basin-scale variability.**

We have added the shifting locations of amphidromic points under SLR scenarios (Pickering et al. 2017, Idier et al. 2018, Haigh et al. 2020) – see reviewer comments below, lines 182 and 218-219 from submitted paper. We also have substantially rewritten the paragraph on the impact of MSL rise on tide – see reviewer comment below, line 226 from submitted paper.

We share the questions of the reviewer, but it is challenging to determine what sorts of coherent patterns might be expected in the Atlantic caused by long-term NAO changes, only from papers on MSL rise effect on tide (e.g. Schindelegger et al. 2018). Note that Schindelegger et al. (2018) show that the sign of the M2 change is correctly reproduced at 80% of the tide gauges, but trend values tend to differ by a factor 3 to 5. Moreover, the response to a 0.5 m MSL rise is opposite between Cuxhaven and Brest (see Figure 6 from Schindelegger et al. (2018)), whereas the observed changes are of the same sign (positive) at these two stations (Figure 6, red dots, or Figure 3 (a) from our paper).

We conducted further investigations to test if the magnitude of sea-level pressure changes induced by large-scale atmospheric circulation (Figure 9, few tens of hPa) can generate the observed decadal-scale M2 changes (few cm). Note that we now express the Figure 9 in terms of hPa (and not cm). It is directly the difference of winter sea-level pressure between a NAO+ year (1989) and NAO- year (1969). Note also that M2 changes due to large-scale atmospheric circulation are only a small part of the total observed changes (few cm), the changes being also due to MSL rise (see the statistical analysis above).

The underlying mechanism invoked in the present paper (i.e. the influence of the atmospheric circulation on the tide) is very close to the one in Huess and Andersen (2001), except that we are at a larger time scale (decadal instead of seasonal). Huess and Andersen (2001) explains partly M2 seasonal variations through the effect of atmospheric circulation. They run a barotropic model in the North Sea, forced (1) with tides only and (2) with both tides and meteorological fields. Results show that the seasonal modulation is better captured when the model is forced with both tides and meteorological fields (Plate 2, top right, amplitude higher that 10 cm in the German Bight) rather than with tides only (Plate 2, top left, amplitude lower that 5 cm in the German Bight). It is important to underline that the model is barotropic, and that there is no effect of stratification, which may also play a role in M2 changes (see 3.3.6 in the review of Haigh et al, 2019).

At seasonal scale (instead of decadal scale, in the present paper), we computed monthly (instead of yearly) M2 variations at Cuxhaven over 5 years (2010-2015). Results show a seasonal cycle with a range of around 15 cm, maximum in summer, and minimum in January (which is consistent with Huess and Andersen (2001)). Similarly to Figure 9 in our paper (now in hPa), we computed the difference of monthly sea-level pressure between January and July (sea-level pressure data come from NOAA 20th-Century Reanalysis, Compo et al. 2011). We obtain values very close to the ones in Figure 9 (few tens of hPa). This shows that the order of magnitude of sea-level pressure changes (few tens of haPa) is consistent with M2 observed changes at Cuxhaven (few cm). The assumption is not strictly proven, but

provide reasonable new insights worth to be brought to the attention of the community for further investigations. As mentioned in the paper, dedicated simulations should be conducted to go further, and confirm or discard this hypothesis.

**In other words, a more clear hypothesis of what a basin scale shift in M2 tides might look like and whether it is detectable (given current understanding of processes) might help with the interpretation. What sort of excursion in M2 would you expect the NAO to cause, based on how it affects sea-level? Setting up a hypothesis with specific criteria that can be proven or disproven might help. It's ok, in my opinion, to have a paper with a non-detect result (a possible outcome here). However, since the set of non-detect papers is infinitely large, the added value could come from adding scientific or statistical insight into the problem. A great example of this is the Haigh et al. 2014 paper which showed that one would need to wait a couple decades before being able to analyze recent sea-level acceleration at an acceptable level of confidence. Is something similar true here? My qualitative guess is that it might be hard to see a basin scale coherence in data, but that regional scale effects that are driven by a similar process like sea-level fluctuations can perhaps be detected (see for example the Devlin et al. Papers). Basically, the paper would be improved by more specifically investigating what sort of change is needed (and what sort of data quality/signal to noise is needed) before it might become possible to discern coherent climate effects on tides across the entire basin.**

We have added in the paper some new insight from the statistical analysis. M2 changes are correlated at first order with mean sea level rise, and at second order with NAO, but only at some stations. These are mainly located in the northern part of North Atlantic. There are correlated on the northeast side, but anti-correlated on the northwest side. This suggests a basin scale coherence in the data. Dedicated simulations should be conducted to go further on this hypothesis.

**Detailed comments:**
**Line 14: Would also cite the review of Talke & Jay, 2020, since the historical changes in tidal range shown therein are relevant to this paper.**
Yes, we added the reference Talke & Jay, 2020.

**Line 19: "Long-term changes in tidal constituents are rather small" Would modify this to specify "at coastal stations". As shown in Talke & Jay 2020 (and refs therein), the secular change at many estuary and tidal river gauge stations is huge.**
Yes, we added "at coastal stations". We also mentioned that changes are larger in many estuaries and rivers, referencing to Talke & Jay 2020.

**Line 20 "still poorly understood"– Not sure I would say this. Some of the mysteries are being solved (see the review papers), while some issues remain. Maybe rephrase?**
Yes, we rephrased. The physical causes of these changes are generally difficult to understand. The complexity comes first from the combination of local and regional changes. Moreover, regional changes may be a combination of different processes, largely dependent from each other, and which interact together – it is then challenging to identify separately which are the processes at the origin of the changes, and in which proportion are they contributing.

**Line 46-48: Check grammar; grammar of list Is not quite right.**
Yes, the list has been correctly rewritten.

**Line 60-63: There are some M2 results for 19th century US stations that you could/should use. See for example the supplement of Talke et al., 2014 or Chant et al., 2018 for New York and Sandy Hook. See Ray & Talke (2019) for Eastport and Portland. See also Talke et al., 2018 for Boston. Finally, there are multiple tidal ranges shown in the Talke & Jay 2020 review paper. These can be divided by two to get an estimate of M2 over time.**

The value of M2 at Portland from Ray & Talke (2019) was already added in the paper (blue star on Figure 3 (b)). Following the reviewer suggestion, we also added New York, Eastport and Boston, by relaxing the criteria of selection of stations (we now select stations with time series starting before 1930, instead of 1920). As mentioned previously, we also added values of M2 in 1862 at Eastport (from Talke and Ray, 2019), New York (from Talke and Ray 2014), and we mentioned the observed decrease of M2 at Boston between 1870s and 1920s (Talke et al., 2018).

**Also, Sandy Hook data from around 1910 is available at the NOAA site. The datum is wrong, but that shouldn't matter for tidal analysis.**

Sandy Hook is very close to New York which has been included in the study (see above a previous response). We have added in the limitations and perspectives, that other relevant stations could be analyzed, among them Sandy Hook.

**Table 1: Since you are using Cuxhaven, why not also use some of the Dutch stations? In the records I have, Hoek van Holland starts in 1900, and Delfzijl starts in 1876. Maybe there are earlier ones as well, see for example the Hollebrandse 2005 thesis. You could check if they are in the Gesla dataset and/or contact the Dutch. Data used to be available at waterbase.nl, but not sure that works anymore.**

As mentioned previously, Hoek van Holland and Delfzijl are not in UHSLC database, and not in the GESLA database. Figure 1 shows that there is no Dutch data in UHSLC dataset. However, we agree that having more points in the North Sea would be interesting. As we can easily download French stations online (data.shom.fr), we led a similar analysis on Calais and Dunkerque tide gauges, located in the North of France (North Sea). Calais starts in 1941, with only two years 1941-1942, and then a gap until 1965, and data from 1965 up to now. Dunkerque starts in 1956. The results confirm that the variations at these two stations are similar to the variations at the 3 other stations in the North East Atlantic (Newlyn, Brest, Cuxhaven, Figure 3(a)). M2 increases from 1960 to 1990, and then becomes more steady since 1990. Similarly to the three other stations in North East Atlantic, the trends are decreasing when they are computed only since 1990. However, the main difficulty with these short time series (since 1960) is that the correlation between NAO and MSL is significant, as over this period (1960-2018) NAO and MSL are increasing together. For example, at Dunkerque, the correlation coefficient is 0.53. It is then not possible to fit model 1 and model 2, in the statistical analysis assuming them independent. For this reason, we choose not to include in the paper these short records. However, we propose to mention the consistency in the variations of M2 between Calais Dunkerque and the other stations over North East Atlantic.

**Table 1: Why is Fernandina (1897-present) not used?**

As mentioned previously, Fernandina does not match with our selection criteria: this station has only 62 years with data, our criteria being at least 80 years with data. Moreover, data cover two distinct periods (1898-1923 and 1985-2018).

**Table 1: Unclear what the meaning of mean sea-level is. What is the datum? Why not include the trend, rather than an absolute measurement (which is not necessarily meaningful).**
The MSL is referenced to an arbitrary reference. As mentioned by the reviewer (and also by the Editor), this is not necessarily meaningful. This column has been deleted.

**Line 69: "This constraint resulted in excluding between 1 and 9 years". Unclear what you mean. You mean for each station?**
Yes, we added "for each station".

**Line 70 seasonal variation: where? Again, non-coastal stations will see more variability. Also, in the North Sea the change is higher (e.g., Graewe et al. 2014). So, maybe be specific and mention the Atlantic.**
The seasonal variation is significant in coastal areas and polar regions (Müller et al. 2014). In the North Atlantic, the largest values are over the North East Atlantic (English Channel and the North Sea, see Figure 3 in Müller et al., 2014). As suggested by the reviewer, we have added "in the North Atlantic", and also referred to Gräwe et al. (2014).

**Line 74 to 91: This would seem to be pretty standard nodal correction theory. Unless you can explain what is unique about your approach, would suggest greatly condensing this and simply citing an older study that discusses this in more detail**
We have moved the technical details concerning nodal corrections to an Appendix.

**Line 98-110: What is the rational for using the more complex method here? As you later state, it doesn't lead to significantly different results. I guess it's interesting that it can work for small time series. Do you see any evidence of changes to nodal cycle? There are a few papers on this recently. However, am not convinced that there is a physical reason for these observations, vs. just statistical noise. Could be something to look into, though if there is a coherence between nodal cycle variations in the western and eastern Atlantic, would be worth commenting on. Otherwise, not sure that you need the complex approach to nodal cycle characterization.**
As underlined by the reviewer, the main interest of this approach is that it can work with short time series. Following the reviewer's suggestion, we simplified this part on nodal modulation, and have moved it to an Appendix.

**Line 117, Equation 5: Can you explore/motivate the use of the standard deviation a bit more? A potential issue is that sigma may also reflect errors in the gauge data (e.g., timing errors, etc). Some exploration would be good as to whether this is a factor. For example, does sigma change as a function of time? If there is a decrease in sigma around 1990 or 1995 when new digital gauges started being used, at least in the US then it might indicate that instrumental issues are potentially affecting your results. See for example Zaron & Jay, 2014, who concluded that some constituent trends in the Pacific are spurious. Using some sort of method to validate the causes of sigma would therefore be good. How can we be sure that a few years of non-optimal data are not biasing sigma? The method of Zaron & Jay, 2014 could be used, or the method used by Talke et al. 2018 to assess timing errors could be used (see their supplement).**

Here, we used the standard deviation computed over the period 1910-2010 only to normalize the data, in order to compare all the stations together (i.e. on the same figure). This allows to clearly see on

Figure 3 (b) the change that occurs around 1980 (M2 decreases over the period 1980-1990 at the 6 stations – note that now we have 3 more stations).

We did not investigate the changes of standard deviation with time, but we agree with the reviewer that this kind of exploration would be interesting. However, here, the sigma is used only in order to normalize M2, so any bias of the sigma due to few years of non-optimal data should not impact the results of the study.

**Figure 2: This figure may have some educational/explanatory value, but it's not really a new result. One could consider removing.**
We propose to move this figure to the Appendix, related to nodal corrections.

**Line 128 maybe remove "are essential, as they"? Doesn't' really add much to sentence**
Yes, this has been removed.

**Line 156 "is no linear trends" should be "is no linear trend" or "are no linear trends"**
Yes, this has been corrected.

**Line 158 "curve is flattening" should be "curve flattens"**
Yes, this has been corrected.

**Line 159 Remove "yet" in "yet noticed"**
Yes, this has been corrected.

**Line 163-164 Not sure that the lack of an astronomical explanation automatically implies a solid earth-ocean-atmosphere coupling system cause. Am not even sure what is meant by that. A few more logical steps are needed before a reader can believe that**
The sentence has been removed.

**Figure 3: Charleston is a harbor city with a channel that has probably been subjected to dredging, though I haven't looked into it extensively. Can you discuss how/whether this impacts results?**
Yes, we have added that Charleston has probably been subjected to dredging. Channel deepening increases the water depths, which reduces the effective drag, leading to tidal range amplification, that may be particularly large in estuaries (Ralston at al., 2019; Talke and Jay, 2020).

**Line 165: The Delfzijl station starts in 1876, so would be worth comparing to Cuxhaven. It's probably somewhat impacted by long term changes to the Ems estuary tides.**
As mentioned previously, Delfzijl was not analyzed (selection criteria). However, it is cited as a relevant station in the perspectives of the paper.

**Then again, Cuxhaven is probably influenced by the large change to Elbe tides. See for example Winterwerp et al. 2013.**
Following the reviewer suggestion, we have mentioned in the paper that Cuxhaven is located in the Elbe estuary, and that some river engineering works, as narrowing and deepening, may induced tidal amplification (Wintewerp & Wang, 2013; Wintewerp et al., 2013)

**In general, there are quite a few papers out of Germany (e.g., Jensen et al. 2003, 2005 conference papers, and maybe Mudersbach et al. 2013 (?) that discuss a big increase in tidal range from about 1960 to the 1990s on the German coast. More recently, I've been told this has slowed or reversed (though I'm not sure there is a paper on that yet). Another good reference is the Hollebrandse 2005 Master's thesis on Dutch gauges.**

In the present paper, variations at Cuxhaven (Germany) show an increase in M2 from 1960 to 1990s, followed by a decrease from 1990. As suggested by the reviewer, we have added that different authors noticed a similar increase of tidal range from 1960 to 1990 in the southern North Sea. Hollebrandse (2005) found a gradual increase of tidal range during the period 1955-1980 at all the stations of the Dutch coast (5 stations including Hoek van Holland) and the German coast (7 stations). Mudersbach et al. (2013) found a significant increase in M2 amplitude at Cuxhaven since around the mid-1950s.

**Line 166 The Talke & Jay (2013, 2017) paper and report are good references for sea-level/tide data archaeology, as are Peauvreau 2008 and some of the papers by Marta Marcos.**
Yes, we have added references to Talke & Jay (2013, 2017), Pouvreau 2008 (already cited in the paper), and Marcos et al. (2011) for data archaeology. We also have added these references line 327 of the submitted paper, in the conclusions.

**Line 172 Again, note the Winterwerp et al. 2013 paper that includes the Elbe. There are probably some German references too. The Talke & Jay 2020 and Haigh et al. 2020 reviews discuss tidal resonance (see also references therein).**
Yes, we have precised that the environmental setting of Cuxhaven in the Elbe estuary could introduce some amplification (Winterwerp and Wang, 2013; Winterwerp et al., 2013).

**Line 182 The Ray (2006) and Ray & Talke (2019) papers discuss change in M2 trend in the 1980s in the Gulf of Maine maybe reference.**
Yes, we have added that this increase in Gulf of Maine was reported by Ray (2006) and Ray and Talke (2019).

**Line 186 There are many other papers that have explored Gulf of Maine resonance besides Ray & Talke. That is not perhaps the best example. See e.g. the discussion and references in the Talke & Jay 2020 or Haigh et al. 2020 review, in addition to the Garret and Godin reference.**
We have also referred to the reviews Talke & Jay (2020) and Haigh et al. (2020).

**General comment: The Godin 1993 reference, and also for that matter the Arbic and Garret and Arbic et al.papers, are interesting because they discuss how resonance on a small scale (Gulf of Maine, Continental Shelf) can affect the larger Atlantic. See also the Platzman papers on resonance from the 1970s. All this could/should be discussed and investigated, since it gets at the idea that there might be a mechanism through which western and eastern Atlantic tides could be coupled. Is there reason to believe there might be? In a sense, this is an implicit hypothesis that is being investigated here, through correlation with climate indices. However, it would be helpful to motivate and explore physical mechanisms as well. Further, it might be helpful to explicitly pose a hypothesis in the introduction, such as "is there any evidence for correlated/coupled changes in tides that might provide evidence for cross-Atlantic connectivity"?**
Yes, the higher correlations with NAO are located in the northern part of North Atlantic, suggesting possible coupled changes between the eastern and western coasts.

**Line 190-194: Why not use the Eastport data points from Ray & Talke, or at least discuss? The composite Pulpit Harbor/Bar Harbor data set might also be worth discussion. Boston is a possibility, too, though it is influenced by local processes as well (see the Talke et al. 2018 paper. . .).**

The values at Eastport and Pulpit/Bar Harbor from Ray&Talke (2019) were not used because these stations were not selected. Now, we added Eastport (from Talke and Ray, 2019), New York (from Talke and Ray 2014), and we have mentioned the observed decrease of M2 at Boston between 1870s and 1920s (Talke et al., 2018).

**Line 198 : It would be good to compare Atlantic City to Sandy Hook and The Battery (see Talke et al, 2014 and Chant et al., 2018). In fact, the case of Sandy Hook and The Battery/Governors Island are interesting, since there is a marked decrease from the 1860s until the 1920s or 1930s, and then an increase. Chant et al. (2018) show an even bigger change in nearby Newark Bay, though the 19th century data there are based on very short time series. In any case, the results are sort of consistent with the results at Brest. Dredging may have at least somewhat caused the 20th century amplification (see Ralston et al., 2019), and work at the channel mouth may have cause the early 20th century changes (Marmer, 1935). Also, Boston showed a similar, large decrease in tidal range through the 1920s, then an increase. While this is likely in large part local, Talke et al. 2018 did note that it's similar to the pattern observed at Sandy Hook.**

Yes, we mentioned the decrease observed between 1860s until 1920s at 4 stations.

**Line 218-219: Would also look into/discuss amphidrome changes. See the Haigh et al. review and references therein.**

Yes, we have added that the shifting locations of amphidromic points could also play a role (Haigh et al., 2020). In the North Sea, different authors show a possible migration of the present day amphidromes, under a 2 m sea-level rise scenario (Pickering et al., 2012; Idier et al., 2017).

**Line 218-223: These are very short paragraphs and not that well developed. Some more thought would be good. For example, "The trends have to be interpreted carefully" is perhaps an obvious statement (hopefully there is not a case when it is ok to interpret trends haphazardly. . .).**

We have developed the paragraph on amphidromic points (see above, reviewer comment on lines 218-219 from submitted paper), and rephrased the one on trends.

**Line 226 Somewhat misleading statement. Ray & Talke (2019) are referencing other results when they state that MSL rise only partly explains trends. Furthermore, they only focused on Gulf of Maine. Would instead look into some of the studies that have more carefully looked at SLR effects, such as Schindelegger et al. 2018 or Greenberg et al. 2012.**

Yes, this paragraph on possible link with mean sea level rise has been substantially rewritten.

**Line 228 "than mean sea" should be "as mean sea"**

Yes, it has been corrected.

**Line 230-235: The Pickering papers are for large sea-level rise scenarios, but don't retrospectively look at 20th century rise (if memory serves). Hence, is it a fair comparison? There are probably some papers or reports that discuss reasons for North Sea changes more thoroughly please look into and review.**

Yes, this paragraph has been substantially rewritten.

**Figure 6: One could include the Portland sea-level data point from Talke & Ray 2019. If you include the Battery, then a longer data set is possible. Not sure however if this graph is needed or is critical for the story. It is not really a result of this study, just a replotting of other results. There is no clear analysis of how tides might be influenced by SLR it's basically just a literature review.**

We now fit a model on MSL (see model 1 in our statistical analysis), so the figure 6 showing annual MSL becomes more important in the paper. For this reason, we kept it. However, we plotted MSL corrected from land movement (because the MSL model in "model 1 = α MSL" is corrected for land movement), instead of relative MSL.

**Line 247-250. Wouldn't storminess also impact tidal constituents, at least on the shelf or in a harbor? I think there are some references on that. I came across a Pugh reference at some point for the Irish Sea, if memory serves. The Graewe et al. 2014 reference also discuss this for the North Sea, I think. In any case, wind stress and wave breaking and these sort of things represent an input of turbulent kinetic energy and could in theory affect tides at some stations, if there are climate-based shifts in storminess. In the context of this paper, Talke et al. 2014 showed that the probability of large storm tides in New York goes up when the NAO is negative. There are also known NAO effects in Europe (see the Woodworth et al. (2007?) paper). Does this matter for tides? Might be something to at least investigate.**

In the North Sea, the wind stress and the wave breaking affect firstly the surges (Pineau-Guillou et al. 2020, Ocean Modelling). In coastal areas, the wind stress contribution is more effective due to shallow waters, water pileup along the coast, as well as resonant effects (Moon et al., 2009; Bertin et al., 2012). In addition, in nearshore areas, the radiation stress, i.e. the momentum flux carried by the waves, generate nearshore currents and wave setup (i.e. additional surge) when the waves dissipate (Brown et al., 2010; Bertin et al., 2015; Choi et al., 2018). Concerning a possible connection between storm surges and NAO, Menéndez & Woodworth (2010) found a significant correlation between NAO and extreme high waters in the North Sea. Marcos & Woodworth (2017) found a correlation between NAO and skew surges.

Here, we did not study the impact of storminess on tidal constituents. However, we agree with the reviewer that it would be interesting to investigate. We have added in the limitations and perspectives, that dedicated studies are necessary to estimate if changes in storminess could affect significantly tidal constituents.

**By the way, it's not clear to me that a measured decrease in M2 during periods of stormy weather is a real change in M2. Another (perhaps not mutually exclusive) explanation is that depth changes during storms alter the phase speed of the tides, such that they arrive a bit earlier than usual (See for example Horburgh and Wilson 2007). A period with a lot of ups and downs in mean sea-level is going to cause lots of phase speed variations, more spectral spread (cusping), but decreased amplitude. Just as timing errors can cause a decrease in measured M2 (see Zaron & Jay 2014), so would changes in phase speed.**

We agree with the reviewer that in period of stormy weather, the depth changes (because of the surge), and the tide will arrive a bit earlier than usual. As mentioned previously, we did not go further on the impact of storminess, but we have mentioned that dedicated studies would be valuable.

**Line 254 "possible role of stratification" for what? Would clarify, e.g., something like "possible role of stratification on secular tidal trends"**
Yes, this has been added.

**Line 255 "between these processes" what processes? Maybe be specific.**
By "processes", we meant atmospheric circulation and ocean stratification. The sentence has been rephrased.

**Line 266 How can the NAO decrease globally? It is specific to the North Atlantic.**
By "globally", we meant "overall" (and not worldwide). We have corrected the sentence.

**Line 278 "Pushed southern" should be "pushed southerly".**
Yes, we have corrected the sentence.

**Line 289 Might be good to discuss the role of wind earlier. See notes above.**
Yes, we discussed earlier the role of wind, with reference to Devlin et al. (2018). Devlin et al. (2018) shows that the impact of atmospheric circulation (via the wind stress, through Ekman current) on M2 seasonal cycle could be significant and comparable to the effect of permanent (geostrophic) currents.

**Line 291-292 Maybe, but there is quite a bit of variance in all the plots and it seems like a couple curves looking similar could easily happen from random chance. Unless you can figure out the statistical robustness of these results, would perhaps avoid ascribing M2 behavior at a few locations to NAO.**
Yes, this sentence has been removed. The statistical analysis (see above) shows now anti-correlation between NAO and M2 variations in Halifax.

**Line 332 The Devlin et al. papers discuss correlations between sea-level anomalies and tidal anomalies, and possible reasons for them. In a way, you are trying to do something similar, but over a larger time scale. However, there is little statistical correlation or significance testing done here. Would suggest this be done.**
For the seasonal variation of M2, we could have added Devlin et al. 2018 as a reference (even if it focuses of Southeast Asian Waters). However, this sentence relative to the seasonal variation of M2 has been removed (lines 330-332), as we now consider years with at least 75% (following a suggestion from the Editor).

Note that the Devlin et al. 2018 paper is interesting in the context of our study. It shows the impact of the wind stress (via Ekman current) on M2, at a seasonal scale. As mentioned by the reviewer, we are investigating a similar mechanism (effect of the atmospheric circulation) but at a larger time scale (decadal instead of seasonal).

**Please also note the supplement to this comment:**
**https://os.copernicus.org/preprints/os-2020-56/os-2020-56-RC2-supplement.pdf**
Note that we did not notice any differences between the main file and the supplement.

---

## Author Comment (AC2) · 14 Sep 2020

**Response to Referee Comment #1 (Anonymous)**

We thank the reviewer for his careful reading of the paper. The reviewer remarks led to important changes in the manuscript. We went further in the analysis, we have now new results, and the manuscript has been substantially rewritten.

In the following, the reviewer comments are in bold, our replies are in normal font.

**In this paper, nine very long (>80-year) tide gauge records along North Atlantic coasts are analyzed for secular changes in the M2 amplitude. The series are compared both among each other and with climate mode indices in an attempt to relate the observed amplitude changes to large-scale forcing mechanisms. Unfortunately, the paper is of limited scope and the methods are not innovative.**
We agree with the reviewer that the methods may not be innovative, but they are robust, and our tidal analyses are rigorously undertaken and as accurate as one can expect from the state-of-the-art knowledge.

**The arguments put forth to link the observed M2 changes to the North Atlantic Oscillations (NAO) are fallacious (see below) and invalidate exactly that part of the paper that is thought to break fresh ground compared to similar analyses in the past (e.g., Mueller 2011, GRL).**
We bring now quantitative insights on the possible influence of NAO, which was already mentioned by Müller (2011) on the basis of qualitative criteria.

**These shortcomings are not easily redressed in a revision, and I therefore recommend the manuscript to be rejected. Overall, the study offers too little new insight. It merely highlights similarities without investigating the underlying processes.**
We went further in the data exploration to link the M2 variations with the NAO. We have now some new insight from the statistical analysis. We fitted a series of linear regression models, either only MSL-dependent (model 1) or MSL and NAO-dependent (model 2) on M2 variations (see below for the details). We found that M2 variations are correlated at first order with MSL, and at second order with NAO, at many stations. Taking into account the NAO in the model (model 2) systematically increases the correlations (compared to model 1). We estimated the part of the contribution of the NAO compared to MSL in the M2 variations. We found that this contribution is at some stations negligible (<10%), but at others significant, e.g. more than 30% at Cuxhaven or Halifax. These stations are mainly located in the northern part of the North Atlantic. The correlation with the NAO is positive on the northeast side, but negative on the northwest side (except in the Gulf of Maine). This suggests a basin scale coherence in the data.

**Major issues:**
**[-] The authors' explanation of the pronounced M2 increase over 1960–1990 at the three European stations (~3 cm at Brest and Newlyn, ~10 cm at Cuxhaven) in terms of typical NAO sea-level pressure patterns is flawed. The response of sea level (or water column thickness) to changing pressure loading on time scales longer than a few weeks is static isostatic (IB), creating a sea-level difference of about 6–10 cm at the location of the tide gauges, according to the authors' plots (Figures 8 and 9). These changes in water depth are simply too small to cause the observed M2 amplitude trends. Typically, sea-level changes alter the propagation characteristics**

**of tides in shallow water such that one can expect perturbations in the M2 amplitude of 1–5% relative to the imposed water depth change, see the modeling results by Schindelegger et al. (2018). Cuxhaven may be an exception of that rule, although a 6-cm increase in sea level from atmospheric pressure will engender less than 2 cm changes in the M2 amplitude (Figure 8 of Schindelegger et al.).**

Schindelegger et al. (2018) clearly underestimate the M2 response to MSL rise in terms of magnitude (see their Figure 7). Simulations show that the sign of the M2 trend is correctly reproduced at 80% of the tide gauges, but trend values tend to differ by a factor 3 to 5. Moreover, the response to a 0.5 m MSL rise is opposite between Cuxhaven and Brest (see Figure 6), whereas the observed changes are of the same sign (positive) at these two stations (their Figure 6, red dots). In terms of magnitude, simulations show M2 changes of around 1 cm (absolute value) at Brest and Newlyn, and 3 cm at Cuxhaven for a 0.5 m MSL rise (see Figure 6), whereas observed M2 changes for a smaller MSL rise (0.2 m MSL rise over the XXth) are as large as 5 cm , 3 cm and 15 cm at Brest, Newlyn and Cuxhaven, respectively (results from our study). Schindelegger et al. (2018) conclude that "magnitudes of observed and modeled M2 trends are within a factor of 4 (or less) from each other in nearly 50% of the considered cases". These strong discrepancies between simulations and observations have to be carefully interpreted. (1) Numerical simulations are great tools to perform sensitivity studies and understand processes (as in Schindelegger et al. (2018)), but results in terms of values may be far from ground truth for many reasons as wide spatial resolution (~10 km in Schindelegger et al. (2018)), coarse bathymetry, rough parameterizations, tuning parameters, inadequate forcing, lack of coupling (e.g. with atmosphere). (2) The large discrepancies between the model and the observations also strongly suggest that mean sea level rise is not the only process that may explain M2 changes – other large-scale processes, in addition to local processes, may combine and interact together.

We conducted further investigations to test if the magnitude of sea-level pressure changes induced by large-scale atmospheric circulation (Figure 9, few tens of hPa) can generate the observed decadal-scale M2 changes (few cm). Note that we now express the Figure 9 in terms of hPa (and not cm). It is directly the difference of winter sea-level pressure between a NAO+ year (1989) and NAO- year (1969). Note also that M2 changes due to large-scale atmospheric circulation are only a small part of the total observed changes (few cm), the changes being also due to MSL rise (see the statistical analysis).

The underlying mechanism invoked in the present paper (i.e. the influence of the atmospheric circulation on the tide) is very close to the one in Huess and Andersen (2001), except that we are at a larger time scale (decadal instead of seasonal). Huess and Andersen (2001) explains partly M2 seasonal variations through the effect of atmospheric circulation. They run a barotropic model in the North Sea, forced (1) with tides only and (2) with both tides and meteorological fields. Results show that the seasonal modulation is better captured when the model is forced with both tides and meteorological fields (Plate 2, top right, amplitude higher that 10 cm in the German Bight) rather than with tides only (Plate 2, top left, amplitude lower that 5 cm in the German Bight). It is important to underline that the model is barotropic, and that there is no effect of stratification, which may also play a role in M2 changes (see 3.3.6 in the review of Haigh et al, 2019).

At seasonal scale (instead of decadal scale, in the present paper), we computed monthly (instead of yearly) M2 variations at Cuxhaven over 5 years (2010-2015). Results show a seasonal cycle with a range of around 15 cm, maximum in summer, and minimum in January (which is consistent with Huess and Andersen (2001)). Similarly to Figure 9 in our paper (now in hPa), we computed the difference of

monthly sea-level pressure between January and July (sea-level pressure data come from NOAA 20th-Century Reanalysis, Compo et al. 2011). We obtain values very close to the ones in Figure 9 (few tens of hPa). This shows that the order of magnitude of sea-level pressure changes (few tens of hPa) is consistent with M2 observed changes at Cuxhaven (few cm). The assumption is not strictly proven, but we provide reasonable new insights worth to be brought to the attention of the community for further investigations. As mentioned in the paper, dedicated simulations should be conducted to go further, and confirm or discard this hypothesis.

**The authors circumvent the problem by assuming higher sea-level changes in areas distant to the gauges (-20 cm near Island) and picking a 10% sensitivity of M2 amplitudes to water depths from literature – an inordinate value that only holds in very shallow settings (e.g., estuaries) and not across entire shelf regions.**
As all the tide gauges are in coastal areas, and potentially in estuaries or very shallow waters (e.g. Cuxhaven is located in the Elbe estuary) , we focused on values nearshore rather than offshore. Idier et al. (2017) show that depending on the location, the changes can account for +/−15% of regional SLR, whereas Schindelegger et al. (2017) reports relative magnitude of 1-5 % per century in the North Atlantic. The value of 10% is an order of magnitude of the changes in shallow waters. As mentioned previously, Schindelegger et al. (2018) correctly simulate the sign of M2 trends at 36 of 45 stations, but in terms of magnitude, there are strong discrepancies between the model and the observations.

Following the reviewer suggestion, we have substantially rewritten this part, and introduced an analogy with the seasonal variation (see above).

**Moreover, Figure 8 displays higher atmospheric pressures around the European mainland in 1989, implying that sea level actually dropped relative to 1969, opposite to what is shown in Figure 9.**
Yes, we thank the reviewer to point out this error, there was an error of sign. Figure 8 shows the difference between years 1969 and 1989 (and not the contrary). However, we express now this differences in terms of hPa, and the figure has been updated (hPa instead cm for the unit).

**So that's an inconsistency on its own, but more alarmingly for the authors' theory, numerical modeling (see references) strongly suggests that an increase of M2 in the German Bight, as observed at Cuxhaven, actually requires local sea level to rise, not to fall.**
Yes, but the same simulations also suggest that an increase of M2 in the Western Channel, as observed at Brest, actually requires sea level to fall, not to rise. The underlying problem is that these simulations (e.g. Schindelegger et al., 2018; Pickering et al. 2017) show opposite signs between Brest and Cuxhaven, whereas observations show the same sign (red dots on Figure 6 of Schindelegger et al. (2018) or Figure 3 (a) of our paper).

**[-] A key argument is that the low-frequency winter NAO index (Figure 7) is similar to the evolution of the M2 amplitude at the three European stations. Such an important point in the paper should be substantiated by an appropriate plot (in which annual M2 changes would be filtered using the same 9-year running median as the NAO index).**
Yes, we agree with the reviewer. We propose to add a new figure with M2 variations (filtered in the same way than NAO), model 1, and model 2 (see below) at the stations where the correlation with model 2 is significant (p<0.05 and r-value>0.3) and the NAO contribution significant ( β/α>0.25, see

its definition below). This new figure shows that the model 2 better captures the M2 variations than model 1.

**More importantly, the mentioned similarity is never established in a quantitative sense, e.g., by tabulating correlation measures and their statistical significance considering effective degrees of freedom. In fact, the Brest time series in Figure 3a seems to have rather little in common with the NAO time series as it has a dip in the 1980s (when NAO steadily increases) and features an all-time high in the late 19th century (when NAO just erratically switches sign). For Newlyn and Cuxhaven, I expect the correlations to be higher, although the timing of individual peaks might be different. Such phase lags and leads are not easily explained in terms of physics; certainly not within the framework proposed here, because both sea level and subsequently tides would adjust instantaneously to NAO-related atmospheric pressure loading.**
We agree with the reviewer, and to establish the similarity in a more quantitative sense, we conducted a statistical analysis.

1) We computed the correlation between normalized M2 variations and climate indices (NAO and AO). To be consistent, we filtered out M2 variations on the same time window as NAO and AO (9 years). We computed the correlation since 1910, to have similar periods for all the stations. We considered that the correlation was significant only if the p-value was lower than 0.05 (95% significance level). The results are the following: for NAO only, 10 stations out of 12 show significant correlation. We found the strongest positive r-value in the North East Atlantic (Cuxhaven, Brest, Newlyn), with a maximum of 0.58 at Cuxhaven. This confirms the possible causal relationship between M2 variations and NAO, as suggested in the paper. We also found a strong anti-correlation with Halifax (-0.55). For AO, we found similar, but generally overall larger, r-values. This is not surprising as these two indices are highly correlated. We propose to add in the paper a figure showing the r-value at all the stations with (a) NAO and (b) AO.

Note that at Brest the data records starts in 1846. The correlation with NAO is significant from 1910, but not from 1864 (NAO index used in this study starts only in 1864). This can be explained by the M2 larger amplitude over all the XIXth century, which decreases between 1890 and 1910 (Figure 3a). This inconsistency was already noticed by the reviewer. However, the construction of dykes that partially closed the harbor of Brest since the end of the XIXth century may have altered the tide. To go further, the potential role of these successive constructions needs to be investigated (https://en.wikipedia.org/wiki/Brest_Arsenal). Cartwight (1972) made a first attempt to evaluate the influence of reducing the width of access to the harbour but did not take into account a potential role of dredging for which we have no information. This example underlines the complexity of interpretation of the variations when local and large-scale changes occur at the same time.

In the following, M2 variations, MSL and NAO are filtered over 9-year time windows and normalized. MSL are now corrected for land movement (estimations from SONEL website), which led to more consistent MSL trends at the basin scale.

2) We computed the r-value between M2 variations and two linear regression models. First, we fitted M2 variations with  MSL variations only (model 1). We then computed the residual (M2 variations – model 1), and fitted this residual with a NAO linear model. The objective is to estimate the relative contribution of MSL and NAO in M2 variability. Models 1 and 2 may be expressed as:

Model 1 = αMSL
Model 2 = αMSL+βNAO

We checked that there was no significant correlation between NAO and MSL at the stations (there is no correlation at 7 stations, and r-value is between 0.2 and 0.4 at 5 stations; note that there is no NAO-MSL corrrelation at the stations in the north (Halifax, Brest, Cuxhaven, Newlyn), where we found a significant NAO contribution compared to MSL - see below).

The three main results of this analysis are the following: the first result is that at first order, M2 varies with mean sea level (strong r-value for model 1). The second result is that the introduction of NAO in the model (model 2) allows to systematically increase the correlation (stronger r-value for model 2 than for model 1). This confirms that NAO-related mechanisms may explain a part of the variability of M2. At some stations, this increase is quite large. For example, at Cuxhaven, the r-value is 0.65 for model 1, but reaches 0.85 for model 2. The third result is that we can estimate at each station the relative NAO contribution (compared to MSL) in M2 variability. Indeed, as in model 2 MSL and NAO are normalized, the ratio β/α represents roughly this relative NAO contribution. We found a significant NAO contributions at some stations (e.g. more than 30% at Halifax), whereas negligible at others (e.g. 5% at Eastport). Values suggest that the northern part of North Atlantic is more sensitive to NAO, with quite similar values. This suggests a possible basin scale coherence, with correlation on the northeast side, and anti-correlation on the northwest side. We propose to add in the paper a table including the r-value for models 1 and 2 and the ratio β/α .

**[-] The data basis on the European Shelf (three stations) is very shaky. It would be desirable to make the analysis more robust by adding results from tide gauges that are somewhat shorter but still provide good coverage of the period with distinct variability in NAO (1960 onwards).**
We agree with the reviewer that it would be interesting to have more tide gauges. However, this would lead to add short records, whereas the present paper focuses on long-term records starting no later than 1930 and with at least 80 years with data. Moreover, the study would then be closer to Müller (2011), who selected tide gauges with at least 35 years of data prior to the year 1980, leading to 17 stations. Finally, shorter series are affected by stronger correlations between the NAO and the MSL (as they increase since 1960), which is problematic for the statistical analysis, when fitting model 2 to distinguish the influence of the MSL from that of the NAO. From 1910, only 5 stations out of 12 show a significant correlation between MSL and NAO (on average, r=0.28), whereas from 1960, this figure jumps to 9 stations, with higher r-value (on average, r=0.40).

However, to confirm the results on the European Shelf, we led a similar analysis on Calais and Dunkerque stations, located in the North of France (North Sea). Calais starts in 1941, with only two years 1941-1942, and then a gap until 1965, and data from 1965 up to now. Dunkerque starts in 1956. The results confirm that the variations at these two stations are similar to the variations at the 3 other stations in the North East Atlantic (Newlyn, Brest, Cuxhaven, Figure 3(a)). M2 increases from 1960 to 1990, and then becomes more steady since 1990. Similarly to the three other stations in North East Atlantic, the trends are decreasing when they are computed only since 1990. However, the main difficulty with these short time series (since 1960) is that the correlation between NAO and MSL is significant (see above). For example, at Dunkerque, the correlation coefficient is 0.53.

Finally, note that following reviewer #2 suggestion, we select now time records starting before 1930 (instead of 1920), which led to add 3 new stations (New York, Boston, Eastport).

**[-] The Introduction leaves a lot to be desired. It is incoherent, lacks any quantification as to the size of observed tidal changes and does not tell the reader why he/she should bother. A very good example of clarifying the relevance of this subject matter up front is Mawdsley et al. (2015, https://doi.org/10.1002/2014EF000282).**

We thank the reviewer for the reference. Following the reviewer suggestion, the introduction has been substantially rewritten.

**Minor comments (most of these issues are indications of the authors' unsteadiness regarding the physics of tides):**

**[-] The Introduction's first sentence is wrong. Tides have been changing also prior to 19th century, e.g., due to Earth's continental and glaciation cycle.**

We agree that the tides have been changing also prior to 19[th] century (Haigh et al., 2020), this first sentence may be read in the context of the paper, which concerns the period "from 1846 to 2018". We have rephrased to be more precise, following the Editor suggestion: "Tides have been changing due to non-astronomical factors since the XIX[th] century (Haigh et al., 2019; Talke and Jay, 2020)."

**[-] The main tidal constituents in the North Sea are presently not in a state of resonance (as argued on lines 172 and 219). They are rather described as Kelvin waves, dampened as they propagate from the Northwest through the basin in cyclonic fashion.**

We mentioned line 172 of the submitted paper, that amplification could be due to resonance effects and/or propagation in shallow waters. We agree with the reviewer, that the amplification of M2 in Cuxhaven is probably due to propagation in shallow waters rather than resonance (we corrected this point). Cuxhaven is located in the German Bight, shallow depths and shape of the coastline could induce some amplification. Variations in M2 at Cuxhaven are therefore very sensitive to local effects, as the migration of the underwater channels and the evolution of the tidal flats (Jacob et al., 2016). Moreover, Cuxhaven is located in the Elbe estuary, and some river engineering works, as narrowing and deepening, may induce tidal amplification (Wintewerp & Wang, 2013; Wintewerp et al., 2013) (see suggestion from reviewer #2). We have rephrased, mentioning a possible amplification due to resonance effects (e.g. Portland) and/or propagation in shallow waters (e.g. Cuxhaven), in addition to local effects.

**[-] Lines 250–252: First, stratification will not only change in response to heat fluxes, but also due to the advection of water masses, evaporation, salt dilution, etc. Second, in a discussion of stratification effects on "tides", one must use very precise language, in particular distinguish between barotropic, baroclinic, and surface (barotropic + baroclinic) tides. Third, the process identified by Kang et al. (2002) as cause for tidal seasonality in the Yellow/East China Sea is mixing strength (changes of vertical eddy viscosity) and not barotropic-to-baroclinic energy conversion.**

We agree with the reviewer that our description of the links between climate and tidal indices was somewhat sketchy, and this paragraph has been partially rewritten. Ocean and atmosphere are fully coupled, and air-sea fluxes are responsible for the exchange of momentum, water (evaporation and precipitation budget) and heat at their interface. Among the wide range of possible interactions, two mechanisms have been explored for their ability to modify the tide. (1) The momentum flux (wind stress) and the gradient of sea level pressure which acts on the barotropic tide and (2) the water and heat fluxes which induce changes in both temperature and salinity distribution in the ocean. The later effect acts on the stratification which itself could impact the tide in two different ways. The first way is the internal tide generation which transfers energy from barotropic and baroclinic

motion and modifies surface tidal expression (Colosi and Munk, 2006). However, in the present study, most of the observations comes from coastal stations sheltered by wide continental shelves which dampen internal waves and this effect has not lead, so far, to much attention in the North Atlantic. More important is the second effect: the impact of the stratification which acts on the eddy viscosity profile by modifying currents profile and bottom drag over continental shelf and ultimately modifying the M2 surface expression (Kang et al, 2002; Müller, 2012; Katavouta et al., 2016).

**[-] Lines 231–234: The simulations of Pickering et al. (2017) show exactly the opposite of what is described here (that is, their Figure 1a highlights an M2 increase in the German Bight, not a decrease).**
Yes, we thank the reviewer for this remark, it has been corrected. However, what we wanted to point out here (see the following sentence), is that Pickering et al. (2017) showed that the effect of mean sea level rise is opposite between the western part of English Channel and German Bight (decrease/increase), which is not consistent with observations, as M2 varies the same way at the stations located in these areas (Figure 3 (a)). This supports the idea that MSL rise is not sufficient to explain alone the secular changes in tide (and/or that simulations have strong uncertainties).

**[-] I understand the pragmatic approach of normalizing M2 changes to show results from different stations in one plot, but it would still make sense to include some absolute numbers (e.g., by using text or secondary Y axes) to facilitate quantitative comparisons among stations and allow for a meaningful interpretation of results derived later on.**
We understand the reviewer concern, but it is quite difficult to add a secondary Y axis and keep a readable figure. To follow the suggestion, we propose to add in the Appendix a new figure displaying the variations of M2 (without normalizing) at all the stations.

**[-] Annual tidal harmonics are computed from data spanning full years, but the discussion of atmospheric pressure changes only focuses on snapshots from winter months – another inconsistency in the analysis. Surely, if annual averages of atmospheric pressure fields are considered, the magnitude of the static sea-level response would decrease even further.**
We agree with the reviewer that we used only winter AO and NAO indices, which show more variability than annual indices. We conducted a similar analysis with annual indices. Preliminary results show consistent results for the correlation with AO: positive correlation on the east side and positive and negative correlation on the west side, depending on the stations. However, we noticed that 4 stations show no significant correlation using annual values, against only 2 using winter values. For the correlation with NAO, the number of stations without significant correlation jumps from 2 to 7. As underlined by the reviewer, with annual rather than monthly indices, the difference of pressure fields will decrease (being roughly divided by 3 following our results), and as a consequence, the magnitude of the sea-level response will also decrease. This point has been added in the limitations of the paper.

---

## Author Comment (AC3) · 14 Sep 2020

**Response to Editor Comment #1 (Philip Woodworth)**

We thank the Editor for his comments and detailed technical corrections. The Editor comments are in bold, our replies are in normal font.

**I have some detailed, most technical comments on this draft with an editor hat on. The two formal reviews should be inspected for more on the science. There are rather a lot of them but many are trivial.**
**Philip Woodworth**

We carefully answered to the two formal reviews, the details are in the response to comments.

**Title - what does 'climate-scale' mean? At line 9 you refer to 'large-scale' which seems a more sensible description. Or 'basin-scale' maybe.**
We meant at large timescale (decadal to secular). We agree that large-scale is here more relevant. Following Editor suggestion, we changed the title to "large-scale", instead of "climate-scale".

**6 - The trends in M2 amplitude**
It has been corrected.

**7 - from one station to another**
It has been corrected.

**7 - 0.7 mm/yr in the period since 1910**
It has been corrected.

**10 - distribution of water level**
It has been changed.

**14 - Tides have been changing .. factors since the XIXth century ..**
It has been changed.

**22 - large-scale (cf. Line 9)**
It has been corrected.

**31 - scale –> scales**
It has been corrected.

**32 – ditto**
It has been corrected.

**43-44 - I would drop this sentence**
The sentence has been removed.

**45 - thus only an accounted for change in …**
This sentence has also been removed, as it was linked to the previous one (which has been removed).

**47 - years of data**
Here, we meant 80 years with at least 75% of data (to avoid the seasonal effect – see below). To be clearer, we corrected "years with data" by "years of data", and added that we selected only years with at least 75% of data.

**49 - On the east side of the North Atlantic**
It has been added.

**50 - due to too small an M2**
This has been corrected.

**52 - On the west side of the North Atlantticc**
It has been added.

**52 - due to too small a tidal**
It has been corrected.

**55 - in Figure**
It has been corrected.

**60 - synthesised –> summarised**
It has been corrected.

**61 - in 1846 and 1896 respectively**
It has been added.

**64 - performed in order to compute**
It has been added.

**65 - similar to the**
It has been added.

**66 - I would drop 'largely .. community'. It may be true but its wide use is not relevant.**
It has been deleted.

**67 - of the yearly**
It has been corrected.

**69 - M2 correctly.**
It has been corrected.

**70 - seasonal variation of typically a few ..**
**A better reference for this would be:**
**Pugh, D. T. and Vassie, J. M. 1976. Tide and surge propagation off-shore in the Dowsing region of the North Sea. Deutsche Hydrographische Zeitschrift, 29, 163–213, doi:10.1007/BF02226659.**

We added the reference Pugh and Vassie (1976).

**71 - instead of the 50% here**
It has been corrected.
**Table 1 caption line 1 - tide gauge records selected**
It has been corrected.

**Table 1 caption line 3 - .... modulation, estimated trends in M2 amplitude since**
It has been corrected.

**Table 1 - I don't see why you have column 5 (MSL average) which has no importance to this study, MSL being measured relative to an arbitray datum at each site.**
The column 5 was here to give the average MSL over the period 1910-2010, but we agree with the Editor that is has no importance, as the reference is an arbitrary datum at each side. Reviewer #2 also made a similar comment. This column has been removed.

**73 - lead to the exclusion of more years.**
**This is obvious isn't it? So how did your results change with 75%?**
As suggested by the Editor, we tested with 75% instead of 50%. This led to exclude on average 2.92 supplementary years per station. Results are quite similar, but some values of M2 that showed some significant variation are now removed (e.g. first value of Newlyn in 1915 on Figure 3 (a)). We now adopt this new value of 75% (instead of 50%), to avoid the seasonal variation.

**74 - retrieved –> removed**
It has been corrected.

**(Simon, 2007,2013) as described briefly below.**
It has been added.

**Drop 'Here .. method'**
It has been removed.

**75 - reword:**
**.. an 18.6 modulation, separated from a neighbouring line in the tidal potential (m2) whose Doodson number differs in its 5th frequency ... respectively) (cf. Doodson and Warburg, 1941; Pugh and Woodworth, 2014). This .., the negative of the ..**
It has been rephrased and the references were added.

**79 - but it is negligible, its amplitude in the tidal potential being ..**
It has been corrected.

**81 - .. and m2 cannot be separated by a yearly harmonic ..**
It has been corrected.

**83 - expressed schematically**
It has been added.

**86 - are the amplitude and phase lag [not phase shift]**
It has been corrected.

**88 - shift –> lags astronomic –> astronomical is given by**
It has been corrected.

**93 - The negative of the mean ... is expressed simply**
It has been corrected.

**99 - from one station .. We added the default ..**
It has been corrected.

**103 - to this detrended**
It has been corrected.

**110 - drop the comma**
It has been removed.

**113 - please replace the hyphen with a colon. A hyphen looks like a minus sign.**
It has been replaced.

**Fig 2 caption. This should better say:**
**(a) Estimation of the modulation of M2 amplitude (mean removed) at Newlyn, (b) Impact of M2 amplitude …**
We corrected with the following caption "(a) Estimation of the nodal modulation of M2 amplitude (mean removed) at Newlyn (b) Impact on M2 amplitude of the nodal modulation correction at Newlyn".

**123 - ... (NAM) (Hurrell reference). These climate …**
It has been changed. We referred to Thompson and Wallace (2000), and Thomson et al. (2000).

**125 - stations) over long periods**
It has been changed.

**127 - what does 'Variations in the NAO are essential' mean? You mean important?**
Yes, we meant that the variations of NAO are important as they drive the climate variability. It has been rephrased following reviewer #2 suggestion "Variations of NAO drive the climate variability..."

**132 - The normalization involves ..**
**drop 'long-term'**
It has been corrected.

**134 - what do you mean by 'with yearly values' when you have said you are using wintertime values? I would drop these 3 words**
We meant that we had one (winter) value per year. We removed this 3 words.

**This section should mention the AMO also as you use it below.**

As we focus in the present paper on pressure indices (NAO and AO), we finally did not mention the AMO in the paper.

**143 - the eastern**
It has been added.

**146 - consistent with the temporal coverage of the tide gauge measurements.**
It has been changed.

**153 - Brest and Newlyn**
We corrected "Brest, Newlyn and Cuxhaven" instead of "Brest/Newlyn and Cuxhaven".

**154 - drop the brackets**
It has been done.

**155 - .. changes must be at least ..**
It has been done.

**159 – flattened**
We corrected "is flattening" by "flattens" as suggested by reviewer #2.

**yet –> already**
It has been corrected.

**161 - of the tidal**
It has been corrected.

**Fig 3 - nice plot**
We thank the Editor for this comment.

**Fig 3 caption line 3 - The blue star in**
It has been corrected.

**164 - allow to confirm at larger timescale (?)**
We corrected "at larger spatial scale", instead of "at large-scale"

**174 - in Figure**
It has been corrected.

**175 - into two groups**
It has been corrected.

**177 – ditto**
It has been corrected.

**179 - drop 'globally' (twice). They are not global which means 'worldwide' to most people**
We removed "globally".

**181 - ditto**
**increases overall**
It has been corrected.

**182 - decrease, and since 1990 only**
It has been corrected.

**183 - one station**
It has been corrected.

**191 - which provides some confirmation of the hypothesis**
It has been corrected.

**192 - from the Brest**
It has been corrected.

**196 - drop globally**
**decreases overall**
It has been corrected.

**202 - synthesised –> summarised**
It has been corrected.

**204 - one station**
**drop globally**
**positive overall**
It has been corrected.

**207 - found previously**
It has been corrected.

**214 - Lewes? You must mean Portland?**
We mean Lewes, and not Portland. We are here mentioning the stations with negative trends since 1910, i.e. Lewes and Halifax – Portland has a positive trend. We rephrased to be clearer: "6 stations have post-1990 negative trends (Atlantic City, Lewes, Charleston, Brest, Newlyn, Cuxhaven), whereas only 2 stations have post-1910 negative trends (Halifax, Lewes)."

**the latest –> recent**
**and (2)**
It has been corrected.

**226 - in the tide**
It has been corrected.

**227 - mean sea level rise can result in an increase in M2 of .. of the MSL rise ... the same sign as mean ..**

It has been corrected.

**238 - define SONEL**
"Système d'observation du Niveau des Eaux Littorales" has been added.

**240 - falling slightly**
It has been corrected.

**241 - give reference to GoM land movements**
The reference here is SONEL website. This paragraph has been rewritten, as we now correct MSL for land movement (see model 1, in the response to reviews).

**fig 6 caption - remove (see Table 1 column 5) and remove that column - it has no importance.**
It has been removed in the caption, and column 5 of table 1 has also been removed.

**249 - catch –> account for**
It has been corrected.
**when they are forced with a meteorological field. What does this mean?**
Huess and Andersen (2001) run a barotropic model in the North Sea, forced (1) with tides only and (2) with both tides and meteorological fields. Results show that the seasonal modulation is better captured when the model is forced with both tides and meteorological fields (Plate 2, top right, amplitude higher that 10 cm in the German Bight) rather than with tides only (Plate 2, top left, amplitude lower that 5 cm in the German Bight). We rephrased to be clearer, "Huess and Andersen (2001) showed that simulations better account for the seasonal variability of M2, when they are forced with tides and meteorological fields, rather than tides only."

**250 – affect**
It has been corrected.

**254 - in long-term**
We have rephrased following reviewer #2 suggestion "Ray and Talke (2019) suggest a possible role of stratification on secular tidal trends by long-term warming of the Gulf of Maine waters." This suggestion also keeps our aim to suggest a cause.

**256 - and the Atlantic ..**
**The AMO is not referred to as an index in section 2.2.1. Also it is an SST index and not**
**an air pressure one**
As we focus in the present paper on pressure indices (NAO and AO), we have removed the AMO analysis from the paper.

**Fig 7 caption line 1- you said before you used wintertime values not annual ones**
By "annual", we meant that we had one value per year, which is confusing. We corrected with "winter values" instead of "annual values".

**271 - .. could be due to differences in the spatial**
**heights –> level**
It has been corrected.

**273 - year –> winter**
It has been corrected, as well as line 276.

**276 – usual**
It has been corrected.

**278 distributed differently ... south**
We corrected with "southerly" instead of "southern", as suggested by the reviewer #2.

**281 - height –> level**
It has been corrected.

**287 – volume**
It has been corrected.

**288 -preciptation**
It has been corrected.

**289 - on the scale**
It has been corrected.

**Fig 8 caption line 1 - .. pressure over the NE Atlantic**
It has been added.

**297 - coldest –> lowest sea surface temperatures were ..**
It has been corrected.

**Fig 9 line 1 - Changes in mean sea level due to the difference ... (NAO-) assuming an IB response of sea level.**
Figure 9 shows now directly the difference of winter sea-level pressure, expressed in hPa (see response to reviewer #1). However, as suggested by the Editor, we rephrased the line 281 of sumitted paper: the changes in terms of water level may vary from -21 cm to 12 cm, assuming an inverse barometer response of sea level.

**307 – differences**
**calls for –> indicates**
It has been corrected.

**311 - explain the variations alone.**
It has been corrected.

**314 - heights –> level**
It has been corrected.

**317 analysis of the phase lag**
It has been added.

**332 - right! See above. How does that change the results?**
The results are in the response of line 73 (see above). Finally, we removed this limitation, as we now select years with at least 75% of data.

**333 - deep –> fuller**
It has been corrected.

**336 - tide gauges are obviously on the coast! Drop that. Harbours is relevant.**
We rephrased, the tide gauges are located mainly in harbours.

**363 - the e20... refers to Haigh et al. This reference needs correcting.**
It has been corrected.

**Also you have some names with initials before the surname e.g.** Trimble.
It has been corrected.

**Finally, you might want to refer to Talke and Jay (Annu. Rev. Mar. Sci. 2020. 12:121–51) especially from the perspective of changing tides in estuaries such as Cuxhaven.**
We now refer to Talke and Jay (2020), see the response to reviewer #2.

---

## Author Comment (AC4) · 14 Sep 2020

**Response to Short Comment #1 (Haidong Pan)**

We thank Dr Pan for his comments on the paper.

The comments are in bold, our replies are in normal font.

**Understanding the long-term changes of tides is important and useful. This paper investigated long-term tidal change in the North Atlantic and discussed a possible underlying mechanism which I think is pretty interesting. I want to provide some suggestions which may further improve this paper.**
We thank Dr Pan for this comment.

**First, the title of this paper is 'Climate-scale changes of the semidiurnal tide.....', however, the authors only analyzed M2 tide but ignored S2 tide which is also important.**
We agree that the paper focus mainly on M2. However, S2 has not been ignored, and has also been analyzed. Results are briefly presented in the paper (but not shown), see line 319-322 of submitted paper: "Results show that S2 amplitude decreases at all the stations located in the North West Atlantic, and in contrast, tends to increase in the North East Atlantic (not shown). The large-scale decrease of S2 observed in the North West Atlantic is consistent with previous studies, e.g. Ray (2006) in the Gulf of Maine. Further investigations should be definitely conducted to extend this work to more constituents."

**In addition to, the gravitational forcing of S2, oscillations in barometric pressure, changes in ocean temperature, and onshore-offshore wind have also been argued as contributing to the sea surface variations at the S2 frequency (Feng et al., 2015). The non-gravitational component of S2 is called the radiational tide and its amplitude has been estimated to be 10–18% of the gravitational amplitude, depending on geographical region and the physical parameters concerned (e.g., pressure, wind stress, and/or thermal forcing) (Feng et al., 2015). It seems that S2 tide is more easily influenced by changes of atmospherical circulation than M2 tide. Thus, it is necessary to check whether the changes of S2 tide are similar to North Atlantic Oscillation (NAO) which will prove underlying mechanism proposed by this paper.**
We agree that at S2 frequency, there is a combination of gravitational and radiational tide (e.g. Simon, 2013), which is not the case at M2 frequency. For this reason, changes in S2 may have different origins than changes in M2. As mentioned in the paper, S2 decreases at all the stations located in the North West Atlantic, and in contrast, tends to increase in the North East Atlantic. This is different from the observed M2 changes, which is not surprising, for the following reasons. Firstly, S2 response may be different to M2 response, even with similar forcing. For example, Pickering et al. (2017) showed different response to a 2 m sea level rise scenario for M2 and S2 – see Figure 1 (a) and (b) from Pickering et al. (2017). Secondly, some changes in solar radiation could also affect S2 (and not M2). However, this is not in the scope of the paper.

**Second, this paper is very similar to Müller (2011) which found the rapid change in semi-diurnal tides in the North Atlantic since 1980. This paper seems to revisit the Müller's work and change 1980 to 1990. The authors need to clearly describe the difference of two papers.**
The present paper clearly differs from Müller (2011), and as suggested, we will point out more clearly the differences between the two papers.

The differences between Müller (2011) and the present paper are the following (1) Müller (2011) focuses on shorter records (tide gauges with at least 35 years of data prior to 1980) whereas we focus on longer records (tide gauges with at least 80 years of data, starting before 1930), (2) Müller (2011) show recent changes since 1980, whereas we show that the changes started long before the XX$^{th}$ century, and are not linear (which is consistent with previous studies, e.g. Pouveau et al., 2006; Talke et al. 2014; Talke et al. 2018; Ray & Talke, 2019). (3) Müller (2011) already proposed a possible influence of NAO, but without going further in the description of the physical mechanism (this is mentioned line 269 of the submitted paper), whereas we propose a possible underlying mechanism (effect of large-scale atmospheric circulation). (4) We bring now quantitative insights on the possible influence of NAO, which was mentioned by Müller (2011) on the basis of quantitative criteria (see response to reviewers #1 and #2).

**By the way, this paper calculates the post-1990 trend and post-1910 trend to show the rapid change in M2 tide since 1990. I think post-1990 trend is meaningless because the length of post-1990 records is too short. I think that you can calculate the trend of 1910-1990 and post-1910.**
Post-1990 records may be considered as long enough to compute trends. Note that the sea levels have been measured by altimeters only since the 1990s, and post-1990 mean sea level rise is estimated at around 3.1 mm/year (Dangendorf et al., 2017; Meyssignac and Cazenave, 2012). However, we agree that trends computed on such a short period have to be interpreted carefully, and this is mentioned lines 221-223 of the submitted paper: "The trends have to be interpreted very carefully as the M2 variations are not linear, and may increase or decrease depending on the years; as a consequence, the estimated trends depend strongly on the period considered to estimate it. The interannual variability also plays an important role, and when substantial, trends can vary depending on the computational period."

**At last, although it seems that M2 variations are similar to NAO, it is very difficult to prove statistical validity since the data are too short. The authors should point out this in the paper.**
We went further in the data exploration to link the M2 variations with the NAO. We have now some new insight from the statistical analysis. These new results are detailed in response to reviewers #1 and #2.

**Reference:**
**Feng, X., M. N. Tsimplis, and P. L. Woodworth (2015), Nodal variations and long-term changes in the main tides on the coasts of China, J. Geophys. Res. Oceans, 120, 1215–1232.**
**Müller, M. (2011), Rapid change in semi-diurnal tides in the North Atlantic since 1980, Geophys. Res. Lett., 38, L11602.**

---

## Author Response (AR1)

**Response to reviewers**

11 October 2020

The authors addressed all the comments in the detailed point-by-point responses, that are online since 14 September 2020. In the revised version of the manuscript, the additional changes are as follows:

1) There is a substantial increase in the number of long-term station records (18 stations instead of 12):

- We select now tide gauges with time series starting no later than 1940 (instead of 1930). This led to 4 additional stations: Newport, New London, Wilmington and Fort Pulaski.

- Two more stations were added in the North Sea: Hoek van Holland and Delfzijl. They were provided by Rijkswaterstaat (RWS). Note that we do not mention anymore the results at Dunkerque and Calais, as Hoek van Holland and Delfzijl are also located in the south of the North Sea, with longer and more interesting time series.

2) The statistical model 2 is based on a rigorous multiple linear regression with two parameters, NAO and MSL. The introduction of NAO in model 2 increases its predictive performance, beyond the inherent effect of adding a regression parameter. Indeed, on average, the Akaike Information Criterion (AIC) is 106.3 for model 2, instead of 118.6 for model 1.

3) We have moved the nodal modulation part to Appendix A. As it is now in Appendix, we have not shortened the text.

The authors would like to thank the reviewers and Editor for their constructive comments. We tried to do our best to implement them, and we hope that the revised version of the manuscript meets now the high quality standards of Ocean Science.

[revised manuscript text omitted]
". These strong discrepancies between simulations and observations have to be carefully interpreted. (1) Numerical simulations are great tools to perform sensitivity studies and understand processes, but results in quantitative terms may be far from ground truth for many reasons as wide spatial resolution ($\sim$10 km in Schindelegger et al. (2018)), coarse bathymetry, rough parameterizations, tuning parameters, inadequate forcing, lack of coupling (e.g. with atmosphere). (2) The large discrepancies between the simulations and the observations also strongly suggest that MSL rise is not the only process

280 that may explain $M_2$ changes – other large-scale processes, in addition to local processes, may also play a role.

285

 The MSL obtained from tide gauges include a solid Earth component as they are referenced to

290 the land. Consequently, if the land is subsiding,  MSL as observed with a tide gauge will increase (Wöppelmann and Marcos, 2016). Estimates of vertical land motion from SONEL (Système d'Observation du Niveau des Eaux Littorales, www.sonel.org, Santamaría-Gómez et al. (2017) ) show that the stations considered here are quite stable or falling slightly in the North East Atlantic (i.e. vertical land movements smaller than 1 mm/yr). In the North West Atlantic, they are  falling more strongly  (i.e. trends  up to -2 mm/yr),

295  except in the Gulf of Maine, where the land tends slightly to rise. Note that these trends are computed on relatively short periods (i.e. generally < 15 years), making it difficult to infer robust trends over the last century.

Figure 5 shows the annual MSL (after removing the average over the period 1910-2010, and filtering with a 9-year time windows), with and without land movement correction. The correction is applied linearly from SONEL estimates (Santamaría-

300 Gómez et al. (2017) solution), and leads to more consistent MSL trends at the basin scale (Figure 5 (b)). Note that in the following, MSL is systematically corrected for land movement. The correlations between $M_2$ and MSL indicate that $M_2$ varies strongly with MSL (see section 4.2). However, $M_2$ variations show some variability in the North East Atlantic (Figure 2 (a)), which may not be explained with MSL rise alone.

**4.2 Possible link with climates indices**

[revised manuscript text omitted]

To go further in the relative contribution of MSL and NAO in $M_2$ variability, we fitted two linear regression models on $M_2$ variations. In the following, $M_2$, MSL and NAO are filtered over 9-year time windows and normalized. We did not consider stations without $M_2$-NAO correlation (Boston, New London, Charleston, Fort Pulaski, black dots on Figure 7). At all the other 
[revised manuscript text omitted]
 1910-2010 , (a) without land movement correction (b) with land movement correction. MSL are filtered over a 9-year time windows.

[Figure]

**Figure 6.**  Low frequency winter NAO and AO indices, obtained with a 9-year mean filter. Normalized annual $M_2$ amplitude in the North East Atlantic (from Figure 2 (a)) are also plotted in grey.

[Figure]

**Figure 7.** Correlation (r-value) since 1910 between $M_2$ and (a) North Atlantic Oscillation and (b) Arctic Oscillation. Black dots are stations with no significant correlation. $M_2$, NAO and AO are filtered on the same time window (9 years).

[Figure]

**Figure 8.** $r^2$-value since 1910 between $M_2$ and NAO, $M_2$ and MSL, $M_2$ and fitted model $\alpha MSL + \beta NAO$ (model 2), NAO and MSL. $M_2$, NAO and MSL are filtered on the same time window (9 years). Note that there is no orange bar for NAO-MSL correlation when correlation is not significant ($p > 0.05$).

[Figure]

**Figure 9.** Relative contribution of $\alpha$ compared to $\beta$ in the fitted model $\alpha MSL + \beta NAO$. Black dots are stations with no significant $M_2$-NAO correlation. The size of the pie is proportional to the correlation between $M_2$ and the fitted model. Stations with no MSL-NAO correlations are labelled in bold.

[Figure]

**Figure 10.** Variations since 1910 of $M_2$, $\alpha$MSL (model 1), $\alpha$MSL+ $\beta$NAO (model 2). $M_2$, NAO and MSL are filtered on the same time window (9 years).

[Figure]

**Figure 11.** Winter sea-level pressure over the North East Atlantic (a) average over 1850-20 (b) anomaly in 1969 (NAO$^-$) (c) anomaly in 1989 (NAO$^+$)

[Figure]

**Figure 12.** At decadal time scale: (a) Difference of winter sea-level pressure between 1989 (NAO$^+$) and 1969 (NAO$^-$) (b) Annual $M_2$ amplitude at Cuxhaven from 1918 to 2018. A seasonal time scale: (c) Difference of monthly sea-level pressure between January and July 2015 (d) Monthly $M_2$ amplitude at Cuxhaven from January 2010 to December 2015.

[Figure]

**Figure A1.** (a) Estimation of  the nodal modulation of $M_2$ amplitude (mean removed) at Newlyn (b) Impact  on $M_2$ amplitude of the nodal modulation correction at Newlyn

[Figure]

**Figure B1.** Annual $M_2$ amplitude at the 18 selected tide gauges

.

---

## Author Response (AR2)

**Response to Editor Review (Philip Woodworth, 2020/10/14)**
**2020/10/23**

We thank the Editor for his thorough review, comments, and suggestions. The Editor comments are in bold, our replies are in normal font.

In the following, the line numbers correspond to the previous submitted version.

**There are 2 main things I came across that worried me:**

**In Figure 12, your suggestion in (a) and (b) is that higher pressure in the south (in (a)), and the implied redistribution of water masses, leads to larger M2 (in (b)). ok, fine - that is what you are implying in Figure 6 (I think mention of (a) and (b) here should refer in the text back to Figure 6).**

**(I noticed that in your V1 you had figure (a) but said it was MSL and not air pressure - I assume you corrected that now.)**
Yes, it has been corrected.

**But then in (c) we have higher pressure in the south in Jan rather than July and yet M2 is larger in July. That is the opposite way round, or maybe I am missing something.**

**I thought at first that (c) might have been mis-labelled from July-Jan but I think it is correct – see for example Figure 3a of Chen et al., Tellus A, 49, 613-621. They show July-Jan but, allowing for the sign, it agrees with your (c). And as we know, during the winter (Jan) the Icelandic Low deepens a lot, so blue in the north ok.**
Yes, it has been correctly labelled.

**Anyhow, I am confused by all this. Could you investigate?**
Investigating further, the differences of pressure are opposite in the North Sea between Fig 12 (a) and (c). To be less confusing, Fig 12 (a) and (c) now focus on the North Sea, as well as the corresponding paragraph (lines 354-377).

**The second thing is that at line 273 you say the MSL in the following regressions was after VLM correction i.e. Figure 5(b). However, it seems to me that you should surely use the uncorrected MSL (Figure 5a), which is actual water depth depth of whatever origin (VLM or whatever, who cares?) and so has some physical logic to it, rather than corrected for VLM which is a more scientific concept to do with climate budgets etc. You clearly disagreed (line 273) so please say why you made this choice.**
We agree that this correction was pointless in the context of this study, and we now consider MSL without correcting for land movement. In the revised manuscript, the corresponding paragraph has been removed (lines 261-268). We have recomputed the two regression models. The figures 5, 8, 9 and 10 were updated.

**Some more detailed comments are as follows, many trivial:**

**7 - The secular trends ..**
It has been added.

**what does 'overall' mean. Perhaps 'most of them ..'**
Yes, we corrected with 'most of them'.

**12 - distribution of mean sea level (corresponding to water depth) from**
Yes, it has been changed.

**16 - have been observed**
It has been corrected.

**21 - a few**
It has been corrected.

**24 - a few ... in many estuaries**
It has been corrected.

**27 - such as changes**
It has been corrected.

**40 by Pickering**
It has been corrected.

**where does the -20 to 20 cm refer to, around the whole ocean?**
Yes, this refers to the whole ocean, it has been added.

**46 characteristics to**
It has been corrected.

**49 on the M2**
It has been corrected.

**60 mentioned**
It has been corrected.

**61 of the NAO**
It has been corrected.

**66 observed tidal changes**
It has been added.

**96 found no differences in the**
It has been corrected.

**97 using either T_Tide**

It has been added.

**98 temporal sampling was 3 hours**
It has been corrected.

**103 years --> station-years**
It has been corrected.

**104 ditto**
It has been corrected.

**116 centres**
It has been corrected.

**131 were low-pass filtered**
It has been added.

**135 We employed ... (20CR) data set version ???**
We employed the 20 CR version 3 dataset (this was mentioned in the Acknowledgements). We added the version in the text "20 CR version 3 dataset".

**137 sea-level pressure at each grid point**
It has been added.

**why do you use 2015 when you use 2018 for the tide gauge time series below?**
The 20CR version 3 dataset covers the period  1836-2015. For this reason, it was not possible to go until 2018, as for the tide gauges.

**139 pressure at each grid point.**
It has been added.

**144, 146 - what do 'very similar' and 'high correlation' mean? I think you have to refer the reader to the more quantitative statements below**
We have rephrased, mentioning that the variations show similar patterns at all the stations; M2 amplitude decreases up to the 1960s, then increases, and decreases again from the 1990s. We now refer to the 'similar patterns' between Brest and Cuxhaven (and not the 'high correlation').

**149 have noticed**
It has been corrected.

**152 and the shape**
It has been corrected.

**153 such as the migration**
It has been corrected.

**155 such as narrowing**

It has been corrected.

**157 that have gradually**
It has been corrected.

**I don't understand the point being made here, and also below. You are implying that most of the Brest data is now affected by engineering when it is was more open ocean before? In which case you can't really claim it as part of the wide-scale variability in the last century mentioned below? I think this needs rewording. Also I think you need to say here that your focus below will be on the 20th century (after 1910).**
Yes, we have rephrased, mentioning that the high values at Brest before 1910 may be due to local changes, in addition to large-scale changes. We also added that in the following, we will focus mainly on 20th century, as most of the stations start after 1900 (15 among 18 stations)

**164 - no obvious linear trend**
It has been added.

**line 1 Fig 2 caption - .. amplitude 9-year filtered .. The stars in (b) in the 1860s**
In Fig 2, normalized annual M2 amplitude is not 9-year filtered.
We added in the caption "in the 1860s".

**179 0.8 cm respectivey**
It has been added.

**189 in Figure ... in Figure**
It has been corrected.

**193 in Figure**
It has been corrected.

**195 only 0.36**
It has been corrected.

**17.5 cm mean amplitude**
It has been corrected.

**I don't understand the sentence 'The very slow ...'. You are talking about normalised amplitudes here.**
Here, we are talking about not normalised M2 amplitudes – see line 192 "The second outcome is that the rate of increase is very different from one station to another (keeping in mind that M2 is normalized by standard deviation on Figure 2)."
To be clearer, we rephrased "The very slow increase of M2...".

**198 Gulf**
It has been corrected.

**203 trends ... provides some consistency with the hypothesis ... formulated from the**

**analyses of Brest and otherdata in Figure 2(a) i.e. ..**
It has been corrected.

**But then see above, you seem to rule Brest out!**
Yes, we have rephrased "formulated from the analyses of the data prior to the XX[th] century in Figure 2 (a)".

**what is climate-scale? This seems to be from your old title.**
Yes, we have rephrased "long-term variations".

**204 introduces some breaks**
It has been corrected "long-term variations introduce some breaks...".

**210 I think a problem with Halifax is the big gap. Who knows what changes to instrumentation and the port etc. there were in between**
We added a sentence to take into account this comment.
"Note that at Halifax, there is a long gap in the data recording (1898-1919), which raises the possibility of an instrumentation origin in the observed decrease of the M2 amplitude."

**215 that the later results**
Yes, it has been added.

**253 - the normalised trends? But these are not shown in the Table. The unnormalised trends vary a lot so I don't understand line 254**
(Here, we are talking about lines 223 and 224 – and not 253 and 254.)
In the paper, we refer only to not normalised trends (Table 1, Figures 3 and 4). We have reformulated the line 224, to make it clearer:
"In the North East Atlantic, the trends are consistent with each other (in terms of sign), which is not surprising as the stations vary similarly (Figure 2 (a))."

**227 - I don't understand this 9 and 3, and you have two 'negatives'. You have 18 stations in the table and 5 are from NE Atlantic. How do you get to 12? It looks to me that 10 are negative and 8 positive**
Here, we are mentioning that there are more stations with negative trends since 1990 (9 stations with negative trends, see Table 1, column 8), than stations with negative trends from 1910 (3 stations with negative trends, see Table 1 column 7). We have reformulated to make it clearer:
"...9 stations out of 18 have post-1990 negative trends, whereas only 3 stations out of 18 have post-1910 negative trends (Table 1, columns 7 and 8)."

**231 The largest trends for 1910 onwards ?**
Yes, we added "since 1910".

**252 .. to 5 i.e. their ..**
It has been changed.

**255-258 - this sentence on modelling is just rambling and the mention of 10 km, coarseness and rough is all the same thing. I would drop (1), leave that alone, and just keep (2) which is true.**

We have dropped (1), and just kept (2).

**261 - The MSL records ... include**
This line has been removed, as we do not correct anymore for land mowement (see above the answer to the second main thing).

**273 see my second main question above. I am not convinced you have to make VLM corrections, and then you could drop figure 5b, but I realise you would have to recompute the regressions - but that is easy enough.**
Yes, we do not make VLM corrections anymore (see above the answer to the second main thing). We have recomputed the two regression models and updated Figures 5, 8, 9 and 10.

**276 MSL as well as climate indices**
The title of the section has been changed.

**273 - see above**
We did not correctly understand the suggested correction. Here, we mention that the correlations between M2 and MSL (alone) indicate that M2 varies with MSL. The climate indices will be introduced in the next section.

**277 - such as**
It has been corrected.

**281 act on**
It has been corrected.

**fig 5 caption line 2 - MSL values are filtered using 9-year windows.**
It has been corrected.

**294 on the tide**
It has been corrected.

**295 .. variability, the NAO index tends**
It has been corrected.

**fig 6 - I was intrigued that the NAO and AO indices have different magnitudes. Maybe ok.**
**But why don't you use normalised indices anyway?**
We plotted directly the indices in Figure 6, and we used normalized indices for NAO in model 2, to be consistent with the other parameters (M2, MSL) that were normalized. However, the differences between NAO and normalized NAO are small (mean 0.20, standard deviation 0.18), as NAO index is already the difference of normalized sea level pressure between the Azores and Iceland.

**Fig 6 line 1 - M2 amplitudes**
It has been corrected.

**301 are filtered using the same**

It has been corrected.

**Fig 7 line 2 - filtered using**
It has been corrected.

**313 - why do you exclude these 4 for having no NAO correlation? They would appear to be ideal candidates for model 1**
We have now added these 4 stations. There is no significative influence of NAO at Boston and New London. More surprising, there is some NAO influence at Charleston and Fort Pulaski (see updated Figure 8 and 9).

**321 - 0.7 does not sound like 'no significant correlation'**
Yes, we have rephrased. "We checked if there was correlation between NAO and MSL at the stations..."

**322 of the NAO**
It has been corrected.

**323 adding an additional regression ..**
It has been corrected.

**327 contribution of the NAO**
It has been corrected.

**328 whereas it is negligible**
It has been corrected.

**Fig 8 - see above - why don't you have the other 4 in this plot. Also I can't see one with a red bar of 0.7 mentioned above. Also please remove the 1.2 on the y-axis, the max for that is 1.0**
The 4 stations are now in the plot (see above).
0.7 is a linear correlation coefficient (r-value), whereas the variance explained by a given model (r2-value) is plotted in Fig 8.
The y-axis of Fig8 has been modified, and the max is now 1.0.

**line 2 - filtered using the**
It has been corrected.

**fig 9 line 2 - the size of each large dot**
It has been corrected.

**fig 10 line 1 - filtered using**
It has been corrected.

**329 sensitive to the NAO**
It has been corrected.

**332 at all four stations**

It has been corrected.

**The text here doesn't mention the Brest sub-figure**
Yes, we added that at Brest, the improvement is less significant.

**340 pressure gradient**
It has been corrected.

**358 - drop 'very'**
It has been corrected.

**360 They ran**
It has been corrected.

**371 - a few cm, similar to its variations seasonally?**
Yes, it has been added, « similar to its seasonal variations ».

**figs 11 and 12 - say in caption what the contour intervals are**
**fig 12 - see above**
We have added in the caption that the contour intervals are every 2 hPa (Fig 11) and 4hPa (Fig12 (a) and (c)), and we have labelled the contours on Figs 11 and 12.

**385 - whereas variations appear between stations in the ..**
It has been corrected.

**386 from one station to**
It has been corrected.

**395 on the M2 tide**
It has been corrected.

**398 and currents**
It has been corrected.

**405 studies (e.g. Ray, 2006) in**
We corrected 'studies (e.g. Ray, 2006, in the Gulf of Maine).'

**423 - why pick these 2 places and refer to Kemp? I would drop this. There must be many places you would like to look at**
We picked these 2 places, because they were mentioned by one of the reviewers, but we agree that other places could be mentioned. It has been removed.

**storminess on the tide**
It has been corrected.

**459 - space after am2**
It has been added.

**figure (a) - you should say in the caption or text why this is detrended and (b) isn't**

We have added in the caption, that M2 was detrended in (a) to better fit the nodal modulation.

**fig B1 - the delfzijl and hoek van holland have enormous increases in M2 around 1960 which look very odd. Is there no previous mention of this in the literature?**
**Adding them hasn't benefitted the paper much if they are so obviously different.**

Hollebrandse (2015) already mentioned a large tidal range increase at Hoek van Holland during the period 1960-1970. He reported that this increase was likely due to a combination of local changes (construction of the Europoort and the Maasvlakte and the extension of the breakwater Noorderhoofd) and larger scale processes, as 4 other neighbouring stations (Vlissingen, Burghsluis, Scheveningen and IJmuiden) show all a relative strong increase during the same period.

Hollebrandse (2015) also mentioned a large tidal range increase at Delfzijl, between 1960 and 1980. He attributed this to local changes (construction of a breakwater (1963-1966) and the damming of a harbour entrance (1978)).

Finally, the normalised M2 variations are not so different in the North East Atlantic, showing similar patterns (Figure 2 (a)). The addition of Delfzijl and Hoek van Holland in the review process allowed to confirm that changes are partly due to large-scale processes, in addition to local changes.

**references - some titles are not capitalised (e.g. Araujo and Pugh) but some are (e.g. Colisi and Munk). Some journal names are given in full and some are abbreviated. Please use the OSD style.**

It has been corrected. The titles are not capitalised and journal names are abbreviated. We have used the Latex bibliography style 'copernicus', as mentioned in the OSD Latex template.

[revised manuscript text omitted]
". ~~These strong discrepancies between simulations and observations have to be carefully interpreted. (1) Numerical simulations are great tools to perform sensitivity studies and understand processes, but results in quantitative terms may be far from ground truth for many reasons as wide spatial resolution (~10 km in Schindelegger et al. (2018)), coarse bathymetry, rough parameterizations, tuning parameters, inadequate forcing, lack of coupling (e.g. with atmosphere). (2)also~~ strongly suggest that MSL rise is not the only process that may explain $M_2$ changes – other large-scale processes, in addition to local processes, may also play a role.

~~The MSL obtained from tide gauges include a solid Earth component as they are referenced to the land. Consequently, if the land is subsiding, MSL as observed with a tide gauge will increase (Wöppelmann and Marcos, 2016). Estimates of vertical land motion from SONEL (Système d'Observation du Niveau des Eaux Littorales, www.sonel.org, Santamaría-Gómez et al. (2017)) show that the stations considered here are quite stable or falling slightly in the North East Atlantic (i.e. vertical land movements smaller than -1 mm/yr). In the North West Atlantic, they are falling more strongly (i.e. trends up to -2 mm/yr), except in the Gulf of Maine, where the 
[revised manuscript text omitted]

---

## Author Response (AR3)

**Response to Editor Review (Philip Woodworth, 2020/10/26)**
**2020/11/05**

We thank the Editor for his thorough review, comments, and suggestions. The Editor comments are in bold, our replies are in normal font.

In the following, the line numbers correspond to the previous submitted version.

**Many thanks for the latest version (V3) of the paper which I have read. Thanks for attending to the 'MSL without VLM correction' issue. I think that reads better now. But I have two remaining issues with the paper.**

**One issue is the usual set of small edits which I list below. These will not take you long. I am sure you are getting as tired of these things as I am but until the second issue is dealt with I may as well continue to send along these minor edits, which you can accept or not as you want.**
The small edits have been corrected – see below.

**But the second is, as I mentioned last time, to do with the discussion of Figure 12 and the air presssure pattern on seasonal timescales and its implications for M2 amplitude being the OPPOSITE of the relationship on interannual timescales.**

**I see that you changed Fig 12 (a) and (c) to focus on the North Sea area, ok. But the same problem remains, the inference of higher pressure in the south leads to larger M2 (as one learns from (a) and (b)) is OPPOSITE to the inference on seasonal timescales from (c) and (d). You have disposed of this problem with a minor edit to the text at line 361 to say 'and obtained values close to the ones in Figure 12 (a) in terms of magnitude (a dozen of hPa)'. Well, yes, they are similar in magnitude but not in sign!**
We agree that Fig 12 is confusing, and we propose to remove the seasonal analogy, i.e. Fig 12 (b), (c) and (d), as well as the corresponding paragraph « We conducted...available observations ». We only keep Fig 12 (a) (which becomes now Fig 12) on the North East Atlantic (instead of the North Sea).

The seasonal analogy was initially here to answer to the Referee Comment #1, who reported that the changes in sea level pressure were too small to cause the observed M2 changes. That's why we focused on the magnitude of the signal. The idea was to say that if at seasonal scale, a dozen of hPa may generate changes of a few cm, it could also be possible at decadal scale. However, this paragraph has now been removed.

We had differently interpreted Fig12 (b) (c) (d), and possibly misinterpreted, that's why we prefer to remove it. Our interpretation in terms of sign was the following: Fig 12 (c) shows on average a negative pressure difference in the North Sea (-3.6 hPa), which leads to low values of M2 in January, according to Fig 12 (d). By contrary, Fig 12 (a) shows in average a positive pressure difference in the North Sea (6.4 hPa), so we may expect high values of M2 in case of NAO+ (which is consistent with Fig 12 (b)). However, these figures have now been removed.

**Your problem to my mind is that you are trying to make too strong a connection between the air pressure patterns and M2. Let's leave Huess and Andersen aside for a moment. I am reasonably convinced by the work of Mueller (2012) and Grawe et al (2015) that the main reason for the**

**seasonal M2 variability is the seasonal stratification. They may be wrong, but they are the latest sets of work on this topic. If that is the case, then you can try till the cows come home (as we say) to make a connection between air pressures and M2 on seasonal timescales that will look anything like the pattern you show for interannual timescales (if also true). I hope you see what I mean. The seasonal M2 variability could still of course be connected to water depth of a similar magnitude as you have interannually but for entirely different reasons.**

The Fig 12 was here mainly to answer to Referee Comment #1. We agree that there could be other reasons than pressure differences to explain M2 variability, that's why we had mentioned at the end of the paragraph the hypothesis of stratification from Müller et al. (2014) and Gräwe et al. (2014). However, this paragraph has now been removed.

**So I think the para 'We conducted .. available observations' needs to be completely rewritten so show that you understand things seasonally and not dispose of Muller etc. at the end of the para as 'other hypotheses'. They are not - they are the best hypotheses at the moment. You should by all means mention the Huess and Andersen work of course, and keep all of Fig 12 if you want, but make it clear in the text that the simple association of patterns does not have the right sign and not try to hide that with disingenuous words like 'in terms of magnitude'.**

As mentioned previously, Figure 12 (b) (c) (d) and the corresponding paragraph "We conducted...available observations." have been removed. We have rewritten the previous paragraph from "These results suggest that a NAO-related mechanism may explain part of the variability of M2". We now first mention the hypothesis of stratification from Müller et al (2014) and Gräwe et al. (2014), and in a second time, we mention the Huess and Andersen work.

**I have actually spent some time on trying to understand why Huess and Andersen obtained maximum M2 in summer (which I assume is what their phase of 180 degrees means) by turning on winds for surges, which must be dominated by winter westerlies as in your Figure 2(c). I haven't fully understood all that and perhaps that can be returned to by someone else again, but for now I think you should not base most of your seasonal discussion on that, rather than Mueller etc.**

We believe that dedicated simulations should help to estimate the respective role of stratification and/or meteorological forcing (sea-level pressure and winds) in M2 variations.

**I hope you can be patient and address this issue seriously. Until you do that I do not feel I can accept the paper. I am conscious that I am functioning here more as a reviewer more than an editor, so I have copied this set of comments to the executive editor (as I did for my comments on the V2 version) and he can let me know if I am not doing my job correctly. I hope you can see that I am on your side here (because I am intrigued if there really is an NAO connection). However, I can also if you wish send the latest draft to the original two reviewers although I doubt if they will be pleased to see it again.**

We can see your helpful attitude, and we thank you sincerely, it is highly appreciated.

**Small comments on the text and figures:**

**Would you be willing to change the title to read '.... tide along North ...'? It is up to you but I think that would be better.**

Yes, it has been changed.

**line 1 - M2 along North ..**
Yes, it has been changed.

**7 - up to 2.5 mm/yr at Wilmington since 1910.**
It has been changed.

**19 - and Boston**
It has been added.

**20 - at the North Atlantic basin scale .... and at a quasi-global ..**
It has been changed.

**21 some coastal stations**
It has been added.

**23-24 I would remove this sentence. This example is from the Pacific which looks odd in a paper about the Atlantic. You do mention this paper lower down ok so it will be included.**
Yes, it has been removed.

**If you agree then: For example, Ray and Talke ..**
Yes, it has been done.

**32 - Cap --> Cape**
It has been corrected.

**45 – Hawaii**
It has been corrected.

**50 – waters**
It has been corrected.

**65 The first section below ['below' because this isn't the first section in the paper]**
It has been added.

**70 Tide gauge selection**
It has been corrected.

**73 by the Rijkswaterstaat (RWS) in the Netherlands.**
It has been added.

**77 - 249 from UHSLC**
It has been added.

**93 - .... in 1846, 1879 and 1896 respectively …**
It has been corrected.

**96 developed**
It has been corrected.

**99 - I guess this is optional at this stage, but don't you think 3 hour sampling is a bit coarse? Have you checked that this sampling results in the same M2 amplitudes as for hourlies?**
**You could test that using an hourly series and then have something like:**

**... was 3 hours. (We checked with an hourly time series from Brest, with a comparable M2 magnitude to the Netherlands, that 3-hourly sampling did not result in a reduction of M2 amplitude in a tidal analysis compared to hourly sampling.)**
**[If true of course]**
Yes, we agree that 3 hour sampling may be a bit coarse. As suggested, we have conducted tests, and we added the conclusions in the text "We checked with hourly time series from recent years (1971-2018), that 3-hourly sampling did not result in a significant reduction of M2 amplitude in a tidal analysis compared to hourly sampling." Indeed, at Hoek van Holland, we found on average a reduction of only 1.4 cm, whereas M2 varies from 71 cm to 80.5 cm over the whole period (Figure B1). At Delfzijl, we found a reduction of 2.4 cm, whereas M2 varies from 113.8 cm to 137.6 cm over all the period (Figure B1).

**100 .. seasonal modulation affecting the computed amplitudes.**
It has been changed.

**line 3 of Table 1 caption: ... since 1990 up to 2018 in each case (standard ..**
It has been added.

**125 - I would drop 'with yearly values'. What does that mean anyway, you are using winter values**
Yes, it has been removed.

**128 - the NAO index**
It has been corrected.

**137-140 - I would drop from 'We computed' to the end of the para and replace with:**

**However, we made use only of data from 1850 to be more consistent with the temporal coverage of the tide gauge measurements. This will be discussed in Section 4.**
It has been added.

**The reason for this is that the reader will ask why you defined winter as Dec-Feb when the NAO and AO indices are Dec-March. But I come back to this below.**
Yes, we mention later that we define winter as December-February, as suggested.

**142 - you use US spelling here (harbor) and UK spelling (harbour) at line 163 and elsewhere.**
**Please check you use standard spelling throughout (preferably UK).**
We corrected with "harbour" instead of "harbor" at lines 147 and 160.

**155 - .. Cuxhaven could therefore be sensitive**
It has been corrected.

**162 - would this link not be better as a reference in the normal way?**
Yes, we did it.

**168 - drop comma after result**
It has been removed.

**176 archaeology**
It has been corrected.

**182 .. magnitude of unnormalized M2 ..**
It has been changed.

**199-200 I still don't understand this sentence about small amplitude at Key West. You are discussing normalised changes so it shouldn't matter what the actual amplitude is.**
This sentence has been removed.

**209 between the 1870s**
It has been corrected.

**215 However, note that ..**
It has been changed.

**234 - (1) some spatially-coherent changes .. [you have two recents in the same sentence]**
It has been changed.

**238 - I think 1990 here should be 1910 or the paragraph doesn't make sense. Right? If so, I would move this para to line 230 so all the information for 1910- comes in two paragraphs, followed by the one ('The trends') on 1990-.**
Yes, we corrected 1990 to 1910, and moved the paragraph to line 230.

**264 - with 9-year**
It has been corrected.

**278 - current profiles .. shelves ... modifies**
It has been corrected.

**280 circulation on the tide**
It has been corrected.

**When you computed these correlations and the p-values did you allow for serial correlation?**
**If so, say so. If not, the correlations will be less significant than you think.**
Here, we did not investigate serial correlation. A Durbin-Watson test shows that there is some positive correlation in the residuals.

**295 - (the NAO index ..**
It has been corrected.

**300 - two indices are closely related.**
It has been changed.

**310 - NAO and MSL at each station: there ... below. [i.e. drop the brackets]**
It has been changed.

**324 - ... dependent) naturally captures better the ..**

**[this is an obvious statement!]**
It has been changed.

**325 observed since 1990**
It has been corrected.

**333 ... (Figure 11(b)). (We define winter here as December-February.) This way ..**
It has been added.

**335 .. onshore by westerly winds**
It has been added.

**338 I don't understand how you infer -15 to 24 from Figure 12(a). I see +/- 10 on average.**
This was a problem of update: the values in the text (-15 to 24 cm) still referred to the North East Atlantic, whereas the Figure 12 (a) was zoomed in the North Sea. As now, Figure 12 is back on the North East Atlantic, we do not need to change the values.

**342 .. yield centimetric changes in M2 amplitude.**
It has been changed.

**343 is roughly un agreement**
It has been added.

**347 Note that the M2 amplitude**
This paragraph has been removed.

**But the para 345- needs considerable revising, see above**
This paragraph has been removed.

**380 - since 1910 and since 1990**
It has been added.

**Fig 12 - see my comments above, but can the right plots be moved more right as the 'hPa' of the left is confusingly close to the 'M2(cm)' on the right**
Figures 12 (b) (c) (d) have been removed.

**395 tends**
It has been corrected.

**411 may not be**
It has been corrected.

[revised manuscript text omitted]

---

## Author Response (AR4)

**Response to Editor Technical Corrections (Philip Woodworth, 2020/11/10)**
**2020/11/16**

We thank the Editor for his technical corrections. The Editor comments are in bold, our replies are in normal font.

In the following, the line numbers correspond to the previous submitted version.

**line 276 - current profiles**
It has been corrected.

**292 - in your last response you said that you not take serial correlation into account but that a statistical test showed that it existed.**

**Could you please discuss this with Guy who understand statistics well? I think there is no impact on your presentation of actual correlation coefficients (or r or r-squared values as in Figure 7). The problem is that the computed p-values will be too optimistic. So I suggest to be clear that at line 292 you say something like:**

**As a rough test of significance, the correlations obtained are considered as significant only if a simply-computed p-value (without consideration of serial correlation) is lower than 0.05 (95% significance level).**

Following your suggestion, Guy has investigated the time-correlated noise in M2 series using the MLE technique implemented in SARI software (Santamaria-Gomez 2019; reference added to the list). The spectral index of the power-law noise is -1 (flicker noise) on average. Table 1 is now updated with error bars which consider the impact of the noise content on the uncertainty estimate of the linear trends. The manuscript is changed accordingly.

Besides Table 1 the changes are minor. The trend estimates remain statistically consistent with those found previously, for instance by Ray and Talke (2019), although our error bars are larger. Likely, past studies did not consider the noise content in the time series, or used another technique (spectral, AR or autocorrelation). In any case, it was worth carrying out this. Thanks for insisting.

The above study of time-correlated noise was of little use regarding the cross-correlation between M2 and NAO (AO) quantities. We found the classic Pearson's correlation coefficient (its uncertainty and p-value) appropriate to measuring the degree of association between these quantities, regardless of the source of correlation. We wonder what type of "noise" (signal) could be common between M2 and NAO (AO). This can be interesting to explore with more advanced tools; thus, we rather added a note on the perspective of exploring other types of correlation (e.g., Spearman for non-linear correlations) and modelling based on physics.

"The correlations are considered as significant only if the p-value is lower than 0.05 (95% significance level). (Note that other statistics to measure the degree of association between the M2 and NAO (AO) quantities would be worth exploring; for instance, nonlinear association using Spearman's correlation coefficient. In this respect, our study should be regarded as a first step that identifies sites worth considering in future investigations, especially investigating causal relationships with physics-based modelling.)"

**326-358 - I think this reads better than you had before. Thanks for revising this. However:**

**331 - you say that stratification is 'the main hypothesis invoked in Ray (2006) to explain secular changes in M2 amplitude in the Gulf of Maine'.**

**This is not correct. If you read his paper again you will see that Ray concluded he had no idea what could be causing the changes. The words 'stratification' and 'temperature' do not occur at all in the paper. However, stratification is indeed considered as (I would say) 'one of main possible hypotheses' and not 'the main' in Ray and Talke (2019). Please read this again to check their actual statements.**

Yes, it has been corrected. Apologies for this, here we referred to Ray and Talke (2019), and not Ray (2006). Moreover, we agree that this is not "the main" hypothesis, as Ray and Talke (2019) reported "The observations suggest that models seeking to reproduce Gulf of Maine tides must consider both sea level rise and long term changes in stratification." We have now rephrased : "This is one of the main possible hypotheses invoked in Ray and Talke (2019) to explain secular changes in M2 amplitude in the Gulf of Maine".

**line 350-1 - I still can't see how -15 to 24 cm relates at all to Figure 12, which has a maximum of 12 mbar, so that would correspond to +/- 12 cm, wouldn't it? I read your last response but I still cannot look at that figure and conclude -15 to 24. Apologies if I am misunderstanding something (but then so will other readers).**

To be clearer, we have rephrased referring now to the contours shown in Figure 12:
"Assuming an inverse barometer response of sea level, the changes in terms of water level may vary from more than 24 cm in the northwestern part of the area to around - 12 cm in the region that includes most of the northeast Atlantic tide gauges considered in this study."

**453 - the title of the appendix comes after the figure in that appendix.**
It has been corrected.

[revised manuscript text omitted]
 trend uncertainties were estimated considering the noise content in the time series using SARI software (Santamaría-Gómez, 2019). The noise was modelled as a white plus power-law noise, whose spectral index was found to be close to -1 (flicker noise). The results are summarised in Table 1 (columns 7 and 8) and Figures 3 and 4.

[Figure]

**Figure 3.** Estimated trends in $M_2$ amplitude over the period 1910-2018

[Figure]

**Figure 4.** Estimated trends in $M_2$ amplitude over the period 1990-2018

The trends estimated from 1910 vary significantly from one station to another (Figure 3). They are positive overall (up to 2.5 mm/yr at Wilmington), which is consistent with previous findings (Araújo and Pugh, 2008; Ray, 2009; Woodworth, 2010; Müller et al., 2011; Ray and Talke, 2019). They are slightly negative at three stations (Halifax, Newport, Lewes), and one

station shows no  trend (Atlantic City). The estimates are statistically consistent with those found previously by different authors (e.g. 0.14  $\pm$ 0.09  mm/yr at Newlyn compared to 0.19 $\pm$ 0.03 mm/yr in Araújo and Pugh (2008), 0.56 $\pm$ 0.06  mm/yr in Portland, compared to 0.59 $\pm$ 0.04 mm/yr in Ray and Talke (2019)). Note that our error bars are larger, because we considered the noise content in the time series as a white noise plus power law noise (we obtained the same error bars considering white noise only). 
[revised manuscript text omitted]